EMBO
Molecular Medicine

# Pre-clinical development of *AP4B1* gene replacement therapy for hereditary spastic paraplegia type 47

Jessica P Wiseman[1,2], Joseph M Scarrott[1,3], João Alves-Cruzeiro[1], Afshin Saffari [4,5,6], Cedric Böger[4,5], Evangelia Karyka [1,3], Emily Dawes[1], Alexandra K Davies [7,8], Paolo M Marchi [1], Emily Graves[1], Fiona Fernandes [1], Zih-Liang Yang[1,2], Ian Coldicott [1,2], Jennifer Hirst[8], Christopher P Webster [1,2], J Robin Highley[1,2], Neil Hackett [9], Adrienn Angyal[10], Thushan de Silva[10], Adrian Higginbottom[1,2], Pamela J Shaw [1,2,11], Laura Ferraiuolo[1], Darius Ebrahimi-Fakhari[4,5] & Mimoun Azzouz [1,2,3 ✉]

## Abstract

**Spastic paraplegia 47 (SPG47) is a neurological disorder caused by mutations in the *adaptor protein complex 4 β1 subunit* (*AP4B1*) gene leading to AP-4 complex deficiency. SPG47 is characterised by progressive spastic paraplegia, global developmental delay, intellectual disability and epilepsy. Gene therapy aimed at restoring functional AP4B1 protein levels is a rational therapeutic strategy to ameliorate the disease phenotype. Here we report that a single delivery of adeno-associated virus serotype 9 expressing *hAP4B1* (AAV9/hAP4B1) into the cisterna magna leads to widespread gene transfer and restoration of various hallmarks of disease, including AP-4 cargo (ATG9A) mislocalisation, calbindin-positive spheroids in the deep cerebellar nuclei, anatomical brain defects and motor dysfunction, in an SPG47 mouse model. Furthermore, AAV9/hAP4B1-based gene therapy demonstrated a restoration of plasma neurofilament light (NfL) levels of treated mice. Encouraged by these preclinical proof-of-concept data, we conducted IND-enabling studies, including immunogenicity and GLP non-human primate (NHP) toxicology studies. Importantly, NHP safety and biodistribution study revealed no significant adverse events associated with the therapeutic intervention. These findings provide evidence of both therapeutic efficacy and safety, establishing a robust basis for the pursuit of an IND application for clinical trials targeting SPG47 patients.**

**Keywords** AAV; AP4B1; Gene Therapy; HSP; SPG47
**Subject Categories** Genetics, Gene Therapy & Genetic Disease; Neuroscience

## Introduction

Hereditary spastic paraplegia type 47 (SPG47) is a rare childhood-onset neurological disorder and part of a larger group of complex forms of hereditary spastic paraplegias (HSPs). HSPs are characterised by dysfunction and degeneration of long axons in the corticospinal tracts leading to progressive lower limb weakness and spasticity (Finsterer et al, 2012; Kara et al, 2016; Shribman et al, 2019). Children with SPG47 present with progressive spastic paraplegia, global developmental delay and later intellectual disability, secondary microcephaly, early-onset epilepsy and cerebral dysgenesis including thinning of the corpus callosum (Ebrahimi-Fakhari et al, 2020; Ebrahimi-Fakhari et al, 2021a; Jordan et al, 2021). Disease-onset is typically in infancy and most children with SPG47 become fulltime wheelchair users by the age of 10 years (Ebrahimi-Fakhari et al, 2020). While the true prevalence of SPG47 is currently unknown, just over 100 individuals with SPG47 have enrolled in the International Registry and Natural History Study for Early-Onset Hereditary Spastic Paraplegia (NCT04712812) over the past four years.

SPG47 is caused by biallelic loss-of-function variants in the *AP4B1* (*adaptor protein complex 4 beta 1*) gene (Abou Jamra et al, 2011; Bauer et al, 2012; Ebrahimi-Fakhari et al, 2016; Ebrahimi-Fakhari et al, 2018b), one of the four subunits of the adaptor protein (AP)-4 complex, leading to a deficiency of the entire AP-4 complex. Similarly, pathogenic variants in any of the genes that encode the other subunits of AP-4 lead to the same functional loss and similar clinical phenotypes: namely SPG50 (AP4M1), SPG51 (AP4E1) and SPG52 (AP4S1) (Behne et al, 2020; Ebrahimi-Fakhari et al, 2021a). These manifestations of complex HSP in children are termed AP-4-associated HSP (AP-4-HSP).

AP-4 belongs to a family of adaptor proteins (AP-1-5) that selectively integrate transmembrane cargo proteins into vesicles through recruiting the necessary molecular machinery for vesicle

[1]Sheffield Institute for Translational Neuroscience (SITraN), Division of Neuroscience, University of Sheffield, Sheffield, UK. [2]Neuroscience Institute, University of Sheffield, Western Bank, Sheffield, UK. [3]Gene Therapy Innovation & Manufacturing Centre (GTIMC), Division of Neuroscience, University of Sheffield, Sheffield, UK. [4]F.M. Kirby Neurobiology Center, Boston Children's Hospital, Harvard Medical School, Boston, MA, USA. [5]Movement Disorders Program, Department of Neurology, Boston Children's Hospital, Harvard Medical School, Boston, MA, USA. [6]Division of Child Neurology and Inherited Metabolic Diseases, Heidelberg University Hospital, Heidelberg, Germany. [7]School of Biological Sciences, Faculty of Biology, Medicine and Health, Manchester Academic Health Science Centre, University of Manchester, Manchester, UK. [8]Cambridge Institute for Medical Research, University of Cambridge, Addenbrooke's Hospital, Cambridge, UK. [9]Independent Consultant, New York, NY, USA. [10]Department of Infection, Immunity and Cardiovascular Disease, University of Sheffield, Sheffield, UK. [11]Sheffield NIHR Biomedical Research Centre, Sheffield Teaching Hospitals NHS Foundation Trust, Glossop Road, Sheffield, UK. ✉E-mail: m.azzouz@sheffield.ac.uk

budding and transport (Hirst et al, 2013). AP-4 specifically mediates trafficking of cargo proteins from the trans-golgi network (TGN) to peripheral sites, including autophagy-related gene 9A (ATG9A), diacylglycerol lipase beta (DAGLB), serine incorporator 1 (SERINC1) and serine incorporator 3 (SERINC3) (Mattera et al, 2017; De Pace et al, 2018; Davies et al, 2018; Ivankovic et al, 2020; Davies et al, 2022). After budding from the TGN, AP-4 vesicles associate with microtubule transport machinery, specifically kinesin-1, for plus-end-directed transport to the cell periphery (Davies et al, 2018; Guardia et al, 2021). Deficiency in AP-4 has been robustly demonstrated to cause missorting and accumulation of ATG9A in the TGN in diverse cell types, including AP-4 deficient HeLa cells, patient-derived fibroblasts (Mattera et al, 2017; Davies et al, 2018; Ebrahimi-Fakhari et al, 2021b; Saffari et al, 2024) and iPSC neurons (Behne et al, 2020; Davies et al, 2022), as well as murine models of disease (De Pace et al, 2018; Ivankovic et al, 2020; Scarrott et al, 2023; Chen et al, 2023). The anterograde transport of ATG9A by AP-4-derived vesicles delivers ATG9A to the distal axon, which is an important location of autophagosome biogenesis (Maday et al, 2012; De Pace et al, 2018; Ivankovic et al, 2020). These ATG9A-positive vesicles cluster in close association with autophagosomes, suggesting they act as a reservoir for autophagosome biogenesis. Thus, missorting of ATG9A in AP-4-deficient cells results in an impairment of autophagy (Ivankovic et al, 2020; Mattera et al, 2017; Davies et al, 2018).

Over the last two decades, detailed molecular characterization, clinical assessment and diagnostic testing of AP-4 deficiency has supported a better understanding of disease phenotypes, manifestations and prevalence. Currently, there are no disease-modifying treatments and interventions are limited to supportive care, including physical, speech and occupational therapy, in addition to antispasticity/antiseizure drugs (Ebrahimi-Fakhari et al, 2018a; Ebrahimi-Fakhari et al, 2020).

Due to the monogenic aspect of *AP4B1*-related AP-4 deficiency, gene replacement is an ideal strategy for restoring gene function. Viral vector-based gene therapy can deliver the correct version of the gene (*AP4B1*) to the central nervous system (CNS) and thereby re-establish normal levels of the AP-4 protein and appropriate cellular localisation of ATG9A and any other pathways disrupted by AP-4 deficiency. The use of recombinant adeno-associated vectors (rAAV) as a delivery method is of particular interest as rAAVs do not code for viral proteins, are not linked to any known pathology, are nonreplicating and importantly display a non-integrating transduction. AAV-based gene replacement has already been approved by regulatory bodies for use in neurological disorders: Onasemnogen-abeparvovec (Zolgensma) for spinal muscular atrophy (SMA) (Mahajan, 2019) and eladocagene exuparvovec (Upstaza) for aromatic l-amino acid decarboxylase deficiency (Keam, 2022). This, in addition to a growing pipeline of AAV-based clinical trials, demonstrates that gene delivery can be safe and effective. AAV-based therapies for the CNS in preclinical and clinical studies utilise various AAV serotypes/variants, routes of delivery or maintenance methodologies (e.g., immunosuppressants) (Kang et al, 2023). AAV9 has become the preferred serotype for CNS delivery due to its increased capacity to mediate efficient neuronal transduction (Lukashchuk et al, 2016). Emerging novel AAV9 capsids led to improved blood-brain barrier (BBB) crossing and CNS targeting. However, further efforts are needed to define the safety profile of these new capsids before entering human

clinical applications. Furthermore, there are multiple ongoing clinical trials using the original AAV9 capsid and AAV9-based products that were approved by regulators including FDA and EMA (e.g., Zolgensma). The production of original AAV9 at good manufacturing practice (GMP) scale is well-established, an advantage essential for therapeutic approaches that lead to clinical translation (Adamson-Small et al, 2016). Taken together this evidence gave us confidence in the safety and efficacy of an AAV9-based approach which would facilitate regulatory approval to enter clinical trials in SPG47 patients.

In addition to vector serotype, optimising vector delivery routes is of particular interest to maximise widespread gene expression in desired locations. Intravenous (IV) delivery is a popular method of delivery and has been FDA approved for certain gene therapies (e.g., Zolgensma). However, despite the cell tropism of AAV9 to the CNS, IV delivery can lead to viral expression in various off target peripheral organs, particularly the liver and heart (Gray et al, 2011; Lukashchuk et al, 2016). For more direct delivery to the CNS, preclinical and clinical studies have tested routes such as intraparenchymal (injection directly to localised brain tissue) or intra-cerebrospinal fluid (CSF) delivery. Intra-CSF delivery includes lumbar intrathecal, intracerebroventricular (ICV), and intracisterna magna (ICM) routes which lead to various viral distribution patterns. Lumbar intrathecal delivery (via lumbar puncture) leads to viral distribution predominantly within the spinal cord and brainstem. ICV injection leads to spread through the supratentorial brain. ICM delivery can lead to more widespread CNS-AAV distribution, including high levels observed in the cerebellum, brainstem and spinal cord (Lukashchuk et al, 2016; Taghian et al, 2020; Marchi et al, 2022; Kang et al, 2023). Importantly, intracisterna magna injections can be done safely, including in young children (Samaranch et al, 2016; Katz et al, 2018; Taghian et al, 2020).

The current study evaluated the efficacy and safety of AAV9-mediated gene transfer of the human *AP4B1* (*hAP4B1*) gene in vitro and in vivo. This was achieved through a series of proof-of-concept experiments in *AP4B1* knockout HeLa cells, patient fibroblasts and iPSC-derived neurons, along with in vivo studies utilising a recently developed and characterised *Ap4b1*-knockout mouse model (Scarrott et al, 2023). We have investigated viral gene therapy efficacy through a long-term study after neonatal treatment (P2-3) and a dose-response study after treatment in adult mice (P60). We report here the rescue of key molecular, cellular, morphological and behavioural disease features observed in the in vitro and in vivo disease models. Completing the preclinical package, we carried out IND-enabling safety studies including Good Laboratory Practice (GLP) regulatory toxicology studies in non-human primates (NHPs), which provided evidence for a safe clinical development of AAV9/hAP4B1 administered via the ICM for patients with SPG47.

## Results

### AP4B1 gene replacement restored AP4B1 protein expression and ATG9A trafficking in AP4B1-knockout HeLa cells and iPSC-derived neurons from a patient with SPG47

Human *AP4B1* was packaged within an expression cassette comprised of two AAV2 inverted terminal repeats (ITRs) flanking

a Chicken ß-actin hybrid (CBh) promoter, the human *AP4B1* (*hAP4B1*) cDNA, and the human growth hormone poly (A) signal (hGH poly(A)) (Fig. 1A). Other expression cassettes consisted of the same AAV backbone with a V5 sequence (excluding transgene) in place of the *hAP4B1* gene sequence, or the human synapsin 1 promoter (SYN) in place of the CBh promoter sequence. We also generated lentiviral vectors expressing hAP4B1 (LV/PGK-hAP4B1) or eGFP (LV/PGK-eGFP) in order to transduce cell types that were not susceptible to AAV9. The rationale for using the CBh promoter in our AAV9 studies is due to its ability to drive robust and widespread transgene expression in the CNS, including neuronal and glial cells (Gray et al, 2011). Due to stronger expression of the transgene driven by the CBh promoter lower doses of the AAV9 therapy should be required in comparison to other weaker promoters (Lukashchuk et al, 2016). In addition, CBh was compared against a SYN promoter in later animal studies. SYN, a neuronal-specific promoter, was utilised as a contingency plan in case of any potential off-target effects that may have arisen when using CBh promoter.

As an initial proof of concept study for *AP4B1* gene therapy, we utilised an *AP4B1*-knockout (KO) HeLa cell model (Frazier et al, 2016) to investigate the effectiveness of different AAV-hAP4B1 expression vectors. Western blot analysis revealed a complete loss of AP4B1 protein in knockout cells (Fig. 1B). As described previously (Hirst et al, 2013; Scarrott et al, 2023), loss of AP4B1 led to a concomitant reduction in AP4E1 (AP-4 complex subunit), suggesting impaired function of the entire AP-4 complex (Fig. 1B). Transient transfection of these cells with AAV/CBh-hAP4B1 plasmids resulted in detectable expression of AP4B1 and a simultaneous restoration of AP4E1 levels (Fig. 1B). We encountered difficulties with the AP4B1 antibody and the AP4E1 antibody proved significantly more reliable. Consequently, we opted to utilise detection of AP4E1 protein levels to identify restoration of the AP-4 complex in all subsequent experiments. Having confirmed AP4B1 expression from these plasmids, we generated AAV9 viral vectors and transduced AP4B1-KO HeLa cells with two different multiplicities of infection (MOIs), $2 \times 10^5$ vg/cell and $4 \times 10^5$ vg/cell. As expected, immunofluorescent analysis of these cells showed that loss of AP4B1 caused an accumulation of ATG9A at the TGN (Fig. 1C). Compared to untreated or AAV9/V5-empty transduced cells, AAV9/CBh-hAP4B1 treatment led to a reduced ATG9A accumulation at the TGN, suggesting a rescue of ATG9A trafficking (Fig. 1C). This was further supported by western blot analysis of whole cell lysates from these transduced cells which demonstrated that transduction with AAV9/CBh-hAP4B1 at an MOI of $4 \times 10^5$ vg/cell led to a significant increase in detectable AP4E1 protein levels (suggestive of AP-4 complex formation) (Fig. 1D,E) from 18% ± 4% (V5-empty) to 54% ± 12% (hAP4B1 MOI $4 \times 10^5$ vg/cell) ($p = 0.0133$) relative to wild-type (WT) levels. These data indicated that restoration of AP4B1 protein levels via gene replacement approaches was sufficient to restore the AP-4 complex, leading to correct trafficking of ATG9A.

In addition, our gene therapy approach was tested in iPSC-derived neurons from a compound heterozygous patient with SPG47 (LoF/LoF (Loss of Function)) [*AP4B1*, NM_001253852.3: c.1345 A > T, p.(Arg449Ter)/c.1160_1161del, p.(Thr387ArgfsTer30)] versus a clinically-unaffected sex-matched parental control (heterozygous carrier) (LoF/WT). Due to low transduction efficiencies with AAV9 in these cells, we generated lentiviral (LV) vectors from the expression

cassettes described previously (Fig. 1A). Similar to AP4B1-deficient HeLa cells, patient-derived iPSC neurons present with an accumulation of ATG9A within the TGN (Fig. 2A), following LV/PGK-hAP4B1 treatment of varying MOIs (1, 5, 10, 20), fixed iPSC neurons were run through an automated immunofluorescence system detecting the ratio of ATG9A co-localisation with the TGN (Fig. 2A) versus cytoplasm (Behne et al, 2020; Saffari et al, 2024). The ratio of ATG9A inside the TGN in patient neurons was reduced significantly (MOI 0 (uninfected): 5.3 ± 2.3 SD, vs. MOI 1: 2.4 ± 1.1 SD) when transduced with LV/CBh-hAP4B1 (Fig. 2B). All MOIs significantly reduced the ATG9A ratio inside the TGN, to that of control levels indicating restoration of trafficking and redistribution of ATG9A ($p < 2.22e-16$ for MOI 1, 5, 10, and 20). While reduced cell numbers were observed at the highest viral dose (MOI 20), suggesting potential cell toxicity, this did not reach significance (Fig. 2C). In addition, gene replacement experiments with LV/hAP4B1 were carried out on fibroblasts from an SPG47 patient (LoF/LoF) compared with fibroblasts from an age matched healthy control (WT). Patient fibroblasts exhibited the hallmark phenotype and showed accumulation of ATG9A within the TGN (Fig. EV1A). Western blot analysis indicated the absence of the AP4B1 protein, while expression of ATG9A protein is elevated from WT levels (Fig. EV1B,C). LV/hAP4B1 administration increased AP4B1 levels and reduced ATG9A expression in patients' fibroblasts with a dose-dependent effect (Fig. EV1B,C). Taken together, data collected from these separate cellular disease models demonstrated that viral-mediated AP4B1 expression restored the AP-4 complex and rescued ATG9A localisation.

## ICM delivery of AAV9/hAP4B1 vector restored AP-4 complex in the brain and spinal cord of Ap4b1-KO mice

The AAV9 therapeutic vector was tested in vivo using the $Ap4b1^{-/-}$ (*Ap4b1*-KO) mouse model we described previously (Scarrott et al, 2023). We initially performed route of administration analysis to determine which delivery method was likely to give the optimal viral distribution and restoration of the hAP4B1 mRNA expression within the CNS. Two delivery routes were assessed: intravenous (IV, $4 \times 10^{13}$ vg/kg) and intracisterna magna (ICM, $4 \times 10^{13}$ vg/kg). Neonatal mice ($n = 3$ per group) were dosed at postnatal day 2–3 (P2-3) and animals were sacrificed at 2 months post injection for the evaluation of viral genome copies, transgene expression and protein expression throughout the mouse organs. Here we chose to investigate CNS organs: cerebrum, cerebellum and spinal cord, and peripheral organs: liver and heart. AAV9 mediates high gene transfer efficiency to the heart and liver when administered to mice. These two organs were chosen for our studies in mice because of their relevance in relation to reported safety concerns linked to AAV. Vector delivery was analysed through qPCR. ICM resulted in higher viral genome copies in the cerebrum, spinal cord and cerebellum compared to IV delivery (Fig. 3A). Both delivery routes displayed equal viral genome levels in the heart while animals with IV dosing displayed high viral presence in the liver (Fig. 3A). ICM delivery of the therapeutic vector also resulted in increased hAP4B1 mRNA expression in the CNS tissues (highest levels in the cerebellum and spinal cord) (Fig. 3B–D). hAP4B1 mRNA expression in the heart and liver were similar for ICM and IV delivery (Fig. 3E,F). Although IV delivery resulted in increased hAP4B1 mRNA levels in the spinal cord and cerebrum of treated mice, this was insufficient to restore formation of the AP-4

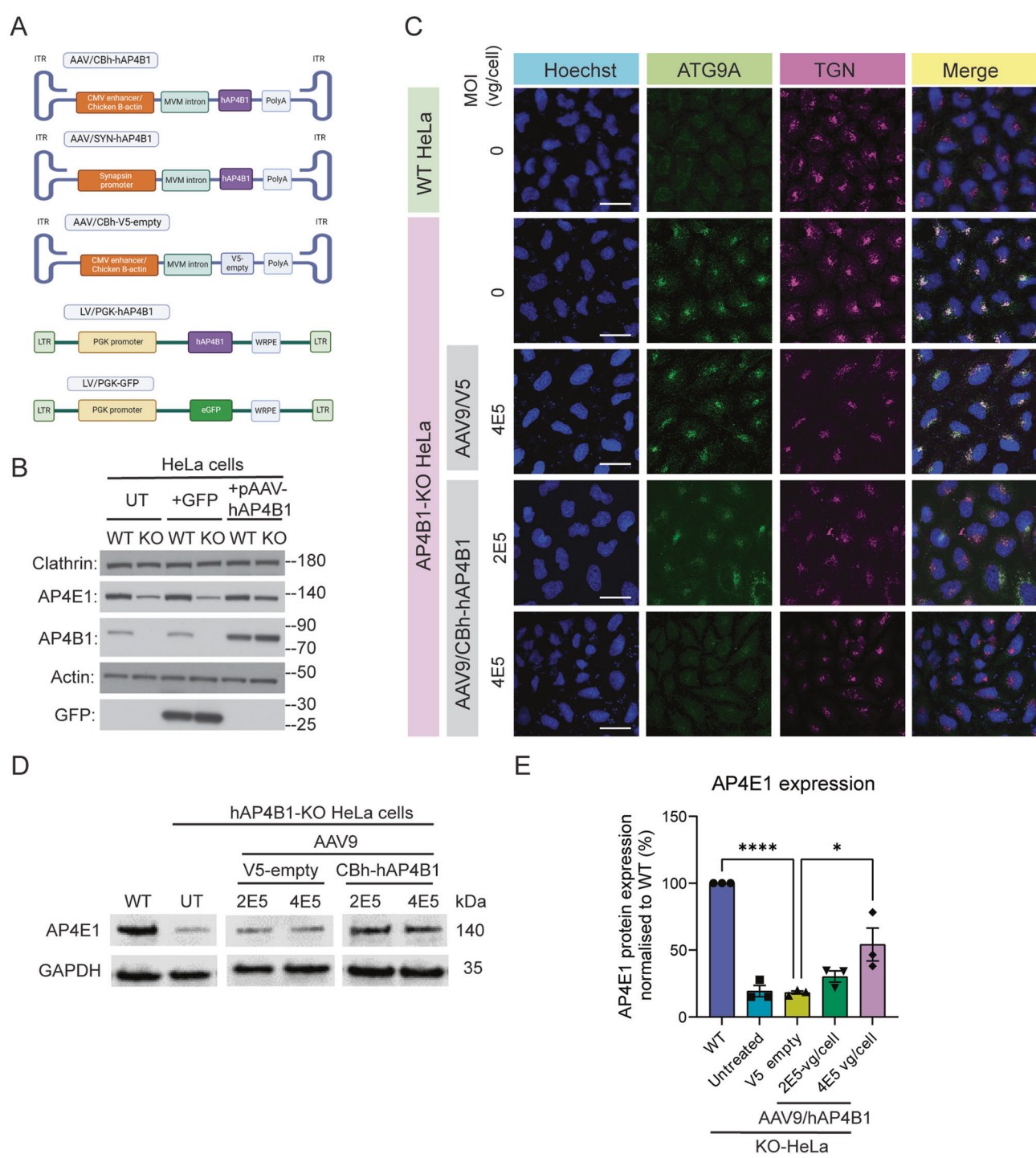

**Figure 1. In vitro development of hAP4B1 viral packaging and transduction of SPG47 cell culture models.**

(A) Schematic of the various packaging plasmids used within AAV9 or LV. (B) Western blot indicating the levels of AP4E1/AP4B1 within WT and AP4B1−/− KO Hela untreated or transfected with pGFP or pAAV-hAP4B1. (C) Fluorescent micrographs indicate AP4B1−/− KO HeLa's exhibit ATG9A accumulation in the TGN and this is reduced with AAV9/CBh-hAP4B1 treatment. Scale bar: 50 μm. (D) Western blot of detection of AP4E1 protein expression in Hela cells transduced with AAV9_CBh-hAP4B1 displayed significant increase of AP4E1 at 4E5 vg/cell multiplicity of infection (MOI) compared with control vector (AAV9/V5-empty). (E) Quantitation of AP4E1 expression level from Western blot ($n = 3$ biological repeats per group). Data are presented as mean ± SEM. Plot (E) was analysed by one-way ANOVA followed by Tukey's post hoc multiple comparisons test. Stars indicate $p < 0.0001$ (****); $p = 0.0133$ (*). Source data are available online for this figure.

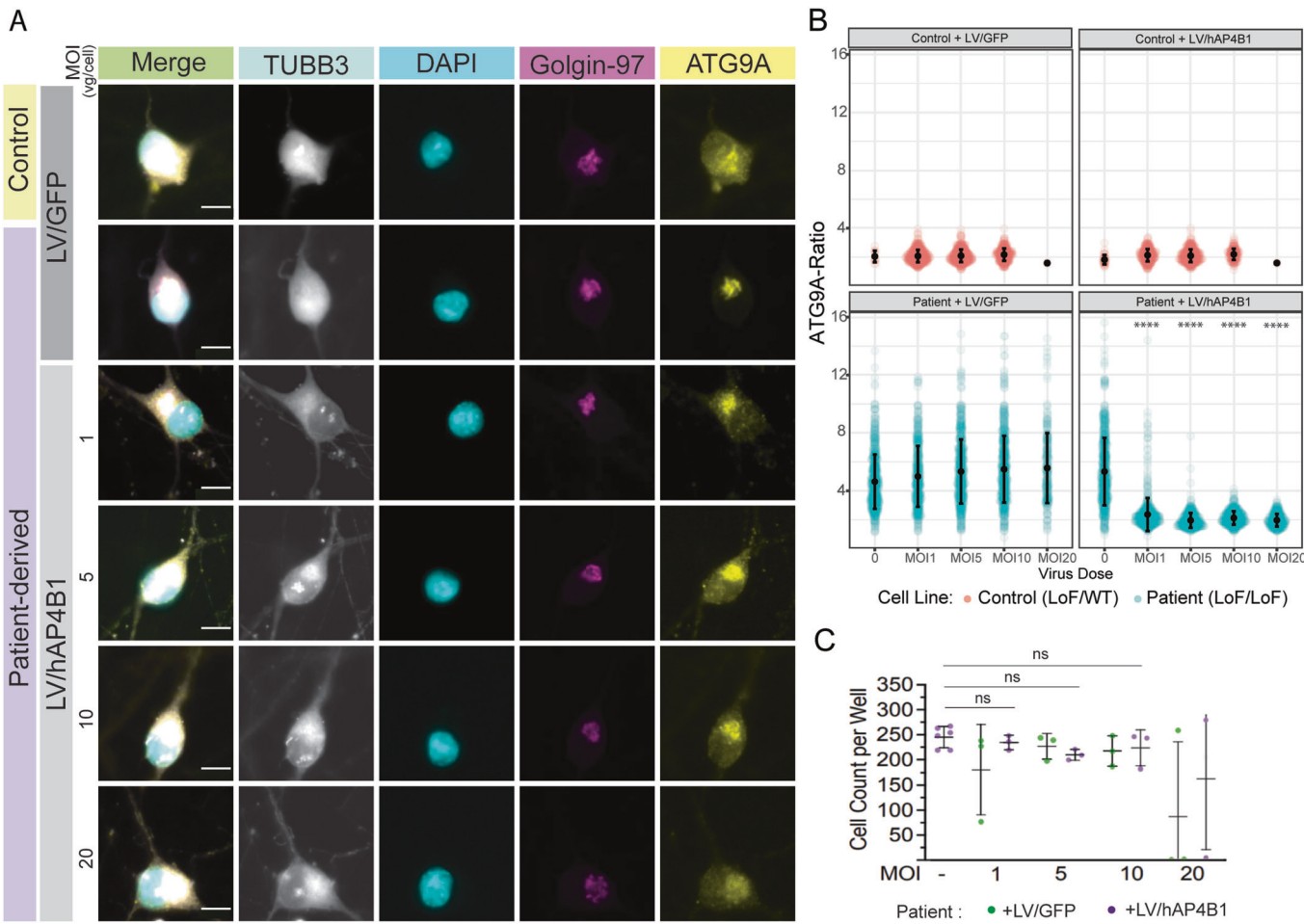

**Figure 2. LV/hAP4B1 vector restored AP4B1 levels and ATG9A trafficking in SPG47 patient iPSC-derived neurons.**

(A) Patient-derived neurons (LoF/LoF) stained with ATG9A (yellow) and Golgin-97 (purple) show mislocalisation and accumulation of ATG9A at the TGN compared with healthy control neurons (LoF/WT). LV/hAP4B1 caused dispersal of ATG9A and a reduction at the TGN (white asterisks indicate rescued cells). Scale bar: 10 μm. (B) Patient-derived neurons treated with LV/hAP4B1 demonstrated a significant reduction in ATG9A ratio at the TGN at all MOIs (1, 5, 10, and 20). Data are presented as mean ± SD, $n = 2$ biological repeats. Data analysed by Mann–Whitney U test followed by Benjamini-Hochberg for multiple comparisons. (C) Cell viability is reduced at MOI 20. Data are presented as mean ± SD. $n = 3$ biological repeats. Data analysed by one-way ANOVA followed by post hoc Dunnett's multiple comparisons test with respect to control. ns = not significant. Source data are available online for this figure.

complex; this is inferred by the low or no expression of AP4E1 protein in western blot analysis (Fig. 3G,I). However, ICM delivery of AAV9/CBh-hAP4B1 led to a significant increase in AP4E1 protein, suggesting restoration of the AP-4 complex in CNS tissues: cerebrum (Fig. 3G,H; 25% increase) ($p = 0.0111$), spinal cord (Fig. 3I,J; 37% increase) ($p = 0.0084$) and cerebellum (Fig. 3K,L; 16% increase) ($p = 0.0207$). Overall, these data indicated that ICM delivery leads to a greater gene transfer to the CNS and efficient restoration of the AP-4 complex.

## ICM AAV9/hAP4B1 gene therapy in neonatal mice significantly improved motor function in Ap4b1-KO mice

*Ap4b1*-KO mice exhibit mild motor function deficits detected through hind limb clasping and rotarod performance (Scarrott et al, 2023). These phenotypes were used as a measure of the functional efficacy of AP4B1 gene therapy. In addition to testing the efficacy and expression

of our primary vector AAV9/CBh-hAP4B1, we set out to investigate a neuronal-specific vector to assess impact of restricting gene replacement to neuronal cells. For this reason, we introduced a hAP4B1 replacement driven by the neuronal-specific promoter synapsin (AAV9/SYN-hAP4B1). Two cohorts of mice were treated with the two therapeutic vectors: AAV9/CBh-hAP4B1 and AAV9/SYN-hAP4B1 where mice were dosed at P2-3 with either vector at a dose of $4 \times 10^{13}$ vg/kg. Cohort 1 was an all-female cohort ($n = 4$ per group) and were sacrificed at 2 months of age whereupon brains were extracted and processed for biochemical and anatomical analyses. Cohort 2 was an all-male cohort ($n = 8$ per group) which was observed over 9 months post treatment and underwent motor function assessment using a battery of behavioural tests including hind limb clasping, accelerated rotarod and open field.

The hind limb clasping assessment over 9 months demonstrated significant decline with age of the *Ap4b1*-KO mice (Fig. 4A). The hind limb clasping phenotype observed at 3, 6, and 9 months post treatment

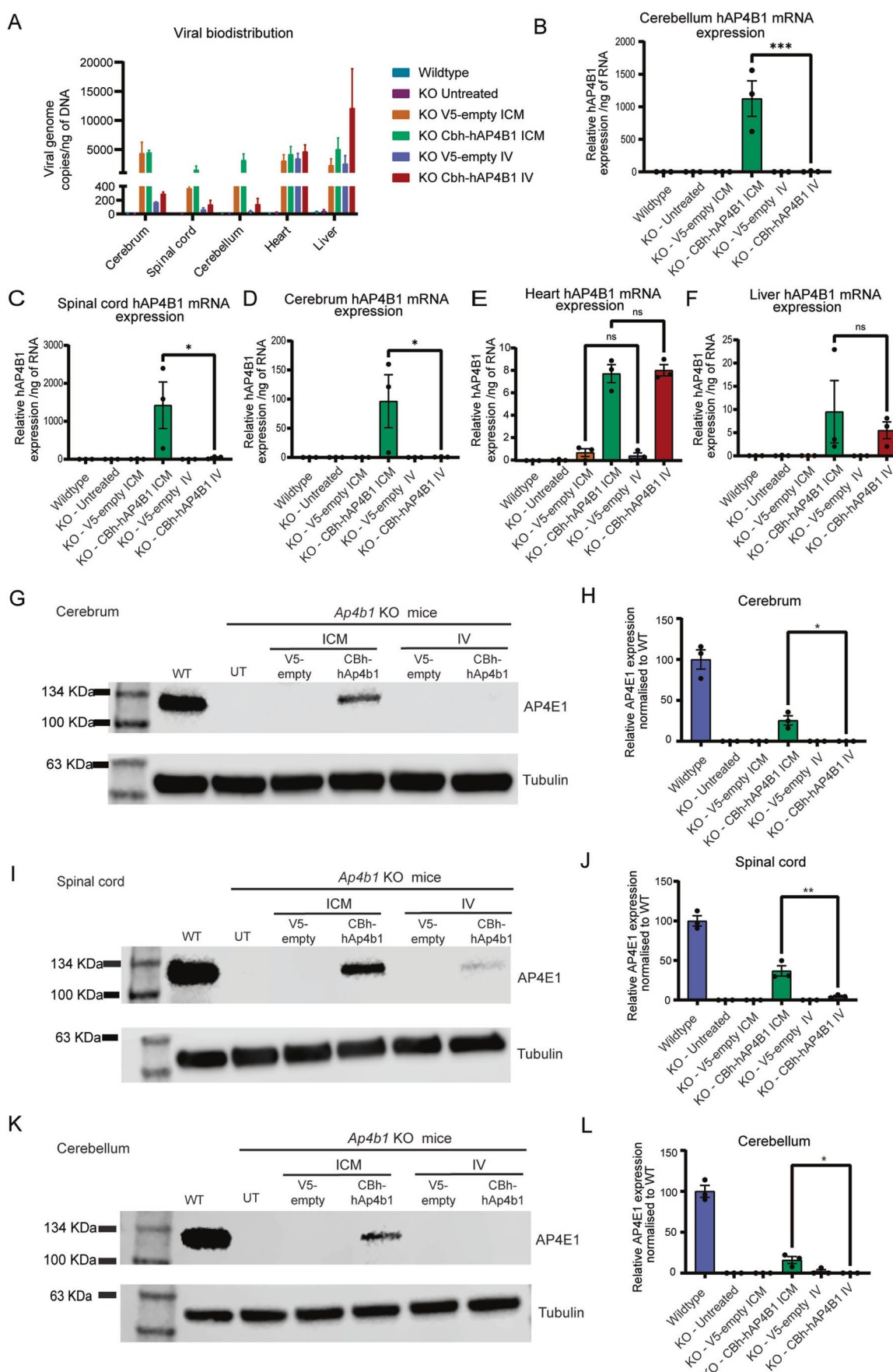

◄ **Figure 3. Intracisterna magna (ICM) delivery of AAV9/hAP4B1 shows greater transgene expression and the restoration of the AP-4 complex in the CNS compared with intravenous (IV) delivery in a *Ap4b1*-KO mouse model.**

(A) qPCR of total genomic DNA extracted from both peripheral and CNS tissues showing viral distribution of treated mice through ICM or IV delivery; cerebrum, spinal cord, cerebellum, heart, and liver. (B–F) RT-qPCR of total RNA extracted from CNS (cerebrum; cerebellum; spinal cord) and peripheral tissues (heart; liver) hAP4B1 mRNA expression in ICM delivery treated mice is elevated in the cerebrum, cerebellum, spinal cord compared to IV delivery. (G) Western blot detection of Ap4e1 showed a partial rescue of the Ap4e1 protein in the cerebrum ICM delivery, and no rescue with IV delivery. (H) Relative expression analysis displayed Ap4e1 was significantly increased in the cerebrum with ICM treatment. (I) Western blot detection of Ap4e1 showed a partial rescue of the Ap4e1 protein in the spinal cord ICM delivery, and no rescue with IV delivery. (J) Relative expression analysis displayed Ap4e1 was significantly increased in the spinal cord with ICM treatment. (K) Western blot detection of Ap4e1 showed a partial rescue of the Ap4e1 protein in the cerebellum ICM delivery, and no rescue with IV delivery. (L) Relative expression analysis revealed Ap4e1 was significantly increased in the cerebellum with ICM treatment. All data are presented as mean ± SEM, $n = 3$ mice. (B–F) were analysed via a one-way ANOVA with Tukey's multiple comparisons test. (H), (J) and (L) were analysed via unpaired t-test. Stars indicate $p \leq 0.05$ (*); $p \leq 0.01$ (**); $p \leq 0.001$ (***); ns = not significant. (B), $p = 0.0001$; (C), $p = 0.0206$; (D), $p = 0.0312$; (H), $p = 0.0111$; (J), $p = 0.0049$; (L), $p = 0.0207$. Source data are available online for this figure.

(MPT) (clasping scores $00.75 \pm 0.25$, $1.5 \pm 0.32$ and $2.00 \pm 0.43$ respectively) in AAV9/V5-empty control mice was improved with both AAV9/CBh-hAP4B1 and AAV9/SYN-hAP4B1 treatment. Clasping scores were significantly reduced to $0.54 \pm 0.20$ with AAV9/CBh-hAP4B1 ($p = 0.0272$) and $0.50 \pm 0.18$ with AAV9/SYN-hAP4B1 ($p = 0.0335$) treatment at 6 months post treatment (MPT). At 9 months when clasping severity worsens in untreated mice clasping scores are reduced to $0.72 \pm 0.23$ with the CBh vector and $0.87 \pm 0.22$ with the SYN vector at 9 MPT yet only CBh vector treatment reached significance ($p = 0.016$) (Fig. 4A). Our second motor function assessment, rotarod test, was carried out at 4 and 6 MPT. Rotarod testing at 4 MPT revealed no statistical differences between treatment groups, (Appendix Fig. S1). However, Rotarod assessment at 6 MPT revealed that AAV9/V5-empty control treated *Ap4b1*-KO mice had a significantly reduced latency to fall score ($204 \, s \pm 16$) compared to that of WT mice ($282 \, s \pm 8.4$) ($p = 0.0115$) at 6 MPT (Fig. 4B). Treatment with AAV9/CBh-hAP4B1 and AAV9/SYN-hAP4B1 vectors resulted in a rescue of this phenotype by bringing latency to fall period close to that of the WT mice (CBh: $274 \pm 11 \, s$, SYN: $252 \pm 22 \, s$, this data had no significant difference from the WT, $p = 0.9658$, $p = 0.5899$ respectively) (Fig. 4B).

Open field tests showed that the *Ap4b1*-KO mice exhibit a slight tendency to explore more than their WT counterparts, this difference, however, was not statistically significant (Appendix Fig. S2). Although open field was carried out in the mouse model characterisation study and showed a significant difference between *Ap4b1*-KO mice and WT mice, we observed contrasting results. The reason for this is most likely due to the difference in test modality between the open field assessments carried out. In our characterisation paper, open field was assessed through manually counting the number of grid line crossings a mouse crossed within a box over a 10-min period. Whereas in this study we used an automated open field analysis platform which tracked the movement of the mice over 5 min through a camera and recognition software which then gave a value of total distance travelled (Appendix Fig. S2).

## ICM AAV9/hAP4B1 gene therapy in neonatal mice led to significant improvement in biochemical and anatomical phenotypes in *Ap4b1*-KO mice

As presented in Scarrott et al, 2023, *Ap4b1*-KO mice display two anatomical brain defect phenotypes: a reduced corpus callosum thickness and enlarged lateral ventricles, and a biochemical phenotype: prominent perinuclear mislocalisation of ATG9A in

various brain regions. Assessment of corpus callosum thickness and lateral ventricle size at 2 months post neonatal-treatment revealed that both AAV9/CBh-hAP4B1 and AAV9/SYN-hAP4B1 therapeutic vectors led to a significant improvement of these phenotypes when compared with control group (AAV9/V5-empty) (Fig. 5A,B). Control V5-empty treated mice had on average an 0.8 fold reduction ($\pm 0.3$) in corpus callosum thickness compared to WT ($p < 0.0001$), while both vector treatments increased corpus callosum thickness, $0.93$ ($\pm 0.18$) fold change from WT level for CBh and $0.9$ ($\pm 0.23$) for SYN (Fig. 5A). Our analysis of the lateral ventricle size revealed that in *Ap4b1*-KO AAV9/V5-empty control mice average lateral ventricle measurements were significantly larger ($2.50 \pm 0.3$ fold) than WT mice ($p < 0.0001$), while hAP4B1 replacement led to rescue of this defect (Fig. 5B). Data sets for both therapeutic vectors showed no statistical difference from wild-type data indicating that AAV9-mediated hAP4B1 gene replacement led to efficient rescue of the two anatomical phenotypes ($p = 0.1839$, $p = 0.1709$ for CC, $p = 0.2657$, $p = 0.8368$ for LV, CBh and SYN respectively). At the 9-month timepoint, both therapeutic vectors restored Atg9a trafficking demonstrated by a reduction in Atg9a perinuclear accumulation in the cortex, hippocampus, brainstem and cerebellum (Fig. 5C).

Plasma and CSF collected from mice at 9 months underwent neurofilament light chain (NfL) assessment via an ultrasensitive S-Plex assay. NfL, a marker of neurodegeneration, has been reported to be released into the CSF and plasma in mouse models of neurodegenerative disease (Giacomucci et al, 2022) and in AP-4 patients (Alecu et al, 2023) but has not yet been investigated in AP-4 mouse models. NfL levels in the plasma were significantly elevated in the untreated ($40.6 \, ng/ml \pm 2.1$) and control ($36.5 \, ng/ml \pm 1.5$) *Ap4b1*-KO mice. Interestingly treatment with both therapeutic vectors led to complete rescue of this phenotype (Fig. 5D) ($p < 0.0001$ for CBh, $p < 0.0001$ for SYN) (note, WT and vector treated groups gave signals below the lower limit of detection of the assay). NfL levels in the CSF of control mice did not reach significant levels above the WT mice, however, the data followed a similar trend to the plasma data with a small treatment-dependent decrease in NfL after treatment with both vectors expressing AP4B1 (Fig. 5D). These data suggest that NfL from plasma can be used as a biomarker for treatment efficacy in human clinical trials in SPG47 patients.

Viral distribution assessments at 9 months post treatment with AAV9/CBh-hAP4B1 or AAV9/SYN-hAP4B1 displayed varying distribution. The CBh vector showed higher levels in the cerebrum compared to the SYN vector, equal copy numbers were observed in

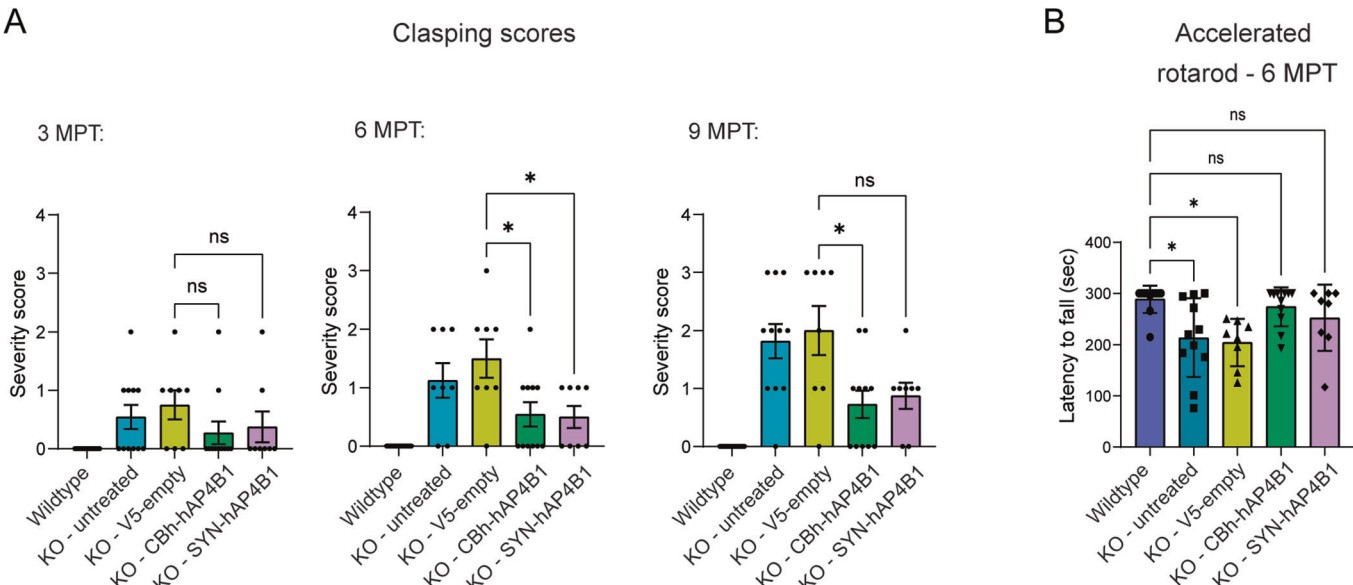

**Figure 4. Neonatal ICM treatment with AAV9/hAP4B1 in *Ap4b1*-KO mice showed significant motor function improvements over 9 months post treatment (MPT).**

(A) Hind-limb clasping assessment showed an increase in severity from 6 months to 9 months of age. At 6 months treatment with both CBh-hAP4B1 and SYN-hAP4B1 show a significant reduction in clasping severity (CBh, $p = 0.0272$, SYN $p = 0.335$). While at 9 months clasping scores are reduced with both vectors only CBh-hAP4B1 vector reaches significance CBh ($p = 0.0165$, SYN $p = 0.0711$). (B) Accelerated rotarod performed at ~p180 showed that untreated and control treated KO animals had significantly reduced latency to fall performance compared to WT animals. KO-UT $p = 0.0150$; KO-V5 $p = 0.0115$. CBh-hAP4B1 and SYN-hAP4B1 treated animals had improved latency to fall scores with no significant difference from the WT animals. $n \geq 8$ per group. All data groups are males. Data are presented as mean ± SEM and are analysed by the Kruskal–Wallis test with Dunn's multiple comparison test. Stars indicate *$p < 0.05$; **$p \leq 0.01$. MPT: month post treatment. Source data are available online for this figure.

the cerebellum and higher copy numbers of the SYN vector were displayed in the brainstem (Fig. 5E). Both vectors showed low viral distribution within the liver. The viral cDNA transgene that is transcribed to mRNA in these tissues followed a similar pattern but CBh vector resulted in a more consistent expression throughout the brain regions and spinal cord (Fig. 5F).

Lastly, we examined the presence of calbindin spheroids within the deep cerebellar nuclei (DCN). Scarrott et al previously demonstrated the presence of calbindin-positive spheroids in the DCN of *Ap4b1*-KO mice (Scarrott et al, 2023). Our data corroborated these findings, showing a significant presence of spheroids in untreated (61 ± 0.5) and AAV9-V5-treated knockout mice (54 ± 4) compared to WT mice (0 ± 0) ($p < 0.0001$ for UT and V5) (Fig. EV2). However, treatment with both AAV9/CBh-hAP4B1 and AAV9/SYN-hAP4B1 vectors significantly reduced the presence of spheroids to almost WT levels (3.6 ± 1.8 for CBh; $p < 0.0001$ and 4.7 ± 2.6 for SYN; $p < 0.0001$) (Fig. EV2).

Overall, these data demonstrate a significant recovery in the trafficking of ATG9A, an improvement of anatomical abnormalities and a significant rescue of the motor function deficits usually observed in *Ap4b1*-KO mice, suggesting a restoration of AP-4 function. These results are strongly indicative of a positive trajectory following treatment of AAV9/hAP4B1 in *Ap4b1*-KO mice, when treated at neonatal age. Phenotypic improvements observed with AAV9/CBh-hAP4B1 vector treatment were slightly superior to that observed with the AAV9/SYN-hAP4B1 vector. In addition, a more widespread mRNA expression across the CNS was observed with the AAV9/CBh-hAP4B1 vector. Based on this

finding we decided to take forward the AAV9/CBh-hAP4B1 vector for dose-response studies in *Ap4b1*-KO adult mice.

## ICM AAV9/CBh-hAP4B1 gene therapy led to a dose-dependent improvement in Atg9a trafficking, anatomical abnormalities and motor function deficits in adult *Ap4b1*-KO mice

As neonatal animals are more susceptible to brain plasticity and recovery from AAV-initiated inflammatory responses (Gray, 2016), we chose to perform a dose-response study in adult mice to consolidate the efficacy of this therapeutic strategy. In addition, this dose-response study allowed us to estimate a more relevant clinical dose for treating patients. We therefore treated adult mice (~P60) with the AAV9/CBh-hAP4B1 vector at 3 doses: (i) $5 \times 10^{12}$ vg/kg ($n = 12$), (ii) $3 \times 10^{12}$ vg/kg ($n = 12$), and (iii) $2 \times 10^{12}$ vg/kg ($n = 12$).

To determine whether the treatment doses could restore Atg9a trafficking within the brains of adult mice, we performed immunohistochemistry analysis for Atg9a staining in the cortex, hippocampus, cerebellum and brainstem at 4 months post treatment. As reported in the neonatal study, Atg9a accumulated in the perinuclear region of neurons in all brain regions of *Ap4b1*-KO mice (Fig. 6A). In mice treated with a dose escalation of the AAV9/CBh-hAP4B1 therapeutic vector, we observed a dose-dependent reduction in Atg9a perinuclear accumulation across all brain regions (Fig. 6A). Higher magnification images revealed a redistribution of Atg9a from the perinuclear region throughout the cytoplasm which improved with increasing dose (Fig. 6A inset).

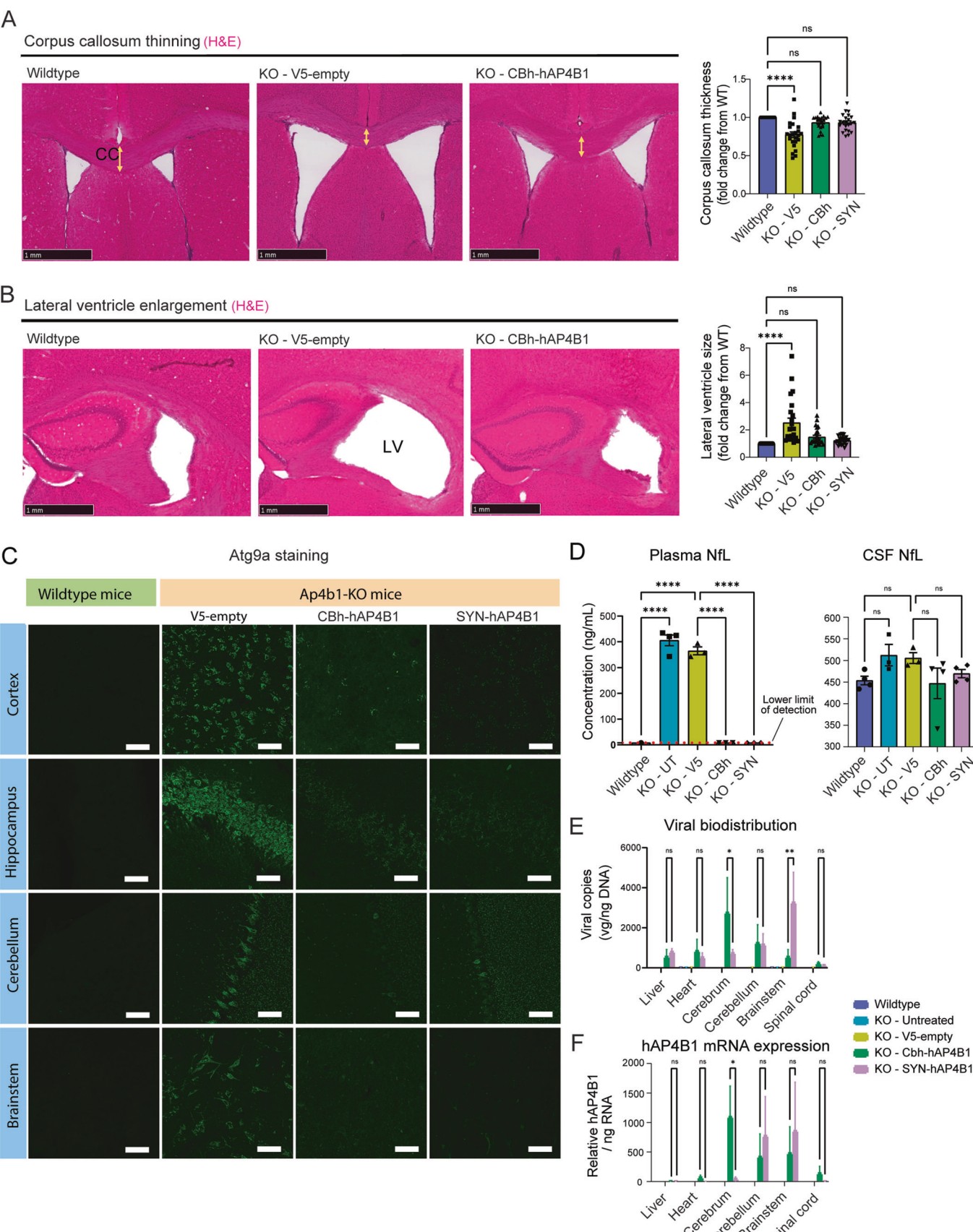

◄ **Figure 5.  Neonatal ICM treatment with AAV9/hAP4B1 in *Ap4b1*-KO mice showed significant improvement of anatomical and biomolecular phenotypes.**

(A, B) representative haematoxylin and eosin (H&E) stained coronal brain sections revealing corpus callosum (CC) thinning (A) and lateral ventricle (LV) enlargement (B) in control treated *Ap4b1*-KO mice (AAV9/V5-empty) compared with wild-type mice. CBh-hAP4B1 and SYN-hAP4B1 vectors significantly increased corpus callosum thickness ($p < 0.0001$) and reduced lateral ventricle size ($p = 0.0001$) (2 MPT). (C) Representative Atg9a stained micrographs displayed Atg9a accumulation and upregulation at 9 MPT; both CBh and SYN vectors reduced Atg9a expression and perinuclear accumulation in the cerebellum, brainstem cortex and hippocampus. Scale bar 50 μm. (D) Plasma neurofilament light chain (pNfL) S-PLEX assay demonstrated an elevated level of pNfL in *Ap4b1*-KO mice at 9 months old compared to WT mice ($p < 0.0001$). CBh and SYN both reduced pNfL levels to WT levels ($p < 0.0001$). NfL S-PLEX assay to analyse levels in the CSF did not show any significant increase in diseased mice. (E) qPCR of total genomic DNA shows good viral biodistribution in areas of the brain with lower expression in peripheral tissues and the spinal cord. CBh-hAP4B1 showed an increased distribution in the cerebrum compared to SYN-hAP4B1 ($p = 0.0360$). Whereas SYN-hAP4B1 showed an increased distribution in the brainstem ($p = 0.0016$). (F) RT-qPCR of total RNA extracted demonstrated that hAP4B1 mRNA was expressed throughout the CNS, with low levels in peripheral tissues with both vectors. CBh vector gave more consistent expression throughout CNS. With significantly higher expression within the cerebrum ($p = 0.0246$). (A, B) $n \geq 3$ per group, 8 measurements for CC (A) or 10 for LV (B) were plotted per mouse. (C), $n = 4$ per group. (D), $n \geq 3$ per group. (E, F) $n \geq 4$ per group. All data groups are males. Data are presented as mean ± SEM. and are analysed by a one-way ANOVA with Tukey's multiple comparison test. Stars indicate $*p < 0.05$; $**p \leq 0.01$, $****p \leq 0.0001$. Source data are available online for this figure.

Quantification of the perinuclear localisation of Atg9a revealed significant accumulation in the cortex ($24.8 \pm 2.7$ fold change from WT level), hippocampus ($29.9 \pm 4.4$ fold change), brainstem ($24.1 \pm 3.9$ fold change) and cerebellum ($30.8 \pm 4.9$ fold change) of V5-empty treated mice ($p < 0.0001$ for all regions). The AAV9/hAP4B1 high-dose treated group displayed the greatest improvement in Atg9a distribution (Fig. 6B). While low-dose treatment did not significantly reduce Atg9a accumulation in any brain region, mid-dose treatment showed its greatest reduction in the cerebellum ($12.5 \pm 3.1$ fold change; $p = 0.0006$), yet also a significant reduction in the hippocampus ($p = 0.0055$) and brainstem ($p = 0.0349$). High-dose treatment displayed significant reduction in all brain regions with the largest rescue effect on the cerebellum ($8.5 \pm 2.1$ fold change; $p < 0.0001$), hippocampus ($9.6 \pm 0.7$ fold change; $p < 0.0001$) and brainstem ($5.3 \pm 0.7$ fold change; $p = 0.0002$). The Atg9a perinuclear localisation in the cortex was the least affected by all treatments. See Fig. EV3 for Atg9a staining with Hoechst counterstain. These findings can be explained by hAP4B1 mRNA biodistribution carried out at 2 MPT where brainstem and cerebellum showed the highest transgene mRNA expression for high-dose mice (Fig. 6C).

At 4 months post treatment, we assessed motor function via accelerated rotarod performance and hind limb clasping. Similar to previous studies, KO-UT and KO-V5 treated mice demonstrated a significant reduction in rotarod performance ($247 \pm 18$ s for KO-UT ($p = 0.0013$), and $239 \pm 22$ s for KO-V5 ($p = 0.0006$) compared to WT mice ($369 \pm 27$ s). In animals treated with the therapeutic vectors, we observed a dose-dependent rescue in this motor performance, with mid- and high- doses showing a near complete rescue in latency to fall. Mid-dose ($325 \pm 18$ s) and high-dose ($329 \pm 16$ s) mice had significantly higher latency to fall scores than control KO-V5 mice ($p = 0.05$ for mid-dose, $p = 0.035$ for high-dose) (Fig. 6D). Hind limb clasping was also evaluated at 4 MPT in control KO mice. Treatment with all three doses reduced clasping severity scores, yet, statistical significance was only achieved for the high-dose group (Fig. 6E) (high dose: 0.5 severity score ($\pm 0.2$); V5-empty 2.2 severity score ($\pm 0.8$), $p = 0.0171$).

Assessment of corpus callosum thickness at 4 months post treatment demonstrated a significant reduction in corpus callosum thickness in untreated and V5 control mice (Fig. 6F). In contrast to treatment of neonates, treatment in adults did not show any significant improvement. This finding is likely to be due to the late stage of intervention. Lateral ventricle assessment in this study also

showed no significant differences between groups, although a trend of increased ventricle sizes in untreated animals was observed (Fig. 6G). Corpus callosum thickness and lateral ventricle findings support the notion to apply treatment at an early stage after diagnosis to increase the chance of therapeutic efficacy.

Lastly, we examined whether calbindin-positive spheroids could be reduced when treatment is applied in adult mice. Similar to the findings reported following post-neonatal treatment (Fig. EV2), both untreated and AAV9/V5 control-treated KO mice at 6 months of age displayed a high number of calbindin-positive spheroids within a defined area of the DCN (Fig. EV4) ($p < 0.0001$). Dose escalation with the therapeutic vector revealed a dose-dependent decrease in spheroid presence (Fig. EV4). Both mid-dose and high-dose treatments significantly reduced the number of calbindin-positive spheroids ($p = 0.0066$ for mid-dose, $p = 0.0004$ for high-dose), with the high-dose treatment achieving a 59% reduction (Fig. EV4).

## AAV9/CBh-hAP4B1 potency assay reveals a dose dependent-increase on the translocation of ATG9A from the TGN to the cell periphery confirming restoration of ATG9A trafficking

To support the clinical development of this gene replacement vector, we next developed an in vitro potency assay. The use of *AP4B1*-KO SH-SY5Y cells (Davies et al, 2022) in combination with an automated high-throughput imaging platform (Saffari et al, 2024), allowed us to standardise and perform dose-response experiments using the AAV9/CBh-hAP4B1 vector at MOIs between $1 \times 10^3$ and $1.6 \times 10^7$ vg/cell (Fig. 7). This revealed a dose-dependent rescue of ATG9A distribution to *AP4B1*-WT SH-SY5Y levels. An MOI of $1.6 \times 10^7$ vg resulted in a 68.36% ($\pm 12.88\%$ (SD)) rescue of ATG9A translocation ($p < 0.0001$) (Fig. 7A). Higher MOIs between $4 \times 10^6$ and $1.6 \times 10^7$ resulted in a lower cell count, indicating a potential impact on cell viability or proliferation (Fig. 7B). The translocation of ATG9A from the TGN was supported through immunofluorescent micrographs of treated neurons (Fig. 7C), neurons treated with $1.6 \times 10^7$ vg exhibited an ATG9A expression similar to WT neurons. Multi-parametric profiling of the TGN (Saffari et al, 2024) showed only minor changes to TGN morphology, as an indicator for cell toxicity, for MOIs of less than $4 \times 10^6$ vg (Fig. 7D). Collectively, these data establish a reference range for dose-response experiments and suggest an MOI of $2 \times 10^6$ vg as a safe and efficient dose for restoration of ATG9A translocation in cells. At

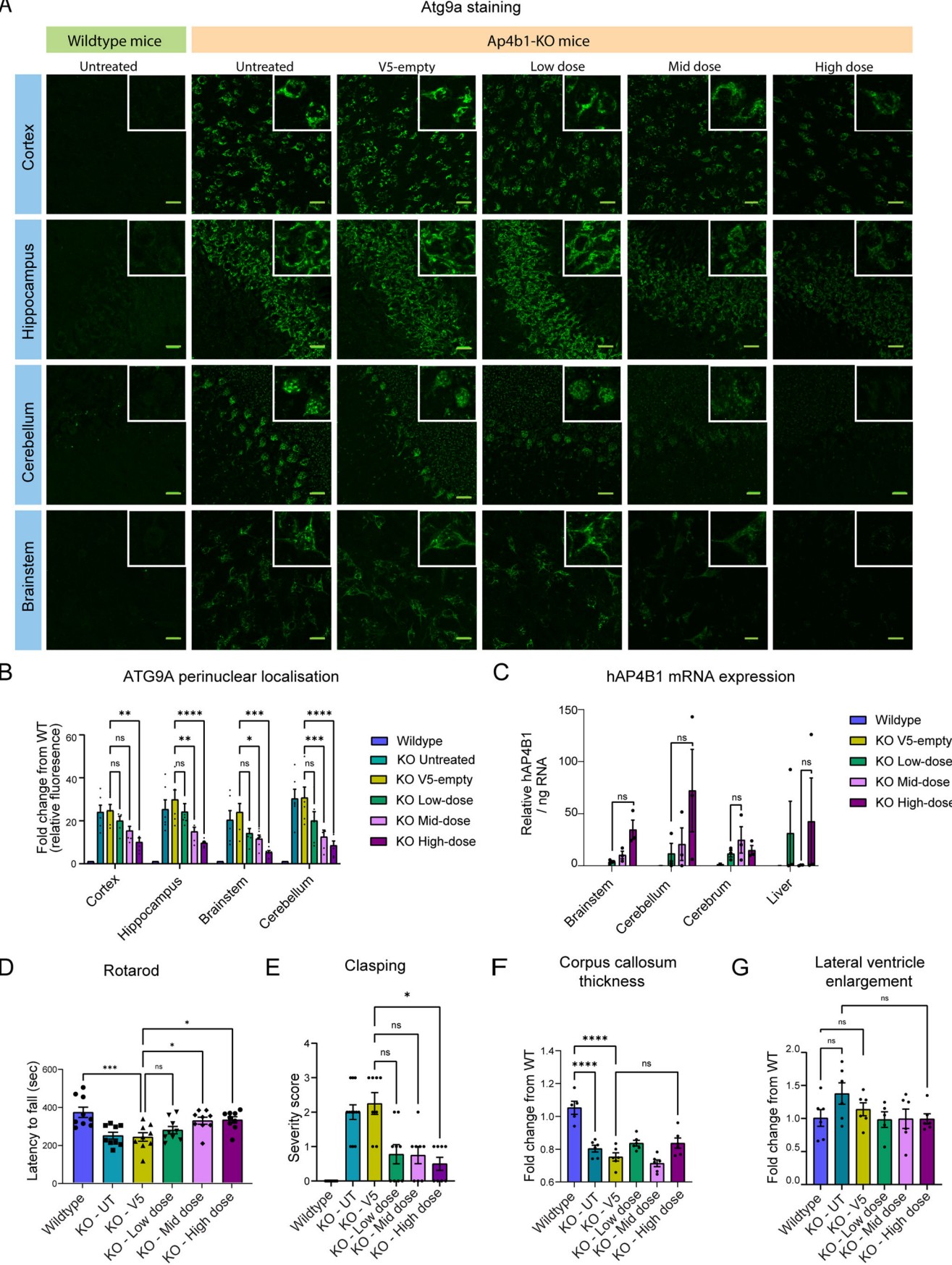

**Figure 6. Dose-dependent improvements 4 months following treatment with AAV9/hAP4B1 in adult (P60) *Ap4b1*-KO mice.**

(A) Representative Atg9a stained micrographs showing a dose-dependent reduction of Atg9a perinuclear accumulation in the brain regions: cortex, hippocampus, cerebellum, brainstem (inset higher magnification shows Atg9a dispersal with increasing dose). See Fig. EV3 for Hoechst images. Scale bar 20 μm. (B) Shows a dose- and location-dependent reduction of ATG9A perinuclear localisation. Only high-dose vector significantly reduced Atg9a expression in all four brain regions (cortex ($p = 0.0064$), hippocampus ($p < 0.0001$), brainstem ($p = 0.0002$) and cerebellum ($p < 0.0001$)). While mid-dose significantly reduced atg9a expression in the hippocampus ($p = 0.0055$), brainstem ($p = 0.0349$) and cerebellum ($p = 0.0006$)). ($n = 6$ mice). (C) RT-qPCR of total RNA extracted at 2 MPT displayed a dose-dependent hAP4B1 mRNA expression in brain regions: brainstem, cerebellum and cortex ($n = 3$ mice). No significance between groups. (D) Rotarod assessment 4 months following mid- ($p = 0.0500$) and high-dose ($p = 0.0350$) treatment (~p180) displayed significant improvement of the latency fall compared to that of untreated/control treated *Ap4b1*-KO mice ($p = 0.0006$). (E) All treatment doses show reduction in hind-limb clasping severity score at 4 months following treatment however only high dose reaches significance ($p = 0.0300$). (F) Corpus callosum thinning data demonstrated a significant reduction in KO-UT ($p < 0.0001$) and KO-V5 ($p < 0.0001$) treated mice with no clear treatment effect. (G) Lateral ventricle enlargement data did not reach significance although there was a trend reduction in LV size with treatment. Data are presented as mean ± SEM, $n = 6$ per group (3 males, 3 females) for (B), (F) and (G); $n = 3$ per group for (C) (3 females); $n = 9$ per group for (6 males, 3 females) for (D) and (E). Data were analysed in by a one-way ANOVA with Tukey's multiple comparison test except for (B) which is two-way ANOVA with Tukey's multiple comparisons and (E) was analysed by the Kruskal–Wallis tests with Dunn's multiple comparisons. Stars indicate $p < 0.05$ (*); $p \leq 0.01$ (**), $p \leq 0.001$ (***), $p \leq 0.0001$ (****). Source data are available online for this figure.

this MOI, there was a 33.66% (±4.30%) rescue of ATG9A translocation after 24 h, with a minor change in cell count ($Z = -0.93 \pm 1.36$ (SD)) and no significant changes in TGN morphology. These data and this assay will enable comparisons of different lots of viral vectors in the future.

## AAV9/AP4B1 gene therapy was safe and well tolerated in WT mice in non-GLP safety studies

To elucidate the safety profile of our treatment two separate safety studies were carried out in wild-type mice. We tested the safety of two of our therapeutic vectors (AAV9/CBh-hAP4B1 and AAV9/SYN-hAP4B1), along with a vector encoding the mouse *Ap4b1* gene (AAV9/CBh-mAP4B1) to investigate immune priming and control vector (AAV9/V5-empty). The vector encoding the mouse *Ap4b1* gene was used as a control to be able to detect a possible human-specific transgene-related response. Safety studies in WT mice included study 1: ELISpot reactivity and study 2: long-term toxicology study with histopathological assessment.

In safety study 1 mice were administered AAV9/AP4B1 at $4 \times 10^{13}$ vg/kg via ICM delivery and sacrificed at 3 months post-treatment. To evaluate the immune responses, splenocytes from treated mice were collected and analysed via ELISpot assay. This assay detects INF-γ secretion from activated T cells in response to a stimulant. We assessed if exposure to hAP4B1 peptides would cause an immune response in splenocytes from mice treated with our vectors. We found minimal spot forming units (SPU) (which indicate INF-γ release) in response to the hAP4B1 peptides, across all therapeutic groups (Fig. EV5B); exposure to the hAP4B1 peptides was not significantly different from exposure to the negative control. This highlights that the hAP4B1 protein does not initiate T cell immune response and that the splenocytes were not activated by the vector treatment. hAP4B1 mRNA expression analysis confirmed significant AAV9-mediated transgene expression in the brain with low expression in the liver (Fig. EV5C).

In safety study 2 mice received AAV9/AP4B1 at $1.5 \times 10^{14}$ vg/kg via ICM delivery at P42 and were monitored over a 1-year period. Mice were sacrificed at 28 and 365 days post treatment for histopathological assessment of liver, heart, brain and spinal cord (Fig. 8).

Mice assessed in the long-term safety study were monitored daily for mortality, body weight, and clinical/behavioural changes over 365 days post treatment. Mice were assessed weekly and

classified as bright, alert, responsive and healthy based on standardised criteria. No abnormalities were observed except for hair loss, commencing at 9 weeks post injection, which was attributed to hetero-barbering. This was observed in all cohorts (including untreated controls) with the exception of female animals treated intravenously with AAV9/SYN-hAP4B1. For some mice this progressed to ulcerative dermatitis and was treated with antibiotic cream. In three cases this was unsuccessful and the condition necessitated humane sacrifice. There was one unexplained death at 80 days post-treatment in cohort 3, which was a male mouse who received ICM injection of AAV9/CBh-hAP4B1. This mouse was humanely sacrificed for lethargy. Male mice showed no significant weight difference between any pair of cohorts ($p > 0.2$) (Appendix Table S1). Female mice from cohort 2 (ICM AAV9/V5) were significantly heavier than other cohorts ($p < 0.001$) but neither of the mice cohorts receiving the therapeutic vector had any weight differences from the untreated control group (see Appendix Table S1 and Appendix Fig. S3). At the 28-day and 1-year sacrifice, complete blood counts and serum chemistry were obtained (Fig. 8A). Elevation in white blood cell count (WBC) was observed in the ICM cohort treated with AAV9/CBh-hAP4B1. This was driven by an increase in lymphocytes; however, by 1 year post treatment these elevations had diminished. Histopathological assessment for liver inflammation, necrosis, steatosis and fibrosis via H&E staining at 28 days and 365 days post treatment revealed no significant differences between groups (Fig. 8B). Brain, spinal cord and heart histopathology stained with H&E and CD68 also showed no significant adverse events between cohorts at 28 or 365 days post treatment (Appendix Table S1).

At 28 days a subset of mice were sacrificed for the assessment of transgene expression in the relevant tissues (liver, heart, spinal, cerebellum and cortex). The data show that ICM injections achieve AP4B1 expression in the cerebellum and spinal cord and to a lesser extent in the cortex. The CBh promoter consistently gave a higher level of expression compared to the synapsin promoter (Fig. 8C). In this set of experiments there was considerable expression in the heart and liver likely from CSF leakage post injection.

## AAV9/AP4B1 gene therapy was safe and well tolerated in a non-human primate (NHP) GLP toxicology study

To further investigate the safety and biodistribution pattern of AAV9/CBh-hAP4B1 in a larger animal model, a GLP toxicology

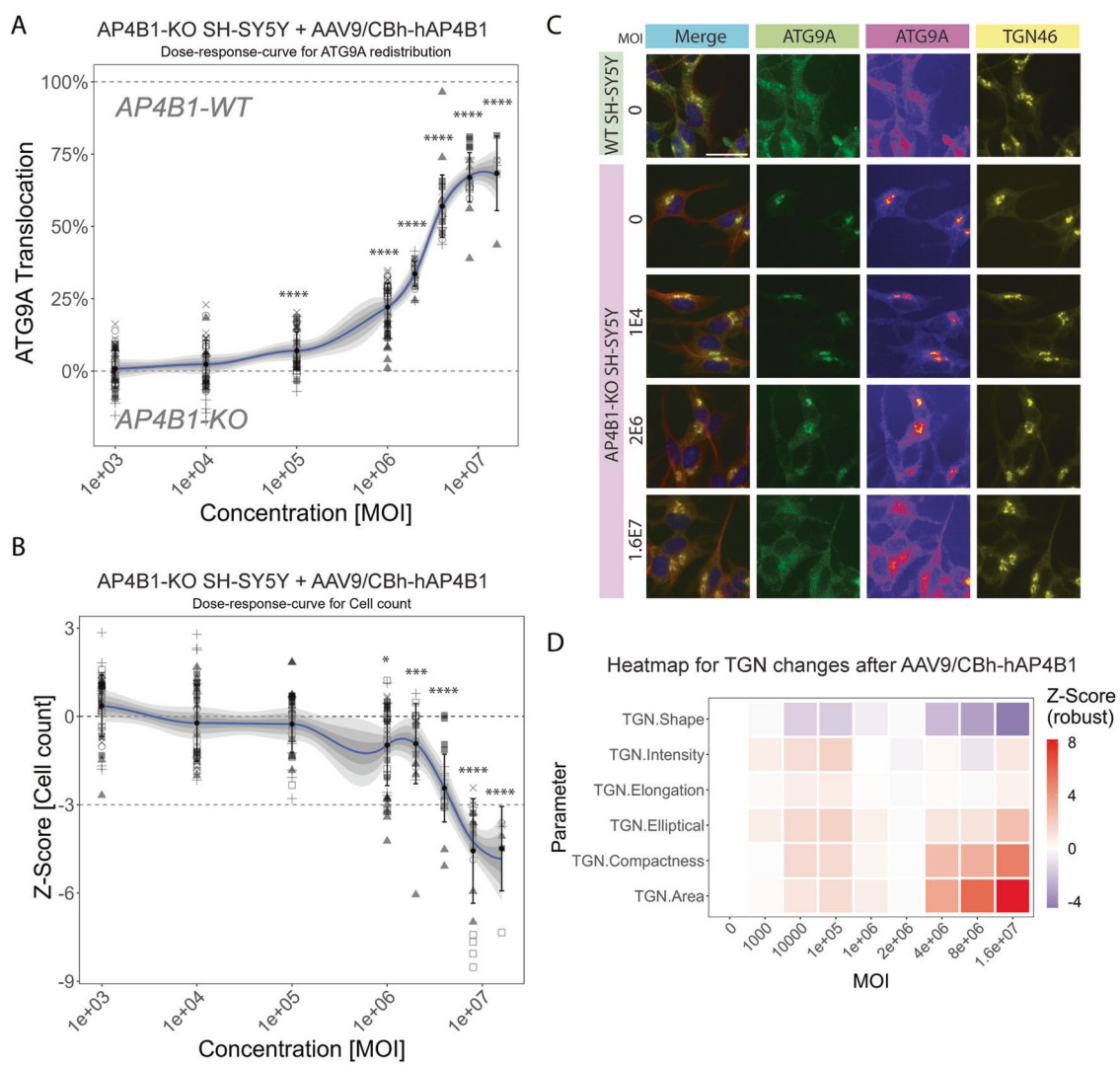

**Figure 7. AAV9/AP4B1 potency assay in *AP4B1*-KO SH-SY5Y confirms restoration of ATG9A trafficking.**

(A) Potency assay of AAV9/AP4B1 in AP4B1-KO SH-SY5Y cells showing ATG9A translocation, calculated on the percent rescue of the ATG9A distribution in the experimental cells from *AP4B1*-KO SH-SY5Y (0% Translocation) back to *AP4B1*-WT SH-SY5Y ATG9A distribution (100% Translocation) depending on transduction with different multiplicity of infection (MOI). Cells were treated 24 h after plating for 72 h. Six individual replicates were analysed each with up to 8 wells. On average 101.800 individual cells were analysed per condition. Shaded areas represent ±1 SD, ±2 SD and ±3 SD. All data points represent per-well means. *p* < 0.0001 for MOI 1E5 – 16E6. (B) Cell counts, presented as Z-Score relative to untreated AP4B1-KO SH-SY5Y cells (dark-grey dotted line at 0), offer insights into the absence of cell toxicity. A Z-Score larger than −3 is defined as an indication of non-toxicity (light-grey dotted line). *p* = 0.0237 for MOI 1E6, *p* = 0.0002 for MOI 2E6, *p* < 0.0001 for MOI 4E6 – 16E6. (C) Representative images from (A) of SH-SY5Y cells treated with different multiplicity of infections (MOI) for 72 h. The merge images show β-3 tubulin (red), Hoechst (blue), the Trans-Golgi-Network (TGN, yellow) and ATG9A (green). Separate channels for TGN and ATG9A, along with fluorescence intensity representations using a colour lookup table for ATG9A, enhance the visualisation and ATG9A trafficking. The scale is set at 20 μm. (D) Multi-parametric profiling assesses TGN changes, considering TGN intensity and descriptors of TGN shape and network complexity. Heatmap visualisation summarises these measurements across different MOIs, normalised to untreated *AP4B1*-KO SH-SY5Y cells. *N* = 6, data presented as mean ± SD. Data were analysed via one-way ANOVA with Turkey's multiple comparisons test. Statistics are compared to AP4B1 KO UT. Stars indicate *p* ≤ 0.001 (***), *p* ≤ 0.0001. Source data are available online for this figure.

study was carried out in cynomolgus monkeys. The design of this study was discussed and guided by the FDA as part of a pre-IND meeting. Animals were divided into three groups; vehicle, low-dose (3.2 × 10$^{12}$ vg/kg) and high-dose (1.7 × 10$^{13}$ vg/kg). Age matched animals were dosed by a single ICM administration and monitored over 1 to 4 months post treatment. Immediately before injection a 1 mL sample of CSF was withdrawn from each animal to avoid elevating intracranial pressure. Blood samples were taken periodically for assessment; there was a mild elevation of alanine

aminotransferase (ALT) (a marker of liver inflammation) in the AAV9/hAP4B1 treated animals at 8 days post treatment (DPT), however, this returned back to baseline levels by 22 DPT (see Appendix Table S2). At a 1-month interim, a subset of animals from vehicle control (1M and 1F) and high dose (2M and 1F) groups were sacrificed. At 4 months the remaining animals in each group were sacrificed; vehicle (1M and 1F), low dose (1M and 2F) and high dose (1M and 2F). Tissue samples were prepared for biodistribution studies or histological examination. Analysis of

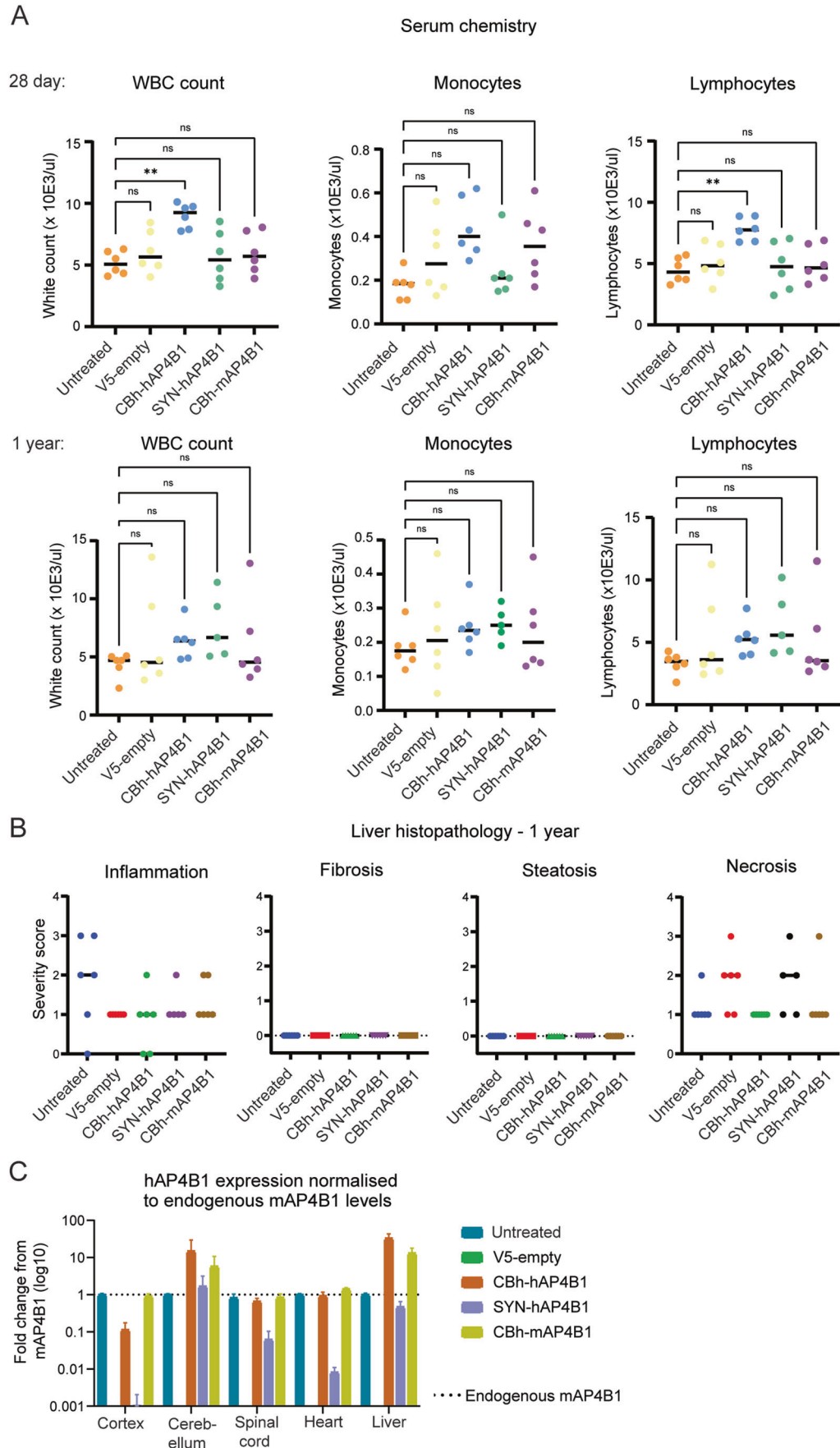

◄ **Figure 8. Long term in vivo safety study in WT mice at 28 days or 365 days following treatment at P42 highlighted no adverse events or histopathology were present.**

Various vector promoter sequences were tested including the our treatment vector (CBh-hAP4B1). (A) Serum chemistry, including: white blood cell, lymphocyte and monocyte counts were analysed at 28 days post treatment (DPT) and 1 year post treatment. (B) Histopathological assessments of WT mouse livers at 1 year following treatment with hAP4B1 vectors showed no adverse effects on inflammation, necrosis, steatosis or fibrosis compared to untreated the cohort. No adverse effects were observed in heart, brain and spinal cord (see Appendix Table S1 for data set). (C) RT-qPCR of total RNA extracted from mice at 28 DPT demonstrated that the CBh promoter gave consistently higher levels of hAP4B1 mRNA expression across the CNS compared to the SYN promoter. Data shows high hAP4B1 mRNA expression within the liver and heart, which indicates CSF leakage in this study. Data are presented as mean ± SEM, 28-day sacrifice $n = 6$ per group (3 males and 3 females); 1-year sacrifice $n = 6$ per group (3 males and 3 females) and were analysed via a one-way ANOVA with Tukey's multiple comparisons test. Stars indicate $p \leq 0.01$ (**), ns = not significant, WBC: $p = 0.0014$, Lymphocyte: $p = 0.0028$. Source data are available online for this figure.

transgene biodistribution and expression revealed that hAP4B1 mRNA was enhanced in high-dosed animals sacrificed at 4 months following injection, particularly in the cerebellum, spinal cord, heart and the haemo-lymphoid system (Fig. 9A). Analysis of viral distribution in these animals at 16 weeks post treatment demonstrated a high expression throughout the brain, spinal regions and the dorsal root ganglions (DRGs) with high vector dosing. The same vector dose resulted in low levels of expression in most peripheral tissues, but considerable viral expression in the liver (Fig. 9B).

Histological examination of tissues revealed that treatment induced microscopic changes in the dorsal root ganglia, spinal cord, sciatic nerve and thymus. Because of the common occurrence of immune cell infiltration in dorsal root ganglia with intrathecal administration of AAV9, a detailed histopathological assessment and nerve conduction studies were performed. At terminal sacrifice (4 months post treatment) there was minimal neuronal cell body degeneration in the DRG and minimal-to-mild mononuclear cell infiltration in both sexes at $1.7 \times 10^{13}$ vg/kg. Minimal-to-mild axonal degeneration was observed in the spinal cord in the male at $1.7 \times 10^{13}$ vg/kg and females at $3.2 \times 10^{12}$ vg/kg. The sciatic nerve also presented minimal axonal degeneration in males and females at both doses. These treatment-related microscopic findings displayed a dose and time dependence in the severity and incidence of histopathological findings. However, severity scores did not exceed 2 (mild) out of 5 (severe) at any point (Appendix Table S2). Similar findings have been previously reported in the literature as AAV9 induced pathologies in NHPs (Chen et al, 2023; Hordeaux et al, 2020; Hordeaux et al, 2018; Hinderer et al, 2018), and are therefore not unique to our gene therapy vector. There were no mortalities or major clinical observations over the duration of the study, including no significant change to body weight. Although microscopic changes were observed in the DRG and spinal cord, nerve conduction studies were all normal (Fig. 9C), indicating no serious nerve damage or dysfunction with either dose of therapeutic vector.

In conclusion, ICM administered AAV9/CBh-hAP4B1 by a single injection in age-matched cynomolgus monkeys was well tolerated at both doses with minor and well-established immune responses to AAV9. Therefore, both doses ($1.7 \times 10^{13}$ and $3.2 \times 10^{12}$ vg/kg) are considered safe.

## Discussion

The development of novel effective therapeutic interventions for rare neurological disorders presents a major scientific and clinical challenge (Tambuyzer et al, 2020). Given the implications of neurological diseases on the quality of life of patients, and burden on caregivers and healthcare systems, there is a significant unmet need for safe and efficacious therapies (Yang et al, 2022; Sandilands et al, 2022; Saffari et al, 2024). Gene replacement, in particular AAV-mediated gene delivery, is a popular platform for investigation for rare diseases (Tambuyzer et al, 2020). Monogenic recessive diseases are uniquely positioned to be targeted by gene replacement of the single defective gene. As previously mentioned, pathogenic variants in any of the four genes that encode the subunits of the AP-4 complex lead to AP-4 deficiency (Ebrahimi-Fakhari et al, 2021a). This makes AP-4-HSP an ideal candidate for gene replacement therapy. In 2023 an AAV9/AP4M1 gene replacement strategy was utilised by Chen et al, for SPG50 (caused by the pathogenic variants in the AP4M1 subunit), who reported the restorative effects of an AAV9/AP4M1 vector administered through the intrathecal (IT) delivery route. Their gene therapy is now in the initial stages of clinical trials treating young children with SPG50 (Chen et al, 2023). This result is promising with regards to the prospect of treating SPG47 patients with a similar gene replacement strategy. Our studies have established an effective gene therapy for SPG47. We initially showed phenotypic rescue from in vitro cell lines of disease including patient iPSC-neurons. This phenotypic rescue was validated in neonatal and adult in vivo studies on *Ap4b1*-KO mice where ICM delivery of our therapeutic vector AAV9/CBh-hAP4B1 demonstrated efficacious outcomes. Moreover, long-term WT mouse safety studies and GLP toxicology studies in NHPs gave evidence for no significant adverse treatment-related events.

In our proof-of-concept studies, we generated data from four separate cell models of SPG47 disease. Both HeLa and SH-SY5Y *AP4B1*-KO in vitro studies strongly indicated that delivery of the *AP4B1* gene to deficient cells could rescue the hallmark phenotypes of these in vitro disease models, including AP-4 complex formation (through AP4E1 upregulation) and subsequent improvement of ATG9A trafficking away from the TGN. Dose escalation studies of LV/hAP4B1 applied to patient iPSC neurons demonstrated a dose-dependent rescue of ATG9A trafficking from MOI 1 to 10, with MOI 20 starting to reduce cell survival. Since lentiviral vectors are known to cause this effect at higher MOI, and toxicity is observed in the control group (LV/GFP) as well, it is unlikely to be a result of transgene overexpression. Dose escalation in patient fibroblasts also demonstrated a dose-dependent increase of the AP-4 complex and in turn a reduction in ATG9A expression to WT levels for MOI 10 and 20. MOI 20 did not show cell toxicity in fibroblasts, indicating cell type specific differences (Fig. EV1). These experiments show that *AP4B1* gene replacement can rescue the hallmark disease phenotype in the various cell line models of AP4B1 deficiency.

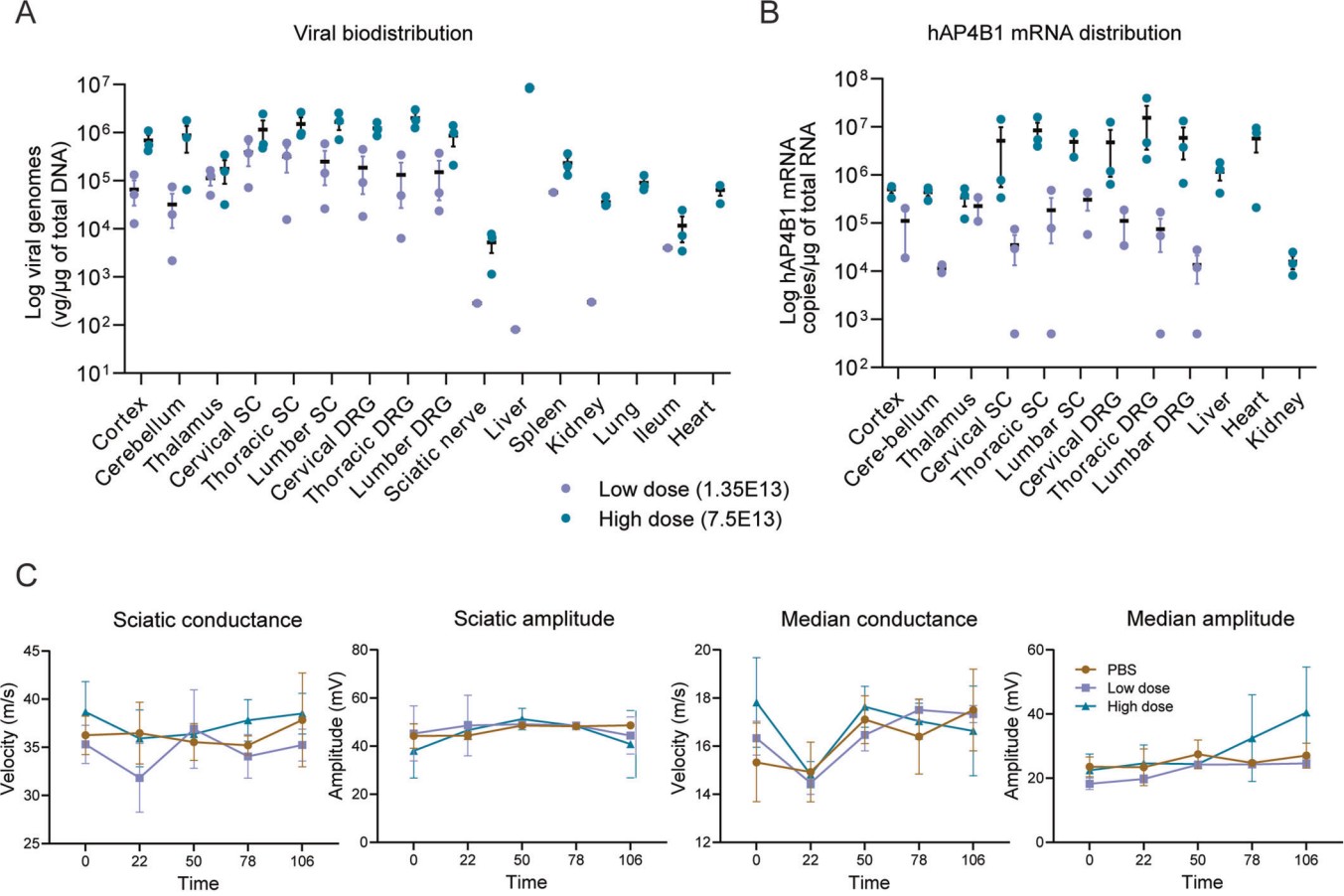

**Figure 9. In vivo safety study in WT NHPs showed good viral biodistribution and hAP4B1 expression across the CNS, minimal-mild macroscopic changes in DRG, spinal cord and sciatic nerve 1 and 4 months following ICM treatment of AAV9/hAP4B1, with no adverse effect on nerve conductance.**

V5-empty, low ($3.2 \times 10^{12}$ vg/kg), and high ($1.7 \times 10^{13}$ vg/kg) doses of AAV9/hAP4B1 vector were administered intra-cisternal to aged matched NHPs ($n = 2$ NHPs for V5-empty, $n = 3$ NHPs per treatment group (low or high dose). At 29 days and 113 days after injection, NHP organs were harvested. (A) Viral vector distribution assessment by qPCR of total DNA extracted from tissues at 16 weeks post vector administration. (B) hAP4B1 transgene expression in various tissues by RT-qPCR of total RNA extracted from NHPs at 16 weeks post vector administration. (C) Nerve conduction velocity (NCV) tests were performed at baseline and demonstrated no difference between test article and control (see Appendix Table S2 for data set). All data are presented at mean ± SEM. Source data are available online for this figure.

The initial in vivo studies aimed to determine the most efficient delivery method for viral distribution, AP4B1 mRNA and protein expression throughout the CNS. These investigations compared IV and ICM delivery routes 8 weeks post-administration. Results revealed extensive viral distribution in CNS tissues, including the cerebrum, cerebellum, and spinal cord, following ICM delivery in comparison to IV administration. Consequently, ICM delivery demonstrated the highest levels of hAP4B1 mRNA expression in the CNS with low expression in the periphery. ICM delivery enabled restoration of Ap4e1 protein levels in the cerebrum, cerebellum, and spinal cord, indicating successful exogenous AP4B1 expression and partial restoration of the AP4 complex. While the AAV9 vector can penetrate the BBB, low transgene expression is observed with IV administration because efficient transduction in brain regions and the spinal cord require significantly higher doses. However, increasing the viral dose raises the risk of high viral load in peripheral organs like the heart and liver, which are more readily transduced when AAV9 is administered intravenously and could cause adverse effects.

In our studies, we observed variation in vector genome copies in tissues between two AAV9 vectors (AAV9-CBh-hAP4B1 vs AAV9-V5-empty). There are three potential explanations: (i) The AAV9-CBh-hAP4B1 sequence is larger than AAV9-V5-empty, potentially causing different capsid conformations, therefore affecting cellular uptake. Figure 3A shows AAV9-CBh-hAP4B1 consistently has higher levels than AAV9-V5-empty, especially in the liver. (ii) It is known that AAV packaging and vector yield is reduced when the transgene sequence is small, which is the case for V5 sequence. (iii) Experimental Variability: Minor differences in injection procedures and immune clearance among mice may also contribute to observed variations. In addition, the three factors listed above may also explain the variability in vector genome distribution seen in brain tissues when comparing AAV9-CBh-hAP4B1 and AAV9-SYN-hAP4B1 vectors (Fig. 5E).

We decided to adopt a single-dosage approach in all our therapeutic efficacy and safety studies. This rationale was based on the facts that AAV9 gene therapy requires only one administration for a long-lasting effect. The AAV9 vector remains as an episome in

transduced cells, leading to stable, long-term transgene expression in non-dividing cells such as neurons. Our studies revealed sustained transgene expression and therapeutic effects for 9 months post-administration. Since AAV9 does not integrate into the genome, repeated dosing is needed only in tissues with high cellular turnover. Single administrations have proven effective for various CNS diseases (Kang et al, 2023), supporting our rationale for not investigating repeated dosing for this genetic disease.

Efficacy studies in neonatal treatment (~P2) and young adult mice treatment (~P60) showed recovery of various phenotypic parameters. In both studies, pathological readouts were assessed by motor function (rotarod and clasping), anatomical (corpus callosum thickness and lateral ventricle enlargement), biomolecule fluorescence (ATG9A localisation) and biomarker studies (neurofilament concentration). Administering AAV9/CBh-hAP4B1 and AAV9/SYN-hAP4B1 to neonates led to full restoration of rotarod performance six months after treatment, along with a significant improvement in the clasping phenotype, showing a 60.6% reduction in clasping severity at 6 MPT and 66.9% at 9 MPT. These outcomes not only demonstrate a rescue of the motor function phenotype, but they also indicate a stable therapeutic impact on motor function. In our adult treatment study both mid- ($4 \times 10^{12}$ vg/kg) and high-dose ($5 \times 10^{12}$ vg/kg) treatment groups showed a complete rescue of rotarod performance, and all three doses displayed a significant rescue of the clasping phenotype by 4 MPT. This indicates that the mild motor function phenotypes observed in these mice can still be rescued with a late-stage intervention.

While both AAV9-CBh-hAP4B1 and AAV9-SYN-hAP4B1 significantly improved corpus callosum thickness and ventriculomegaly phenotypes in the neonatal study, no effects were observed when treated at P60. It is likely that these brain anatomical abnormalities develop and exacerbate during neonatal brain development, supporting early intervention with gene therapy to prevent further deterioration or allow for enhanced recovery due to greater neural plasticity (Gray, 2016). Mature neurons are post-mitotic, making anatomical rescue unlikely, though function can still be altered/restored. Despite better outcomes with neonatal treatment, this does not optimally model the clinical scenario for patients. The age of presentation of developmental delay is generally in infancy, but diagnostic confirmation of SPG47 ranges widely, from as early as 12 months to as late as 12 years of age (Ebrahimi-Fakhari et al, 2018b). Treatment at P60 in mice is suggested to translate to approximately 6–7 years in humans (Dutta and Sengupta, 2016); therefore, treatment at this age can address whether the therapy can halt and reverse the disease progression in those diagnosed later in life. Although no anatomical changes were seen with adult treatment, significant functional recovery of motor phenotypes was observed in Ap4b1-KO mice with our high dose. Our preclinical data will guide the design of future human clinical trials.

Interestingly, we detected a significant rise in plasma neurofilament light chain (pNfL) levels in Ap4b1-KO mice at 9 months of age compared to WT mice. It is important to note that we did not observe any difference in pNfL levels from these mice at 2 months of age, indicating that this phenotype, in terms of neuronal injury, becomes more pronounced with age. This observation aligns with a previous study that reported elevated levels of pNfL in patients with AP-4 deficiency. Patients exhibiting higher pNfL levels were found

to have a positive correlation with severe generalised-onset seizures and developmental stagnation (Alecu et al, 2023). In the neonatal study, both AAV9/CBh-hAP4B1 and AAV9/SYN-hAP4B1 vectors successfully restored pNfL levels to below the lower limits of detection of the assay. These data suggest that the therapeutic vectors can prevent axonal degeneration from the intervention stage onward and pNfL could serve as an ideal biomarker for AP-4 deficiency clinical trials.

Correcting the molecular phenotypes of AP-4 deficiency in vivo, particularly targeting the AP-4 cargo ATG9A, holds significant promise for the treatment of the associated neurodegenerative pathology. We observed in the neonatal study that AAV9/AP4B1 treatment demonstrated a reduction in Atg9a perinuclear expression (accumulation in the TGN) in various brain regions including, the brainstem, cerebellum, hippocampus and cortex. In the adult dose escalation study, we conducted a robust analysis of the perinuclear accumulation of Atg9a in the same four brain regions. We observe a clear dose-dependent reduction in the perinuclear localisation of Atg9a leading to restoration of normal cell localisation of AP4 cargo, indicating a significant rescue of Atg9a trafficking. High-dose treatment demonstrated notable efficacy in the recovery of Atg9a trafficking across all brain regions. Intriguingly, beyond the dose-dependent effect, the Atg9a localisation study suggests a location-dependent recovery of Atg9a trafficking. Brain regions in proximity to the injection site or regions that expressed higher levels of AP4B1 mRNA corresponded to a greater degree of recovery. Multiple studies from several independent groups have converged on aberrant autophagy due to Atg9a missorting as the mechanism underlying neuronal pathology in AP-4-deficient patients (Mattera et al, 2017; Davies et al, 2018; De Pace et al, 2018; Ivankovic et al, 2020). In keeping with this, overexpression of Atg9a in primary neurons from Ap4e1 KO mice rescued defects in the clearance of axonal protein aggregations (De Pace et al, 2018). Therefore, improved ATG9A trafficking is likely to have a protective effect on neurons and could be linked to the reduced impairment of motor function observed with treated animals.

To support the in vivo efficacy data, three safety studies were performed to assess the safety/toxicity of ICM administration of AAV9-CBh-hAP4B1. These included a short-term study in C57BL-6J mice (for splenocyte reactivity assays), a non-GLP 1-year study in C57BL-6J mice and a GLP regulatory 4-month study in NHPs. Splenocyte reactivity assays detected no reactivity with the AP4B1 peptides. During the one-year study, mice treated with our therapeutic vectors did not display any adverse symptoms or major histopathological changes compared with control animals. Transgene biodistribution in the 1-year study showed a higher transduction of CNS tissues with the AAV9-CBh-hAP4B1 vector compared with the AAV9-SYN-hAP4B1 vector. Higher levels of expression were also observed within the liver and heart with the AAV9-CBh-hAP4B1 vector however this did not correlate with any long-term adverse events. Although the SYN promoter was not initially included in the earlier experiments, it was investigated in the long-term neonatal study and the WT safety/biodistribution mouse study. SYN promoter was added as a potential contingency plan in case the long-term studies showed that the AAV9-CBh-hAP4B1 vector resulted in adverse effects. Unlike the CBh promoter, which is strong at transducing peripheral organs, the SYN promoter is a more neuronal specific promoter. Nevertheless, the CBh promoter exhibited no off-target effects, demonstrated enhanced transduction of brain regions (in both

the neonatal long-term study and the WT safety study), and often showed a slightly higher degree of rescue of the disease phenotypes in the neonatal treatment experiments.

Toxicology studies conducted in NHPs, which serve as a better predictive model for human responses, revealed mild axonal degeneration and inflammation in various areas including the dorsal root ganglia (DRG), spinal cord, and sciatic nerve, with the DRG showing the most pronounced effects. These effects were increased on a dose- and time-dependent basis; at 4 months post treatment high-dose ($1.7 \times 10^{13}$ vg/kg) animals had elevated axonal degeneration and inflammation yet this did not reach above a mild severity score. While some adverse microscopic findings were documented, robust nerve conduction tests were completely normal indicating there was no functional impairment due the mild treatment-related degenerative damage that was observed.

Although inflammatory damage to the DRG in NHPs have been reported as a safety concern for AAV-mediated CNS gene therapy; these are mild-to-moderate in most cases and are characterised by mononuclear cell infiltrates, neuronal degeneration and secondary axonopathies of the spinal cord (Perez et al, 2020; Hordeaux et al, 2020). Hordeaux et al (2020) conducted a meta-analysis of various preclinical NHP studies that compared administration route, dose, time course, capsid, promoter, transgene and animal age and the general outcome was that various parameters can affect severity of DRG pathology. These studies demonstrated that dose and age at the time of administration significantly affected severity, whereas juvenile animals had less severe DRG degeneration than adults. It was also revealed that severity of DRG pathology was heightened at 1–5 months post injection, less severe in the initial stages (<1 month) and subsided 6 months post-injection. This suggests the animals can recover from these microscopic pathologies in nervous tissue over time. Interestingly, the study compared 5 capsids, 5 promoters and 20 different transgenes and it was found that the treatment-related DRG pathology was consistent regardless of vector design (except for secreted transgenes which increased severity by 20–25%). In all cases, none of the animals receiving the therapeutic vector displayed any adverse clinical signs and very few animals showed signs of altered nerve-conduction velocity. In animals with altered velocity this correlated with the severity of the peripheral nerve axonopathy (Hordeaux et al, 2020).

One GLP-compliant study using AAV9-mediated gene therapy for a neurodegenerative disease studied the response in 13 rhesus macaques for 90 days post administration (Hordeaux et al, 2018). Minimal-to-moderate dorsal root ganglia toxicity was observed to be asymptomatic. These are similar findings to those in piglets or macaques that are administered treatment systemically (Hinderer et al, 2018). DRG neurons are evidently highly transduced through both intravenous and intrathecal delivery routes of treatment, this is likely because DRG structures are highly vascularised and exposed to the flow dynamics of the CSF. Despite this, evidence suggests that AAV-induced DRG damage at the lower doses are unlikely to become a clinical concern and patients can recover from the microscopic changes (Hordeaux et al, 2018). Chen et al, in their development of gene therapy for SPG50, similarly observed DRG pathology with their lower dose of $8.4 \times 10^{13}$ gc/animal. However, they reported DRG pathology alongside reduced sensory conduction with their high-dose treatment of $1.68 \times 10^{14}$ gc/animal (Chen et al, 2023) and this dose was avoided in their clinical trials. The

lower dose used in the Chen et al study is comparable to the higher dose for our NHP study, and despite the presence of some microscopic pathologies in this study treatment was also well tolerated. The effects on DRG inflammation in our study were not extensive and are already known effects of clinical AAV9 treatment. Thus, our high dose treatment can be deemed acceptable in the context of treating a severe neurological disorder.

The choice of delivery method for gene therapy holds promise for reducing dosage and potential toxicities while enhancing transduction in key regions of the CNS. Extensive studies in NHPs have been conducted to anticipate the vector's distribution in humans across different administration routes. For instance, (Meyer et al, 2015) demonstrated that CSF delivery could achieve comparable CNS transduction with a 30-fold lower dose compared to intravenous injections. (Hinderer et al, 2014) found that intracisternal injections were up to 100-fold more effective than intrathecal lumbar puncture for brain gene transfer and more efficient for spinal cord gene transfer. In addition, (Rosenberg et al, 2018) showed that intracisternal delivery provided broader CNS distribution compared to intraventricular and intraparenchymal routes. These findings imply that delivering treatments through the cisterna magna in humans may enhance gene transfer efficiency and therapeutic efficacy while requiring lower dosages. CSF leakage and peripheral tissue transduction has been detected in lumbar puncture and cisterna magna administration of rodents and NHPs (Chandran et al, 2023; Ohno et al, 2019; Hinderer et al, 2014; Meyer et al, 2015). Transduction levels of peripheral tissues, in particular the liver, vary among different literature reports. In our neonatal study, we observed low levels of transgene expression in the liver, while significant liver transduction was observed in our adult study (injection at P60). This observation aligns with our safety studies: Safety Study 1 in neonates showed low levels of liver transduction, whereas Safety Study 2 (injection at P42) demonstrated high levels of liver transduction. Importantly, no adverse physical or histopathological effects were observed over a 1-year period in the mice with significant liver transduction. In this case, it is possible that the age of the mouse influences the fluid dynamics of the cerebrospinal fluid (CSF) and its potential for leakage into the peripheral system. Another approach to reducing the dose of AAV involves integrating a potent promoter (i.e., human cytomegalo-virus (CMV) enhancer/chicken beta-actin) into the vector design to boost transgene expression. This theoretically enables a decrease in the therapeutic dose of the AAV9 vector and mitigates AAV9-related neuronal toxicities. Zolgensma adopted this promoter in their AAV9-based vector therapy for spinal muscular atrophy. Similarly, in our investigation, we were able to incorporate the strong promoter chicken β-actin hybrid (CBh). However, due to its size, this promoter combined with larger transgene lengths often exceeds the limits of AAV9 packaging capacity so this strategy cannot always be implemented. Strong promoters can carry the risk of heightened off-target effects in peripheral tissues (Lukashchuk et al, 2016), therefore, rigorous safety and toxicity studies must be conducted to assess the potential for off-target effects. Taken together, our safety studies demonstrated a low risk of off-target effects of hAP4B1 overexpression when using the CBh promoter. A previous report demonstrated that AAV with CBh promoter mediates strong gene transfer in neuronal and glia cell types in various structures of the brain, spinal cord and dorsal root ganglia (DRG) (Gray et al, 2011).

Following submission of our Pre-IND (investigational new drug), FDA guidance to us was to scale up human dosing by cerebrospinal fluid (CSF) volume. We now have a draft clinical design including dosing for clinical human trials in SPG47 patients. The planned human trial dose (summarised in Appendix Table S3) was calculated based on our proof-of-concept mouse studies, long-term safety study in mice, GLP regulatory NHP toxicology study and literature (Chen et al, 2023). The high dose treatment that was validated as safe in non-human primate studies (cynomolgus monkeys), normalised to CSF volume, corresponds to a human dose of $4 \times 10^{14}$ genome copies in a 4-year-old child. Based on the adult dose response and non-human primate toxicology studies we propose a safe and efficacious dose range that corresponds to $5 \times 10^{12}$ to $6 \times 10^{12}$ vector genome/ml of CSF. While significant phenotypic improvement was observed in mice, it's possible that a higher dosage may be necessary in humans to achieve comparable outcomes due to the mild phenotypes presented in the diseased mice. Therefore, we may utilise a higher dosage ($6 \times 10^{12}$ viral genomes/ml of CSF) that has been validated as safe in non-human primate studies.

As patients age and neurodevelopment progresses it is probable that the treatment's effect may diminish. This was evident in the loss of efficacy observed on the corpus callosum and lateral ventricle phenotypes in mice treated at P60. Onset of major neurodegenerative phenotypes occurs between 4 and 6, thus targeting disease treatment before this age should increase chances of efficacy. To validate the therapeutic dose and its effectiveness in humans, initial clinical trials will focus on younger patients (1–5 years old). Once these trials prove successful, older patients (>5 years old) can be enrolled to further test the therapeutic vector. It is possible that older patients may require a larger therapeutic dose within our established safe range.

SPG47 is an ultra-rare disease and apart from work on small molecule drugs that have recently been identified to restore AP-4-dependent protein trafficking in neuronal models of AP-4 deficiency (Saffari et al, 2024), there is minimal literature on potential therapeutic strategies for this disease. Our study has demonstrated the safety and efficacy of AAV9/hAP4B1 gene therapeutic approach bringing it close to benefit SPG47 patients. Our next objective is to translate our proposed therapy into clinical trials through an investigational new drug (IND) application.

## Methods

### Reagents and tools table

| Reagent/Resource | Reference or Source | Catalog Number |
|---|---|---|
| **Experimental Models** | | |
| HeLa-M WT cells | Frazier et al, 2016 | N/A |
| AP4B1 KO Hela cells | Frazier et al, 2016 | N/A |
| SPG47 patients Fibroblasts | Provided by Professor H. Houlden (UCL) | N/A |
| Age/gender matched control fibroblasts | Provided by Professor H. Houlden (UCL) | N/A |
| SPG47 iPSC-derived neurons | 2-year-old male diagnosed with *AP4B1*-associated SPG47 | N/A |
| Control iPSC-derived neurons | Fibroblasts from the clinically-unaffected sex-matched parent (heterozygous variant) | N/A |

| Reagent/Resource | Reference or Source | Catalog Number |
|---|---|---|
| SH-SY5Y WT cells | Davies et al, 2018 | N/A |
| SH-SY5Y AP4B1 KO cells | Davies et al, 2018 | N/A |
| C57BL-6 WT mice | Jackson Laboratory: C57BL/6J-Ap4b1em5Lutzy/J | Strain #: 031349 |
| C57BL-6 *ap4b1* KO mice | Jackson Laboratory: C57BL/6J-Ap4b1em5Lutzy/J | Strain #: 031349 |
| Cynomolgus monkey (WT) | Novaxia Pathology Laboratory | N/A |
| **Plasmids/Vectors** | | |
| pTRS-KS-CBh | Provided by Dr S. Gray (University Texas Southwestern) | N/A |
| pAAV/CBh-MCS | Provided by Prof M. Robinson (University of Cambridge) | N/A |
| pLXIN-AP4B1 | Provided by Prof M. Robinson (University of Cambridge) | N/A |
| pLenti/PGK-MCS-Vos | Provided by Prof K. de Vos (University of Sheffield) | N/A |
| pCIneo_V5-N | Provided by Prof K. de Vos (University of Sheffield) | N/A |
| pHelper | PlasmidFactory | N/A |
| pAAV2/9 | PlasmidFactory | N/A |
| Pseudotyped AAV9 | Lukashchuk et al, 2016 | N/A |
| pAAV/CBh-V5-hAP4B1 | Fabricated in house | N/A |
| LV/V5-empty | Fabricated in house | N/A |
| LV/V5-AP4B1 | Fabricated in house | N/A |
| AAV9/V5-empty | Vector builder | N/A |
| AAV9/CBh-hAP4B1 | Cornell | N/A |
| AAV9/SYN-hAP4B1 | Vector Biolabs | N/A |
| AAV9/CBh-mAP4B1 | Vector Biolabs | N/A |
| **Antibodies IF** | | |
| Rabbit-anti-ATG9A | Abcam | Cat# ab108338 |
| Sheep-anti-TGN46 | Bio-Rad | Cat# AHP500G |
| Anti-Golgin 97 | Abcam | Cat# 169287 |
| Mouse-anti-beta-Tubulin III | Synaptic Systems | Cat# 302304 |
| Mouse-anti-beta-Tubulin III | Sigma | Cat# T8660 |
| Mouse-anti-calbindin | Abcam | Cat# ab82812 |
| Anti-rabbit-Alexa Fluor 488 | Thermo Fisher | Cat# A-11073 |
| Anti-sheep Alexa Fluor 558 | Thermo Fisher | Cat# A-11016 |
| Anti-mouse Alexa Fluor 647 | Thermo Fisher | Cat# A-21235 |
| Hoechst dye | Sigma | Cat# 94403 |
| **Western Blot** | | |
| Pre-cast 4–20% mini-PROTEAN gels | Bio-Rad | Cat# 4561095 |
| 245 kDa protein ladder | Abcam | Cat# ab116028 |
| Anti-mouse AP4E1 | BD Biosciences | Cat# 612019 |
| Anti-rabbit GAPDH | CST | Cat# 14C10 |

| Reagent/Resource | Reference or Source | Catalog Number |
|---|---|---|
| Anti-rabbit IgG (HRP) | Thermo Fisher | 31460 |
| Anti-mouse IgG (HRP) | Thermo Fisher | 31430 |
| **Oligonucleotides and other sequence-based reagents** | | |
| M18s forward primer | 5′-GTAACCCGTTGAACCCCAT-3′ | N/A |
| M18s reverse primer | 5′-CCATCCAATCGGTAGTAGCG-3′ | N/A |
| hAP4B1 forward primer | 5′-CTGGTGAACGATGAGAATGT-3′ | N/A |
| hAP4B1 reverse primer | 5′-GACCCAGCAACTCTGTTAAA-3′ | N/A |
| **Chemicals, Enzymes and other reagents** | | |
| RIPA buffer | 50 mM Tris-HCL pH 6.8; 150 mM NaCl; 1 mM EDTA; 1 mM EGTA; 0.1% v/v SDS; 0.5% v/v deoxycholic acid; 1% Triton X-100 | N/A |
| 1x protease inhibitor cocktail | Sigma-Aldrich | Cat #539136 |
| RNeasy® lipid tissue mini kit | QIAGEN | Cat #74804 |
| Proteinase K | Thermo-fisher | EO0491 |
| QIAzol® lysis reagent | QIAGEN | 79306 |
| NEBcloner kit | New England Biolabs | N/A |
| QIAEX II® gel extraction kit | QIAGEN | 20021 |
| QuantiNova SYBR® Green qPCR kit | QIAGEN | 208052 |
| Brilliant III Ultra-Fast SYBR® Green QPCR kit | QIAGEN | 600883 |
| Mouse IFN-γ ELISpot BASIC kit | Mabtech | #3321-2A |
| **Histology** | | |
| Charged glass slides | Starfrost | #MBB-0302-55A |
| Haematoxylin | Cellpath | #RBA-4205-ooA |
| Eosin solution | Leica | #3801590E |
| Xylene | Fisher | #X/0200/17 |
| DPX mounting medium | Cellpath | SEA-1300-00A |

## Preparation of experimental models

All our cell lines are tested regularly for mycoplasma contamination.

### *AP4B1*-KO HeLa cells

HeLa-M cells and *AP4B1* knock-out HeLa cells (Frazier et al, 2016) were provided by Prof M. Robinson (University of Cambridge). The HeLa cells were maintained in a growth media (Dulbecco's modified Eagle media (DMEM)), supplemented with 1X penicillin-streptomycin and 10% foetal bovine serum (FBS). Cells were grown in T25 flasks at 37 °C under 5% $CO_2$.

### Fibroblasts lines

SPG47 patient fibroblasts and control fibroblasts were kindly provided by Professor H. Houlden (University College London). Fibroblasts were maintained in DMEM, supplemented with 10% FBS, 1% Pencillin-Streptomycin and Uridine, and grown in T75 flasks.

## Patient-derived iPSC neurons

Fibroblasts from a 2-year-old male diagnosed with *AP4B1*-associated SPG47 carrying the following compound-heterozygous variants: NM_001253852.3: c.1160_1161del (p.Thr387ArgfsTer30)/c.1345 A > T (p.Arg449Ter) and fibroblasts from the clinically-unaffected sex-matched parent (38 years old) carrying the heterozygous variant c.1160_1161del (p.Thr387Argfs*30) served as positive controls. Fibroblasts were reprogrammed to iPSCs as described previously (Eberhardt et al, 2021; Teinert et al, 2019). iPSC-derived neurons were generated using induced NGN2 expression following previously published protocols (Saffari et al, 2024).

## Vector design and preparation

### *Plasmid design*

The original pAV2 vector backbone was published in Laughlin et al (1983). The CBh promoter—including the optimised MVM intron region—was amplified by PCR from the pTRS-KS-CBh-eGFP plasmid kindly provided by Dr S. Gray and cloned into the MluI and EcoRI sites of the pAAV/CBh-MCS construct after removal of the CMV promoter by restriction digest. Full-length hAP4B1 cDNA was transferred to pAAV/CBh-MCS by PCR amplification and ligation from the pLXIN-AP4B1 plasmid kindly provided by Prof M. Robinson (University of Cambridge) (Frazier et al, 2016). The hAP4B1 cDNA sequence was cloned between the SalI and HindIII sites of the pAAV/CBh-MCS plasmid. A separate epitope-tagged construct (pAAV/CBh-V5-hAP4B1) was created to enable detection of hAP4B1 protein expression by ICC. A linkerless N-terminal V5 epitope tag was inserted immediately downstream of the hAP4B1 Kozak sequence by Q5-mediated site-directed mutagenesis using a back-to-back primer strategy designed for large insertions. For investigating gene transfer in human fibroblast cell lines, the V5-hAP4B1 transgene sequence was subcloned by restriction digest (SalI/NotI → XhoI/NotI) into the pLenti/PGK-MCS-Vos lentiviral backbone (kindly provided by Prof K. de Vos), downstream of the PGK promoter.

To serve as a negative control for the V5-tagged hAP4B1 construct, the V5 epitope alone was PCR amplified from the pCIneo_V5-N vector (gift from Prof K. de Vos) and cloned into the XbaI site of pAAV-CBh-MCS.

### *Viral vector preparation*

AAV2-ITR transgene transfer plasmids—created as described above—were amplified in NEB Stable *E. Coli* cells (New England Biolabs) and purified using Qiagen Plasmid Plus kits. Adenoviral helper genes (pHelper) and Rep-Cap genes (pAAV2/9) were supplied in *trans* and were obtained commercially through PlasmidFactory. Pseudotyped AAV9 viral vector was produced in-house following the protocol described in Lukashchuk et al, 2016. Lentiviral vector was produced in HEK293 adherent cells by PEI-mediated transient transfection of pMD.2G (VSV-G envelope), pCMVDR8.92 (viral genes supplied in *trans*), pRSV-Rev (HIV-1 *rev*) and the pLenti transfer plasmid containing the V5-empty or hAP4B1 transgene. Lentivirus was purified by filtration of cell

supernatant followed by ultracentrifugation. Purified LV vector was resuspended in 1% BSA in PBS, aliquoted and titred using qPCR analysis of transduced HeLa cells alongside Fluorescence Activated Cell Sorting analysis of a reference LV of known titre expressing GFP.

## Viral transduction assays

### Treatment of HeLa and fibroblasts cells

HeLa-M cells/*AP4B1* knock-out HeLa cells or SPG47 fibroblasts/ control fibroblasts were plated in 6-well plates for protein assays or 24-well plates for immunocytochemistry. AAV9/AP4B1 and LV/ AP4B1 (added to HeLa cells and fibroblasts respectively) were added to the respective culture media without additional viral transduction enhancers. For AAV9/AP4B1 HeLa cell transduction MOI 2E5 and 4E5 were used. For LV/AP4B1 fibroblast transduction MOIs 5, 10, and 20 were used. Following 72 h treatment, cell media was removed from the plate, and washed with PBS and removed. For protein assessment, cells were incubated in lysis buffer for 5 min then removed from the well bottom via a cell scraper. Lysed cell solution was centrifuged for 10 min max speed and the supernatant collected for western blot analysis. For immunocytochemistry, cells were fixed with 4% PFA for 20 min, then permeabilized and blocked for 30 min with 5% normal donkey serum (NDS). 0.1% Triton X-100 in phosphate-buffered saline (PBS). Primary antibodies rabbit-anti-ATG9A at 1:500 (Abcam, Cat# ab108338) and sheep-anti-TGN46 at 1:800 (Bio-Rad, Cat# AHP500G) were used for co-staining and prepared in blocking solution (5% NDS in PBS). Following 1 h incubation with primary antibodies, cells were washed three times in PBS. The secondary antibodies anti-rabbit-Alexa Fluor 488 (Cat# A-11073) and anti-sheep Alexa Fluor 558 (Cat# A-11016) were prepared 1:1000 in PBS and incubated with cells for 1 h. Secondary antibody solution was removed and Hoechst dye (Sigma, cat 94403) at 1:2000 in PBS was added for 10 min. Following this, cells were washed three times in PBS and mounted on glass slides with mounting medium and stored at 4 °C.

### Treatment of patient-derived iPSC neurons

For assessment of ATG9A translocation, neurons were plated in 96-well plates at a density of $4 \times 10^4$ cells per well. Media changes were performed every 2–3 days and virus was administered at different MOIs at DIV (day in vitro) 8. 50% media changes were done at 4 h and 24 h after virus treatment, after that regular media changes every 2–3 days were resumed. Neurons were fixed after 7 days of treatment (DIV 14) in 4% PFA, and permeabilized and blocked using 0.1% Triton X-100/2% BSA/0.05% NGS in PBS. The following primary antibodies were added, diluted in blocking solution overnight at 4 °C: anti-ATG9A at 1:500 (Abcam, Cat# ab108338), anti-Golgin 97 1:500 (Abcam, Cat# 169287), anti-beta-Tubulin III 1:1000 (Synaptic Systems, Cat# 302304 and Sigma, Cat# T8660). Plates were gently washed three times in PBS, followed by addition of Hoechst 33258 and the following fluorochrome-conjugated secondary antibodies for 60 min at room temperature: Thermo Fisher Scientific (Cat# A11005, A-11073, A-21245, A11042, A11073), Abcam (Cat# ab150113) at a concentration of 1:2000, diluted in blocking solution. Plates were then gently washed three times with PBS and protected from light.

High-throughput imaging was performed using the Molecular Devices ImageXpress Micro Confocal Laser system, equipped with an 40× S Plan Fluor objective (NA 0.60 μm, WD 3.6–2.8 mm) using a modified pipeline from our previous study (Saffari et al, 2024). For each well, 36 sites in a $6 \times 6$ square shape were acquired, leaving out the centre square. Images were analysed with a customized image analysis pipeline in the MetaXpress software (Molecular Devices, version 6.7.1.157). For each staining a mask was determined. The intensity cut-off above the local background for each mask was adjusted for each plate to compensate for differences. Areas with a β-Tubulin III mask (cell body) containing a Hoechst mask (nucleus) were determined as cells. Overlapping Golgin-97 and ATG9A masks were defined as ATG9A inside the TGN. ATG9A mask not overlapping with the Golgin-97 mask but inside the cell area was defined as ATG9A outside the TGN. ATG9A ratio was calculated by dividing the F.U. of ATG9A inside the TGN by the F.U. outside the TGN.

For this study, two biological replicates were used. Across the two biological replicates, the Control + LV/AP4B1 group had a total of 13 technical replicates, Control + LV/GFP had 14 technical replicates, and both the Patient + LV/AP4B1 and Patient + LV/ GFP groups had 15 technical replicates. Within each technical repeat a minimum for 100 cells were analysed.

### Treatment AP4B1 knockout SH-SY5Y cells

*AP4B1* knockout SH-SY5Y cells (*AP4B1* KO) were a mixed population of CRISPR/Cas9-edited cells (Davies et al, 2018), kindly provided by Prof M. Robinson (University of Cambridge). SH-SY5Y cells were maintained in DMEM/F12 (Gibco, Cat# 11320033) supplemented with 10% heat-inactivated FBS (Gibco, Cat# 10438026), 100 U/ml penicillin and 100 μg/ml streptomycin at 37 °C under 5% $CO_2$.

## Potency assay development

Undifferentiated *AP4B1*-KO SH-SY5Y cells and wild-type *AP4B1* (*AP4B1*-WT) SH-SY5Y cells (5000 cells each) were plated on laminin-coated wells (5 μg/ml; Thermo Fisher Scientific, Cat #23017–015) in a 96-well plate 24 h before transduction. AAV9/ AP4B1 was added to the normal culture media without additional viral transduction enhancers. The required volume for each multiplicity of infection (MOI), ranging from $1 \times 10^3$ to $1.6 \times 10^7$, was calculated based on genome copies per ml. Six biological replicates were performed with up to eight wells per condition. As a control, 16 untreated *AP4B1*-KO SH-SY5Y wells and eight untreated *AP4B1*-WT wells per plate were used. Cells were treated for 72 h. Cells were fixed with 4% PFA for 20 min, and permeabilized with 0.1% saponin, and blocked with 0.1% BSA and 0.01% saponin in DPBS for 20 min. Primary antibodies rabbit-anti-ATG9A at 1:1000 (Abcam, Cat# ab108338), sheep-anti-TGN46 at 1:800 (Bio-Rad, Cat# AHP500G) and mouse-beta-Tubulin III 1:1000 (Sigma, Cat# T8660) were used. Following 1 h incubation in the primary antibodies, cells were washed three times with blocking solution. The secondary antibodies anti-rabbit-Alexa Fluor 488 (Cat# A-11073), anti-sheep Alexa Fluor 594 (Cat# A-11016), anti-mouse Alexa Fluor 647(Cat# A-21235) and Hoechst 33258 (Cat# H3569) were prepared 1:2000 in blocking solution. Cells were incubated for 30 min and washed three times with DPBS.

All solutions for immunostaining were filtered using a 0.22 μM filter system and plates were sealed opaque for imaging.

High-throughput imaging was performed using the Molecular Devices ImageXpress Micro Confocal Laser system, equipped with an 40× S Plan Fluor objective (NA 0.60 μm, WD 3.6–2.8 mm) using a modified pipeline from our previous study (Saffari et al, 2024). For each well, 24 sites in a 5 × 5 square shape were acquired, leaving out the centre square. Sites were 330 μm apart. Images were analysed with a customized image analysis pipeline in the MetaXpress software (Molecular Devices, version 6.7.1.157). For each staining a mask was determined. The intensity cut-off above the local background for each mask was adjusted for each plate to compensate for differences. Areas with a β-Tubulin III mask (cell body) containing a Hoechst mask (nucleus) were determined as cells. Overlapping TGN46 and ATG9A masks were defined as ATG9A inside the TGN. ATG9A mask not overlapping with the TGN46 mask but inside the cell area was defined as ATG9A outside the TGN. For each cell of every site and plate over 30 different cell parameters were extracted including perimeter, area, average intensity, roughness, homogeneity of the staining, or neurite outgrowth for the different stains. This allowed for in-depth analysis of cellular patterns and morphologic profiling. ATG9A ratio was calculated by dividing the F.U. of ATG9A inside the TGN by the F.U. outside the TGN. The ATG9A translocation was calculated based on the percent rescue of the ATG9A distribution in the experimental cells from *AP4B1*-KO SH-SY5Y back to *AP4B1*-WT SH-SY5Y ATG9A distribution.

The ATG9A ratio is a robust indicator of AP-4 function, offering an excellent quality metric for a high-throughput assay (Saffari et al, 2024). Therefore, the cell count generally has no impact on the ATG9A ratio.

## Western blotting

Lysate protein concentrations were determined using the Pierce BCA assay (Thermo Scientific Pierce™). Samples were prepared at 2 μg/μL in 5X Laemmli loading buffer (10 ml buffer contained: 240 mM Tris-HCL pH 6.8; 8% w/v SDS; 40% glycerol; 0.01% bromophenol blue; 10% β-mercaptoethanol) and denatured by heating to 55 °C for 5 min. Samples were loaded in pre-cast 4–20% mini-PROTEAN® gels (Bio-Rad, cat 4561095) alongside a 245 kDa protein ladder (Abcam, ab116028). For gel electrophoresis, gels were run for 1 h at 150 V in running buffer (25 mM Tris, 192 mM glycine, 0.1% SDS, pH 8.3). Following gels were transferred onto a nitrocellulose membrane using a transfer buffer (25 mM Tris, 192 mM glycine, 20% v/v methanol). Transfer was performed for 30 min at 100 V using the Bio-Rad Criterion™ blotter. Membranes were blocked with 5% milk-TBS (triphosphate buffered saline) for 1 h. Membranes were stained with primary antibodies anti-mouse AP4E1 1:1000 (BD Biosciences, cat 612019) and anti-rabbit GAPDH 1:1000 (CST, cat 14C10) in 5% milk-TBS-T (TBS-0.01% Tween) overnight at 4 °C. Following three 10-min washes in TBS-T, membranes were incubated with secondary antibodies 1:50,000 in 5%-milk-TBS-T (anti-rabbit IgG (HRP) and anti-mouse IgG (HRP)) for 1 h at RT. Finally, membranes were washed three times in TBS-T and images using the Li-Cor Odyssey FC imaging system. Quantification of protein signals was carried out through image studio. AP4E1 protein signal was normalised to the respective

GAPDH protein signal to account for loading differences, then each sample was normalised to the WT of that biological repeat. This means AP4E1 expression for each biological repeat of the WT samples will equal 1.

## Pre-clinical in vivo proof-of-concepts

### In vivo mouse model

C57BL/6J-Ap4b1em5Lutzy/J (Strain #: 031349) mice were generated by Jackson Labs via a CRISPR-Cas9 deletion within a 76 bp region within Exon 1 of the murine Ap4b1 gene. This deletion caused a frameshift mutation and resulted in a truncated mRNA transcript. The WT sequence (with the 76 bp deletion in lowercase) is:

TTGGCGACGATGCCATAccttggctctgaggacgtggtgaaggaactgaagaaggctctgtgtaaccctcatattcaggctgataggctgcgcTACCGGAATGTCATCCAGCGAGTTATTAGGTATCACCAACCTACCATAGAA

Monitoring and genotyping of this murine model were carried out as per our characterisation paper by Scarrott et al, 2023.

For behavioural and imaging analysis, mice/samples were identified by ID numbers or treatment group numbers and the operator was blinded to genotype/treatment during the assays and data collection.

### Route of administration study plan in Ap4b1-KO mice

*Ap4b1*-KO mice (P1–P3) were administered control (AAV9/V5-empty) and treatment (AAV9/CBh-hAP4B1) vectors through the intravenous (IV) or the intra-cisterna magna (ICM) delivery route up to $4 \times 10^{13}$ vg/kg. Wild type and untreated *Ap4b1*-KO mice were also recruited. At 2 months post injection mice were sacrificed for viral genomic distribution assays, hAP4B1 mRNA and Ap4e1 protein expression in various tissues ($n = 3$).

## Administration of viral vectors

### Neonatal IV injection

Mice pups (P1/P3) were anaesthetised (induction in chamber at 4% isoflurane, 4 L O₂/min. Maintenance on nose cones at 1–2% isoflurane, 0.5 L O₂/min for 5 min during injection) and were injected directly into the facial vein using a 33-gauge Hamilton syringe. A maximum volume of solution administered was 25 μl per gram mouse weight. The facial vein is used as an intravenous delivery route in neonatal mice due it being large and more easily accessible than the tail vein. In addition, neonatal mice tails are very delicate, and the tail vein can be prone to injury.

### Neonatal ICM injection

Mice pups (P1/P3) were anaesthetised as above. Mice were placed over a cervical support that allows the neck to be flexed at a 45-degree angle ventrally. Pups of this age have relatively translucent skin and holding the pup in this position means that the cisternal magna space could be visualised through the skin. A stereotaxic apparatus holding a 33 gauge-45-degree Hamilton syringe with an automated perfusion pump was used to accurately inject 5 μl over 5 min. Once the injection was complete pups were warmed briefly in hands and then returned to their cage with mother and littermates. Litters were not disturbed but monitored through the cage until weaning.

### Adult ICM injection

Mice ages ~P60 were anaesthetised in the induction chamber at 4% isoflurane, 4 L $O_2$/min. Mice were then shaved at the back of neck and skull between the ears while anaesthetised. Mice were then placed back into the induction chamber before fixing in place through ear bars and nose cone (2% isoflurane, 0.5 L $O_2$/min). After cleaning the area with Hibiscrub (4% w/v chlorhexidine gluconate), a linear skin incision was made down the midline from the top of the skull to the neck to reveal the neck muscles. The muscle layer over the neck was separated through the midline juncture with blunt forceps and held open with a wire retractor to expose the base of the occipital bone (sharp dissection of this muscle layer can cause unwanted bleeding). A stereotaxic apparatus holding a 10 μL Hamilton syringe was used to accurately inject 10 μl of virus into the cisterna magna. The muscle and skin were then sutured, and mice were returned to a recovery incubator (37 °C), where they remained until they were ambulatory.

### Proof-of-concept in neonatal mice

*Ap4b1*-KO mice (P1–P3) were administered control (AAV9/V5-empty) or two treatment vectors (AAV9/CBh-hAP4B1 and AAV9/SYN-hAP4B1) via the ICM delivery ($4 \times 10^{13}$ vg/kg). Wild type and untreated *Ap4b1*-KO mice were also recruited. A cohort of mice were sacrificed at 2 months post treatment for histopathological analysis on the tissues ($n = 3$). A second cohort of mice were sacrificed at 9 months post treatment for histopathological analysis and viral genomic distribution assays, hAP4B1 mRNA and Ap4e1 protein expression in various tissues ($n = 8$, four mice for histopathology and four mice for DNA, RNA and protein studies). Mice were monitored over the 9 months including taking weight and clasping assessment every fortnight. Various behavioural tests were undertaken including: open field, social recognition, rotarod performance, burrowing and nesting. All tests except rotarod performance showed no significant differences between wild type and *Ap4b1*-KO mice so these tests were dropped from the study analysis ($n = 8$).

### Dose-response study in adult mice

*Ap4b1*-KO mice (~P60) were administered the control vector (AAV9/V5-empty) or one of three doses of the treatment vector (Low dose: $5 \times 10^{12}$ vg/kg, mid-dose: $3 \times 10^{12}$ vg/kg, high-dose: $2 \times 10^{12}$ vg/kg). The dose selection used in this study was based on: (1) our previous studies using the therapeutic vector and AAV9 serotype in general; (2) Our extensive gene therapy studies using ICM as route of delivery for AAV9; (3) studies from the literature, for example recent gene therapy studies targeting the AP4 complex, e.g., SPG50 study (Chen et al, 2023). Wild type and untreated *Ap4b1*-KO mice were also recruited. 12 mice were recruited per treatment group. 3 mice per group were sacrificed at 2 months for a viral genomic distribution assay, hAP4B1 mRNA and Ap4e1 protein expression in various tissues. A cohort of mice were sacrificed at 4 months for plasma/CSF neurofilament, histological and anatomical analysis ($n = 6$). Finally, a cohort of mice were sacrificed at 4 months for viral genomic distribution assay, hAP4B1 mRNA and Ap4e1 protein expression ($n = 3$).

## Motor function recording

### Clasping test

Mice are held by the tail for 10 s and their hind limb splay/clasp is scored out of 3. 0—no clasping hind limbs are spayed. 1—one hind limb clasped for over 2 s. 2—both hind limbs clasped on and off. 3—both hind limbs clasped the majority of the time. Their hind limb clasping score was recorded once every 2 weeks around mid-morning.

### Rotarod test

An accelerating rotarod machine (Ugo Basile 7650) was used to measure the motor function. The rotary system accelerated from 3 to 37 rpm over 300 s then maintained a constant rpm. Mice are placed on the rotor and the seconds of time it takes for the mice to fall is determined as their latency to fall. Prior to the final recording, mice undergo rotarod training of two trials per day over 3 consecutive days. The final recording is taken from two trials, with a minimum of a 5 min rest period between trials. The best performance of latency to fall was used for analysis.

## Pre-clinical safety in vivo studies

### ELISpot safety study in WT BL/6J mice

Wild-type mice C57BL/6 (P1-3) were administered control vector (AAV9/V5-empty) or two treatment vectors (AAV9/CBh-hAP4B1 and AAV9/SYN-hAP4B1) via the ICM delivery ($4 \times 10^{13}$ vg/kg). Mice were sacrificed at 3 months and splenocytes were extracted, dissociated and ran through an ELISpot immunoassay against the hAP4B1 transgene peptides ($n = 4$). These peptides were a library of different combinations of chemically synthesised AP4B1 peptides. The splenocytes from treated mice were exposed to a known T cells activator, Concanavalin A (ConA) which gave a positive signal in all treatment groups, a substance known not to initiate a T cell response, dimethyl sulfoxide (DMSO) which gave a negative signal and the peptide library of AP4B1. The assay detects INF-γ secretion from activated T cells in response to the stimulant. For this we used the Mouse IFN-γ ELISpot BASIC kit (#3321-2A). INF-γ release was indicated through spot formation on the membrane of the ELISpot assay. Spots were then counted through an automated software (ImmunoSpot® Software).

## Non-GLP long-term safety in WT BL/6J mice

See Appendix Table S1.

### Ap4b1-KO mice and monitoring

Clinical observations were noted for all animals within the study each day. If animals did not appear to be bright, alert, responsive, and healthy (BARH), then a member of the veterinary staff and the study director were notified, and a decision would be made. If animals demonstrated a >10% loss of body weight over a week, displayed a decrease in body or movement condition and/or were unresponsive to external stimuli then mice were humanely euthanised and post-mortem examination was carried out.

Mice were typically weaned at 4 weeks of age and housed in clean plastic containers with litter mates of the same gender.

Mice were checked daily in the first two weeks post injection and weekly thereafter.

## GLP safety and toxicology study in WT NHPs

See Appendix Table S2.

## Tissue collection and processing

### Terminal procedure

All mice were perfused with phosphate-buffered saline under terminal pentobarbital anaesthesia, followed by collection of tissue for post-fixation (for anatomical and histological analysis) or flash frozen in liquid nitrogen (for RNA, DNA and protein extraction). Blood/plasma or CSF were also extracted from some mice cohorts.

Mice were administered terminal pentobarbital ~100 µl, and once unresponsive to hind limb pinch the chest cavity was opened and blood was extracted through a cardiac puncture in the left ventricle. Blood was centrifuged for 5 min at 12,000 RPM for plasma separation. The plasma was then transferred to a new tube and placed in liquid nitrogen. Before tissue dissection mice were perfused with 30 ml of sterile PBS through the same left ventricle puncture and a snip to the right atrium. Heart, liver, brain (cerebellum, brainstem and cerebrum) and spinal cord (cervical, thoracic and lumbar) were dissected, placed into screw cap tubes and frozen in liquid nitrogen or whole brains were fixed in 4% paraformaldehyde for 24 h. All tissues were kept at −80 °C until further use. Alternatively for anatomical and histological analysis whole brains were fixed in 4% paraformaldehyde in PBS for 24 h, then 15% sucrose (until brain sinks) and then 30% sucrose (until brain sinks), kept at 4 °C. Brains were then slowly frozen in OCT (optimal cutting temperature) solution over dry ice and stored at −80 °C until sectioning under the cryostat. Whole brains were sectioned using a Leica CM3050S cryostat to generate 30-µm-thick coronal slices and stored in 0.05% sodium azide-PBS at 4 °C.

### Cerebrospinal fluid (CSF) collection

Mice selected for CSF extraction before terminal sacrifice would be placed into the induction chamber (5% isoflurane, 5 L $O_2$/min) until unresponsive before fixing on the nose cone (2% isoflurane, 0.5 L $O_2$/min). An incision down the neck from in-between the ears would reveal the neck muscles. Neck muscles were opened down the midline to reveal the cisterna magna. A small glass capillary with needle-like tip was pierced through the membrane of the cisterna magna and CSF would flow into the capillary while held still for 1–2 min. After this, the terminal dose of pentobarbital was administered, and the terminal procedure carried out as usual.

## Brain tissue preparation

After 24 h in 4% paraformaldehyde fixation brains were placed in 15% sucrose PBS solution for another ~24 h. Once brains had sunk to the bottom of the vial they would then be placed in 30% sucrose PBS solution for ~24 h. Once brains had sunk they were ready for OCT (optimum cutting temperature) embedding. Brain embedding trays were filled with OCT solutions ensuring no bubbles. Brains were removed from the sucrose and excess sucrose solution was removed. Each brain was placed in a single tray and aligned before covering with OCT. The tray was then placed over dry ice where the OCT slowly freezes over and embedded the brain. Orientation of olfactory bulbs was marked and then placed in −80 °C ready for sectioning. Brains were sectioned coronally on the cryostat at 30 µm thickness and placed in 0.01% sodium azide PBS solution in 48-well plates and stored in the cold room until further use.

## Histology analysis

### Haematoxylin/eosin (H&E) staining of brain tissue

For H&E, free-floating brain cryosections were placed and dried on charged glass slides (Starfrost, #MBB-0302-55A). Sections underwent a series of processing steps as follows: 95% alcohol for 5 min, 70% alcohol for 5 min, wash in $H_2O$, haematoxylin (cellpath, #RBA-4205-ooA) solution for 2 min, wash in $H_2O$, Scott's water for 60 s, wash in $H_2O$, Eosin solution (Leica #3801590E) for 2 min, wash in $H_2O$, alcohol dehydration (70% to 90% to 100% to pure 100%) and storage in xylene (Fisher, #X/0200/17) until mounting. Glass coverslips were then mounted using DPX mounting medium (Cellpath, SEA-1300-00A) onto the H&E slides. Histopathological analysis was carried out by qualified pathologists.

### Immunofluorescence staining of brain tissue

Sections selected for ATG9A immunofluorescence staining were chosen depending on anatomical regions. Sections between groups were compared and the same section was selected (2 sections per region). Anatomical regions selected for ATG9A analysis were cortex, hippocampus, cerebellum and brainstem. A minimum of 3 areas per anatomical region were imaged as indicated in Fig. 1 and analysed for ATG9A expression.

Sections were incubated with blocking and permeabilisation solution (5% normal donkey serum (NDS), 1% triton X-100 in phosphate-buffered saline (PBS)) in a 48-well plate for 3 h at room temperature on an orbital shaker. Primary and secondary antibodies were diluted in staining solution (5% NDS in PBS). Sections were incubated with primary antibodies (mouse anti-ATG9A (Abcam, ab108338) dilute 1:100 or mouse anti-calbindin (Abcam, ab82812) 1:1000) overnight. Sections were then washed at room temperature with PBS 2× 30 min. Following this, sections were then incubated with secondary antibodies (donkey anti-mouse IGg fluorophore diluted 1:1000 (Thermo Fisher Scientific)) and Hoechst nuclear stain (1:1000) for 2 h at room temperature. After secondary incubation sections were washed 2× 30 min in PBS. All incubation steps and washes were done on an orbital shaker at 60 rpm with multiple sections per well.

## Microscopy and image analysis

For the anatomical analysis of the corpus callosum width and lateral ventricle size H&E slides were imaged using a Nanozoomer-XR scanner (Hamamatsu). For corpus callosum analysis 8 coronal positions were chosen starting from the first section that demonstrated the full corpus callosum. For lateral ventricle analysis 10 coronal positions were chosen starting at the beginning of the hippocampus. These sections were then stained with haemotoxylin and eosin (H&E). Images were analysed using Visiopharm where corpus callosum width was manually measured with the ruler tool and a region of interest was manually drawn aligning with perimeter of the lateral ventricles to calculate the area.

ATG9A stained brain sections were imaged using a Z stack Confocal (Leica TCS SP5 11) microscope at 63x lens. As expression of ATG9A varies, Z stack images were taken in specific brain locations (Appendix Fig. S4) to allow direct comparison of ATG9A expression between treatment groups. For each brain region (cortex, hippocampus, cerebellum and brainstem) a minimum of

3 images were taken at various locations. Hoechst nuclear stain was also imaged simultaneously. Using Fiji (ImageJ), Z-stack images were processed to obtain maximum intensity projections, then ATG9A perinuclear accumulation was measured. A consistent defined perinuclear region was generated for each brain region using the macro tool function, and the intensity of ATG9A fluorescence within the perinuclear region was measured for each neuronal cell within each image. These values were normalised to the background fluorescence intensity of that image. Anatomical and fluorescence image analysis was performed by an operator blinded to the genotypes or treatment group of each animal.

For the assessment of calbindin spheroid formation within the deep cerebellar nuclei (DCN) of each treatment group a semi-quantitative analysis was performed. Brain sections from the same location of the cerebellum were chosen for staining. Images were taken using the Zeiss LSM 980 Airyscan using the ZEN blue 3 microscopy software. The DCN was located with the 10x lens followed by x40 lens oil immersion z stack imaging of the calbindin stained DCN. Using Fiji (ImageJ), Z-stack images were processed to obtain maximum intensity projections. Calbindin-positive spheroids were manually counted in each image. By comparing wild-type (WT) and knockout (KO) DCNs, we classified a calbindin spheroid as >19.904 $\mu m^2$ using the macro function in ImageJ. Only calbindin spheroids larger than this predetermined threshold was included in our quantification (number of spheroids per image).

## RNA and DNA extraction from mouse tissue

### Lysis of tissue

Tissue sections were lysed in 600 µl RIPA buffer (50 mM Tris-HCL pH 6.8; 150 mM NaCl; 1 mM EDTA; 1 mM EGTA; 0.1% v/v SDS; 0.5% v/v deoxycholic acid; 1% Triton X-100) with a 1x protease inhibitor cocktail (Sigma-Aldrich). Disruption of tissue was achieved using a Precellys Evolution homogeniser (Bertin Technologies) with the following parameters: 5500 RPM at 2 × 30 s intervals with a 20 s pause; or 6800 RPM at 3 × 30 s intervals with a 20 s pause for fatty and non-fatty tissues respectively. Samples were then centrifuged (17,000 × $g$, 20 min, 4 °C) and tissue lysate transferred to a new tube. for storage at −20 °C.

### Mouse tissue RNA extraction

Tissue sections were homogenised using a Precellys Evolution homogeniser (Bertin Technologies) in 1 ml of QIAzol® lysis reagent with parameters: 5500 RPM at 2 × 30 s intervals with a 20 s pause for fatty tissue, and 6800 RPM at 3 × 30 s intervals with 20 s pause for non-fatty tissue. Following homogenisation, RNA was extracted using QIAGEN RNeasy® lipid tissue mini kit (#74804) as per manufacturer's instructions, including an on column DNAase step to ensure an absence of DNA in the samples. RNA was eluted in 30 µl DEPC $H_2O$ and quantified using a nanodrop.

### Mouse tissue genomic DNA extraction

For extraction of genomic DNA 2 µl proteinase K (20 mg/ml) was added to 100 µl of tissue lysate. Following a 1-h incubation at 50 °C, 100 µl phenol was added and samples were centrifuged (17,000 × $g$, 4 min) facilitating separation of the DNA containing aqueous phase. The aqueous phase was then transferred to a new tube and 10 µl of 3 M sodium acetate followed by 300 µl 100% ethanol were added. Samples were incubated overnight at −20 °C and centrifuged (13,000 × $g$, 20 min) before removal of the supernatant. DNA pellets were then washed in 70% ethanol and centrifuged (17,000 × $g$, 10 min). All remaining ethanol was removed, and pellets were air dried for 5 min before resuspension in 15 µl DEPC $H_2O$ and quantification of DNA using a nanodrop.

### Plasmid linearisation

To generate linearised plasmid for use in a standard curve, 1 µg of plasmid was digested according to instructions stated by NEBcloner®. Reactions were set up with 1 µg plasmid, 5 µl of CutSmart buffer, 1 µl of appropriate restriction enzyme and made up to 50 µl with DEPC $H_2O$. Digestion was carried out for 1 h at 37 °C and heat inactivated for 20 min at 80 °C. Linearised plasmid was resolved on an agarose gel and a UV transilluminator used to visualise the bands. Bands were excised using a scalpel and DNA extracted using QIAEX II® gel extraction kit according to manufacturer's instructions. Plasmid concentration was quantified using a nanodrop.

### RT-qPCR for human and murine AP4B1 mRNA expression analysis

Total RNA was diluted to 10 ng/µl in DEPC $H_2O$. RT-qPCR reactions were set up as follows: 2 µl RNA (10 ng/µl); 0.25 µl each of m18s primers (forward: 5′-GTAACCCGTTGAACCCCAT-3′; reverse: 5′-CCATCCAATCGGTAGTAGCG-3′), or 0.5 µl each of hAP4B1 primers (forward: 5′-CTGGTGAACGATGAGAATGT-3′; reverse: 5′-GACCCAGCAACTCTGTTAAA-3′) at 10 mM; 5 µl of QuantiNova SYBR® Green master mix; 0.1 µl Reverse Transcription mix and reactions made up to 10 µl with DEPC $H_2O$. All qPCR runs were carried out using a Bio-Rad C1000 Touch™ Thermocycler. A reverse transcription step was carried out for 10 min at 50 °C before a 2 min denaturation step at 95 °C. cDNA was then amplified with 39 cycles of a 5 s denaturation at 95 °C and 60 °C annealing/extension for 10 s before a cycle of 65 °C for 31 s and a final melt curve analysis. Signal intensities were analysed using Bio-Rad CFX Maestro™ software and the ΔΔCt method with m18s RNA as a reference gene were used to measure fold change in gene expression.

### qPCR for viral DNA biodistribution

To determine viral copy number in tissues, DNA was first diluted to 25 ng/µl in DEPC $H_2O$. qPCR reactions were made up to 10 µl with DEPC $H_2O$ and contained 1 µl DNA (25 ng/µl); 0.5 µl each of hAP4B1 primers (forward: 5′-CTGGTGAACGATGAGAATGT-3′; reverse: 5′ GACCCAGCAACTCTGTTAAA-3′) at 10 mM and 5 µl of Brilliant III Ultra-Fast SYBR® Green QPCR master mix. Plate setups included a standard curve of eight 1 in 10 dilutions of known linearised plasmid concentrations. qPCRs were run using a Bio-Rad C1000 Touch TM thermocycler. An initial 5-min denaturation step at 95 °C preceded amplification of cDNA by 39 cycles of a 10 s 95 °C denaturation and a 60 °C amplification/extension for 30 s. Melt curve analysis was then carried out following a cycle at 65 °C for 31 s. Signal intensities were analysed using Bio-Rad CFX Maestro TM software. The standard curve of known plasmid concentrations was used to calculate viral copy number and log(viral copies) were plotted against Ct values to give a linear curve. Ct values were then interpolated to determine viral copy number for unknown samples.

## The paper explained

### Problem

Hereditary spastic paraplegia (HSP) is characterised by global developmental delay, microcephaly, seizures, epilepsy, malformation of the brain, and hypotonia (low muscle tone). Inherited HSP type 47 (SPG47) is caused by mutations in one of the adaptor protein complex 4 genes (AP4B1). Disease occurs when a mutated copy of this gene is inherited from both parents. Currently, there is no effective treatment for this debilitating condition.

### Results

We designed a new gene therapy to treat SPG47 by delivering a healthy copy of the AP4B1 gene using adeno-associated virus as a carrier (AAV9/hAP4B1). In an SPG47 mouse model, a single injection of AAV9/hAP4B1 into the cerebrospinal fluid led to widespread gene transfer. This therapy corrected several disease features, such as mislocalization of AP-4 cargo (ATG9A), the presence of calbindin-positive spheroids, anatomical brain defects, and motor dysfunction. In addition, the treatment normalised plasma neurofilament light (NfL) levels, a biomarker for neurodegeneration. Preclinical safety studies in non-human primates revealed no significant adverse events.

### Impact

We report an efficacious and safe gene therapeutic approach for SPG47. The successful IND-enabling preclinical efficacy and safety package paves the way for a potential viable treatment for this currently untreatable condition.

## Neurofilament assay

Neurofilament assay was carried out using the S-PLEX ultra-sensitive assay platform. The assay was set up as per instructions from MSD. Whole plasma samples were first diluted 1 in 10 and whole CSF samples were diluted 1 in 20. 25 µl of the diluted sample was used in the assay per mouse. An initial dilution assay was conducted to ascertain the optimal dilution of CSF or plasma for the assay.

## Ethics statement

All in vivo animal experiments were approved by the University of Sheffield Ethical Review Sub-committee, the UK Animal Procedures Committee (London, UK) and performed according to the Animal (Scientific Procedures) Act 1986, under the Home Office Project License P31C8CC9D. C57 black 6 (C57BL/6J-AP4B1[em5Lutzy]/J (Strain #: 031349)) mice were housed in a controlled facility with a 12 h dark/12 h light photocycle (7 am–7 pm light) with limitless access to food and water. Fibroblasts were obtained following written informed consent (protocol approached at the Boston Children's Hospital Institutional Review Board: IRB-P00033016) and all experiments were conducted in accordance with the declaration of Helsinki.

## Statistical analysis

### Statistical tests

All quantitative data in this paper are presented as mean ± SEM or mean ± SD (this is specified in figure legends) and analysed/graphed using GraphPad Prism Software (v 9.0.2). Data were tested for normal distribution via the Shapiro–Wilk normality test. Data sets that passed normality checks were analysed with a one-way ANOVA with repeated measures and Tukey's multiple comparisons test was used for data sets with three or more groups or two-way ANOVA with Tukey's multiple comparisons tests were used for data sets involving nested groups. Data sets that did not pass the tests for normality were analysed through the Kruskal–Wallis test with Dunn's multiple comparisons test. A $p$ value of less than 0.05 was considered as significant for all statistical analyses.

### Sample size, blinding, and randomisation

We regularly work with biostatisticians to determine the sample size for our preclinical studies. Our Statistic Department at the University of Sheffield assisted us with power calculation for the proposed studies. Our sample size and statistical power depended on the disease phenotype under assessment. For instance, clear biochemical phenotypes with low variance between individual animals, such as ATG9A accumulation, required a smaller sample size to observe significant changes between wild type and Ap4b1 knockout mice. In this case, analysing 4 animals per group achieved high statistical significance. The alpha test significance was set to 0.05 for a two-sided test with statistical power of 80%. Conversely, milder behavioural/motor phenotypes with higher variability required a larger sample size. We determined that a minimum of 8 animals per group was necessary to achieve statistical significance between wild type and Ap4b1 knockout mice. Additional animals were included to account for dropout animals, early deaths unrelated to treatment, and potential technical failures in vector transmission.

Animals were randomly allocated to treatment groups to minimise subjective bias. No systematic method or criteria were used in the selection process; animals were simply picked at random. This approach ensured that each animal had an equal chance of being assigned to any treatment group, thus reducing the risk of selection bias. The random allocation process was not formally structured but was conducted in a manner that prevented any intentional or unintentional bias. For behavioural assays and imaging analysis, mice were identified by ID numbers, and operators conducting the data collection and analysis were blinded to the genotype or treatment group. Exclusion criteria from the behavioural studies would be mice that exhibited signs of illness or distress due to mutation-related phenotypes, general ill-health, or injury. However, no mice were excluded from the analysis based on these criteria.

## Data availability

This study includes data deposited in external repositories. Bioimages for Figs. 6 and 8 are deposited in Bioimage archive with the accession ID: S-BIAD1267.

The source data of this paper are collected in the following database record: biostudies:S-SCDT-10_1038-S44321-024-00148-5.

## Peer review information

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

## Acknowledgements

This work was supported by two awards from LifeArc (No. 163978 and P2022-0004, Recipient: 176666) and CureAP4 Foundation. This project has also received funding from the Innovative Medicines Initiative 2 Joint Undertaking under grant agreement No [945473]. This Joint Undertaking receives support from the European Union's Horizon 2020 research and innovation programme and EFPIA [Pfizer and Bayer]. MA is also supported by the European Research Council grant (ERC Advanced Award no. 294745), UKRI MRC (MR/V000470/1), Alzheimer's Research UK award (ARUK-PG2018B-005) and UKRI MRC award (MR/V030140/1). DE-F has received support from the Spastic Paraplegia Foundation, the CureAP4 Foundation, and the National Institutes of Health/National Institute of Neurological Disorders and Stroke (K08NS123552-01). PJS is supported by the NIHR Sheffield Biomedical Research Centre. LF is also supported by the UKRI MRC (MR/W00416X/10). Histopathology assessment was completed with support from the histology core facility at SITraN, University of Sheffield.

## Author contributions

**Jessica P Wiseman**: Data curation; Formal analysis; Supervision; Validation; Investigation; Visualization; Methodology; Writing—original draft; Project administration; Writing—review and editing. **Joseph M Scarrott**: Formal analysis; Investigation; Methodology; Writing—review and editing. **João Alves-Cruzeiro**: Data curation; Formal analysis; Investigation; Methodology. **Afshin Saffari**: Data curation; Formal analysis; Investigation; Methodology. **Cedric Böger**: Data curation; Formal analysis; Investigation; Methodology. **Evangelia Karyka**: Data curation; Investigation; Methodology. **Emily Dawes**: Data curation; Investigation; Methodology. **Alexandra K Davies**: Investigation; Methodology; Writing—review and editing. **Paolo M Marchi**: Data curation; Investigation; Methodology. **Emily Graves**: Data curation; Investigation;

Methodology. **Fiona Fernandes**: Data curation; Investigation; Methodology. **Zih-Liang Yang**: Data curation; Methodology. **Ian Coldicott**: Methodology. **Jennifer Hirst**: Conceptualization; Resources; Methodology. **Christopher P Webster**: Writing—review and editing. **J Robin Highley**: Data curation; Investigation. **Neil Hackett**: Conceptualization; Data curation; Formal analysis. **Adrienn Angyal**: Data curation. **Thushan de Silva**: Data curation. **Adrian Higginbottom**: Data curation. **Pamela J Shaw**: Resources; Funding acquisition; Writing—review and editing. **Laura Ferraiuolo**: Funding acquisition; Writing—review and editing. **Darius Ebrahimi-Fakhari**: Supervision; Funding acquisition; Writing—review and editing. **Mimoun Azzouz**: Conceptualization; Resources; Supervision; Funding acquisition; Investigation; Project administration; Writing—review and editing.

Source data underlying figure panels in this paper may have individual authorship assigned. Where available, figure panel/source data authorship is listed in the following database record: biostudies:S-SCDT-10_1038-S44321-024-00148-5.

## Disclosure and competing interests statement

MA is co-founder of BlackfinBio and Crucible Therapeutics. NH is a paid consultant to CureAP4. PJS is co-founder of Keapstone Therapeutics and Crucible Therapeutics.

# Expanded View Figures

**Figure EV1.  SPG47 patients' fibroblasts show a rescue in ATG9A expression when treated with LV/hAP4B1.**

(**A**) Patients' fibroblast stained with ATG9A (green) and TGN46 (red) show mislocalisation of ATG9A compared with healthy fibroblasts. SPG47 patient cells marked with white asterisks show rescue of mislocalised ATG9A after treatment with LV/V5-hAP4B1. Scale bar 20 μm (**B**) Representative western blot confirms expression of the hAP4B1 within the fibroblasts with increasing viral MOI. (**C**) Representative western blot shows the increase in ATG9A expression in KO fibroblasts ($p = 0.0005$) compared to WT and demonstrates the reduction in ATG9A expression when treated with increasing MOI of LV/hAP4B1 ($p = 0.0443$ for MOI 5, $p = 0.0014$ MOI 10, $p = 0.0011$ MOI 20). Corresponding quantification shows MOI 10 and 20 both rescue the ATG9A phenotype to healthy fibroblasts (control) levels. Data is presented as mean $+/-$ standard error of the mean (SEM), $n = 3$ biological repeats. Data analysed by one-way ANOVA followed by post hoc Dunnett's multiple comparisons test with respect to Ctrl. Stars indicate $p \leq 0.05$ (*); $p \leq 0.01$ (**), $p \leq 0.001$ (***); ns = not significant.

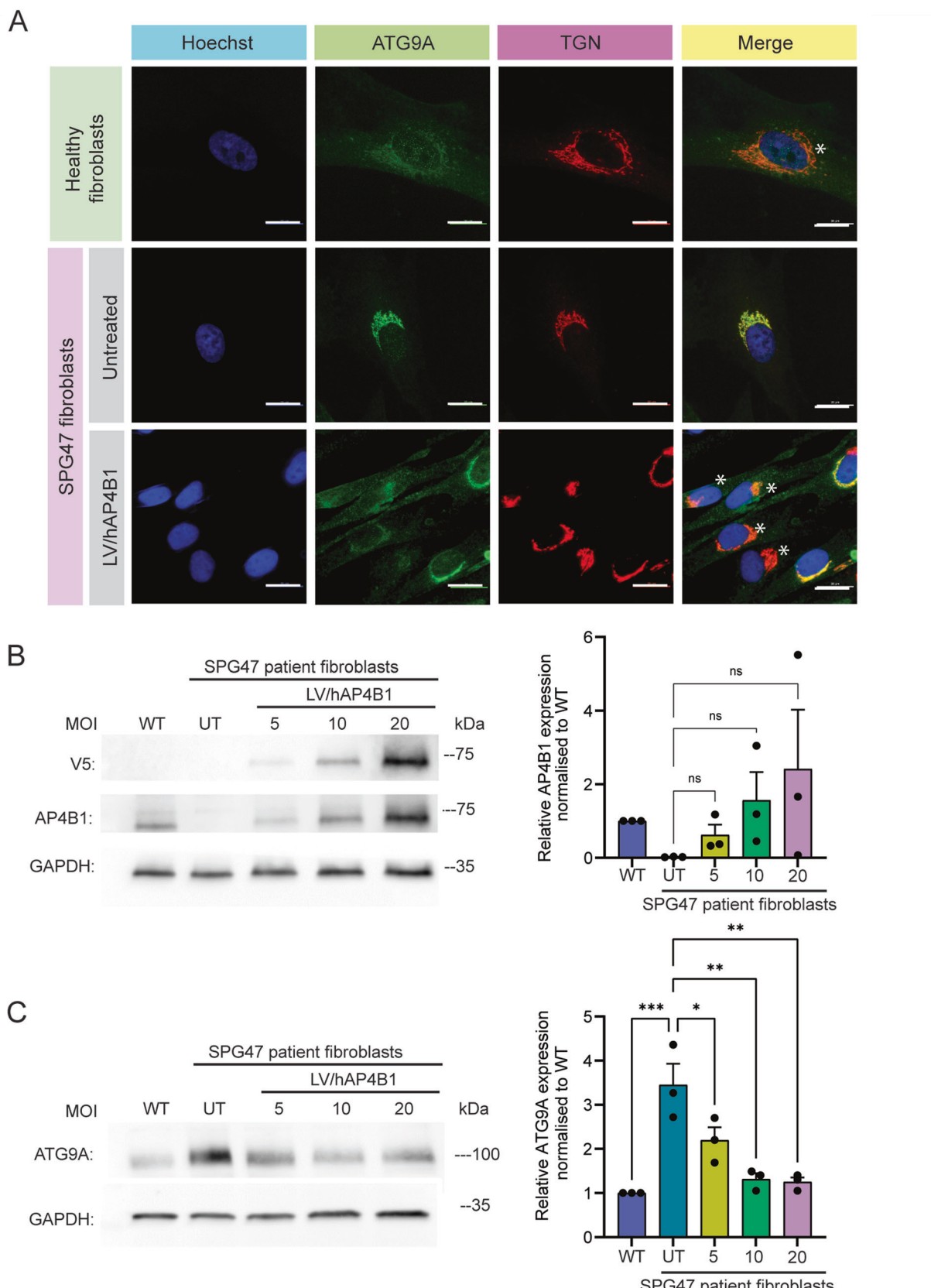

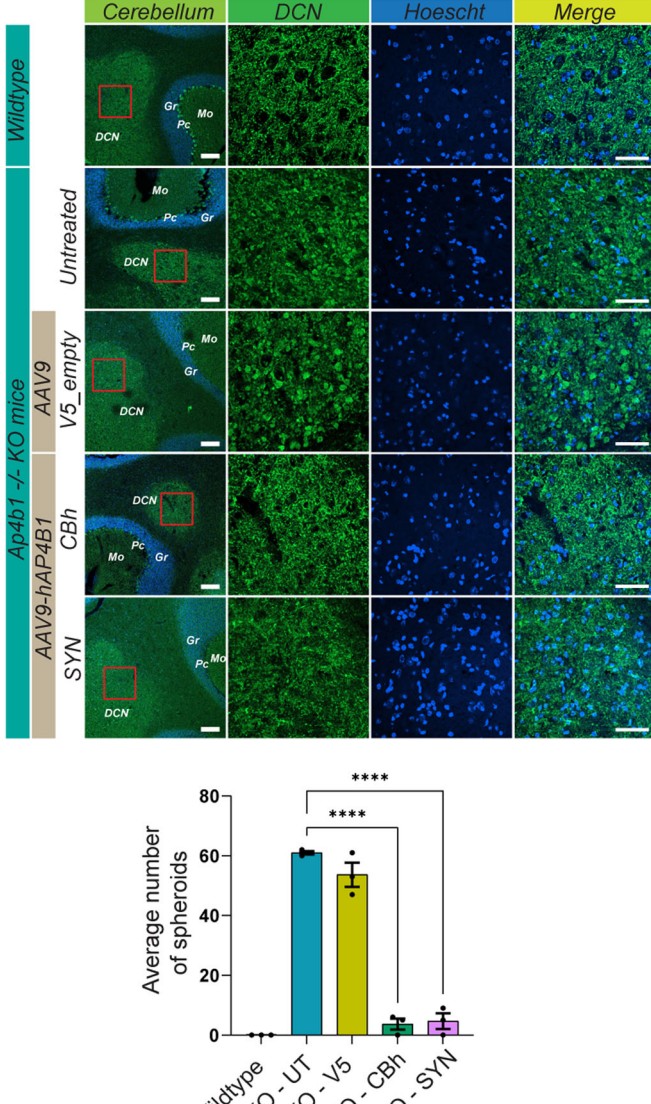

**Figure EV2.  Neonatal ICM treatment with AAV9/hAP4B1 in *Ap4b1*-KO mice rescue Calbindin-positive spheroids in the DCN 9 months following treatment.**

The image panel depicts representative micrographs of the DCN within the cerebellum, stained with Calbindin (green) and Hoechst (blue). The first column shows low magnification images of the DCN and surrounding areas, with labels for the Molecular layer (Mo), Purkinje cell layer (Pc), Granular layer (Gr), and deep cerebellar nuclei (DCN). Scale bar 100 µm. A red box indicates the area of higher magnification within the DCN shown in the second column. The micrographs in the second column demonstrate a clear reduction of calbindin-positive spheroids with both AAV9-CBh-hAP4B1 and AAV9-SYN-hAP4B1 vectors. The third column shows nuclear staining with Hoechst, and the fourth column presents a merge of Calbindin and Hoechst stains. Scale bar 50 µm. The bar graph reveals a larger number of spheroids in untreated and control-treated Ap4b1 KO mice compared to no spheroids in the WT mice. These spheroids are significantly reduced with both CBh ($p < 0.0001$) and SYN ($p < 0.0001$) treatment vectors 9 months following treatment. Data are presented as mean ± standard error of the mean (SEM), with $n = 3$. The data were analysed using one-way ANOVA followed by Dunnett's multiple comparisons test. Stars indicate $p \leq 0.0001$ (****).

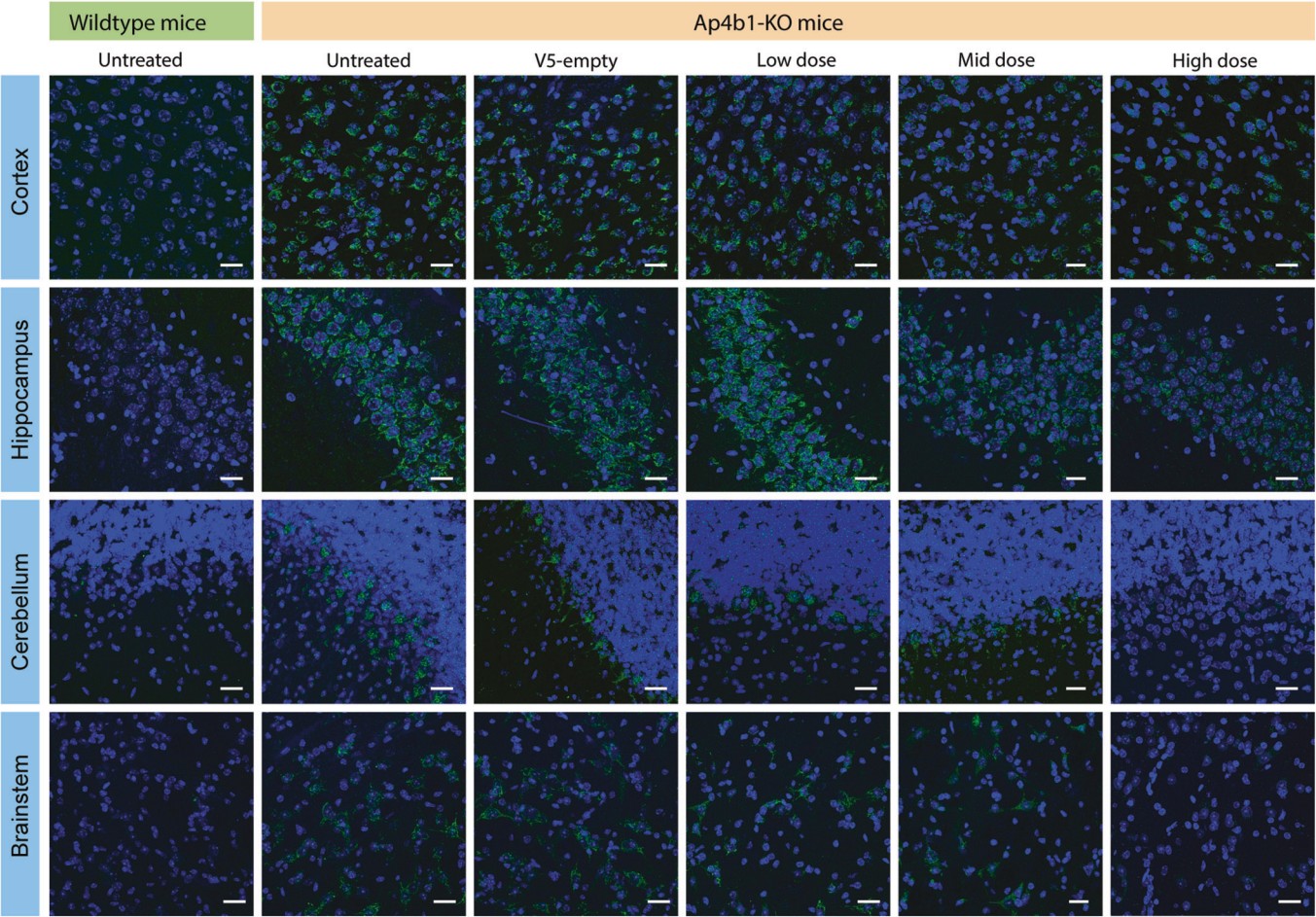

**Figure EV3. Corresponding Hoechst staining to Fig. 6A to show the nuclear organisation within the separate brain regions.**

Atg9a staining (green), Hoechst staining (blue). Representative micrographs showing a dose-dependent reduction of Atg9a perinuclear accumulation in the brain regions: cortex, hippocampus, cerebellum, brainstem. Scale bar 20 μm.

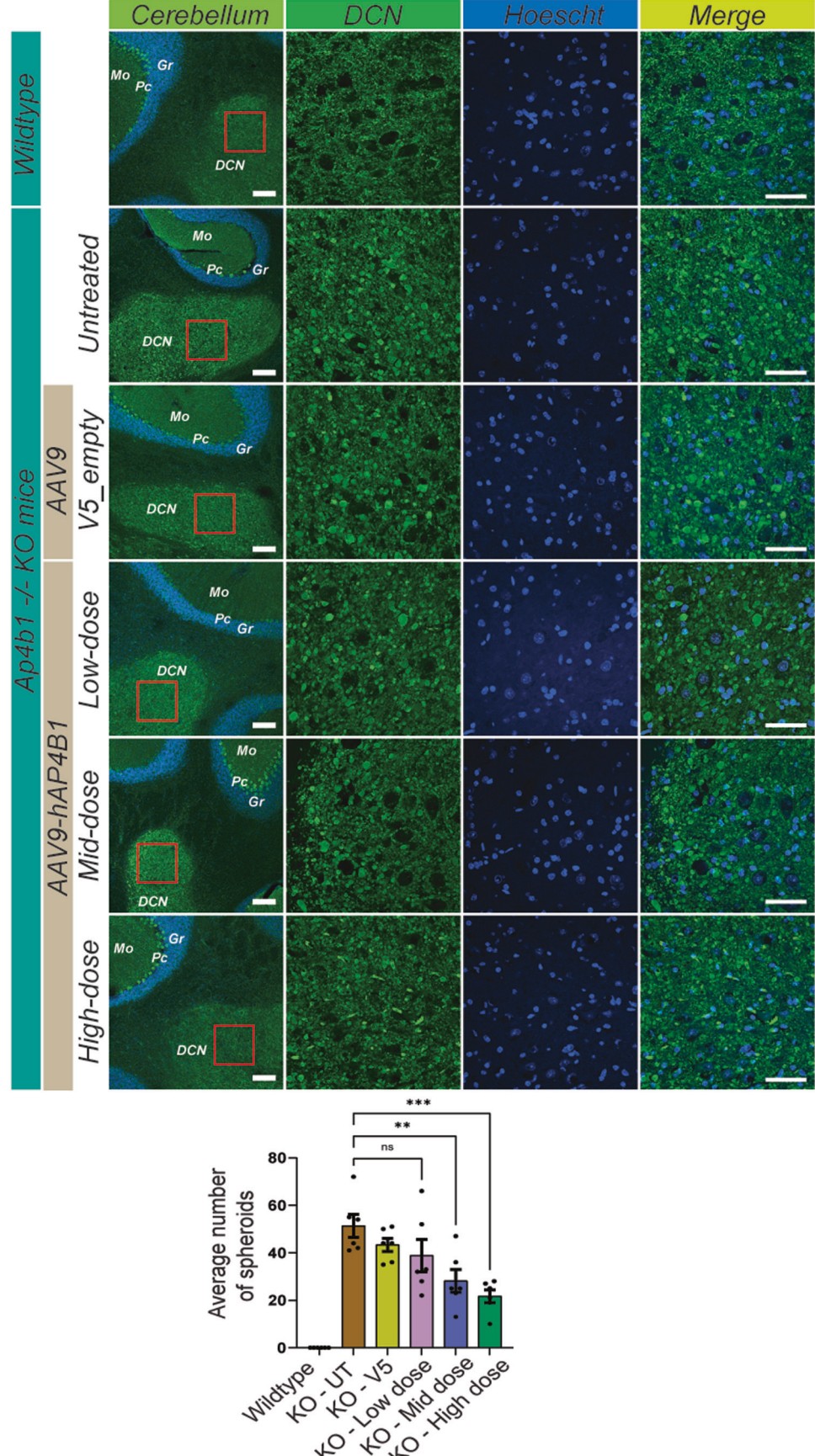

◀ **Figure EV4.   Adult ICM treatment with AAV9-CBh-hAP4B1 significantly reduces Calbindin-positive spheroids in the DCN of Ap4b1-KO mice 9 months following treatment.**

The image panel depicts representative micrographs of the DCN within the cerebellum, stained with Calbindin (green) and Hoechst (blue). The first column shows low-magnification images of the DCN and surrounding areas, with labels for the Molecular layer (Mo), Purkinje cell layer (Pc), Granular layer (Gr), and deep cerebellar nuclei (DCN). Scale bar 100 μm. A red box indicates the area of higher magnification within the DCN shown in the second column. The micrographs in the second column demonstrate a clear dose-dependent reduction of calbindin-positive spheroids with increasing dose of the therapeutic vector (CBh-hAP4B1). The third column shows nuclear staining with Hoechst, and the fourth column presents a merge of Calbindin and Hoechst stains. Scale bar 50 μm. The bar graph reveals spheroids are reduced on a dose-dependent basis with mid- and high-dose significantly reducing the presence of spheroids ($p = 0.0066$ and $p = 0.0004$, respectively). Data are presented as mean ± standard error of the mean (SEM), with $n = 6$. The data were analysed using one-way ANOVA followed by Dunnett's multiple comparisons test. Stars indicate $p \leq 0.001$ (***), $p \leq 0.01$ (**), ns = not significant.

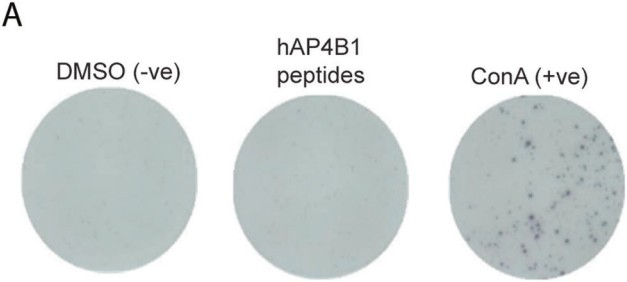

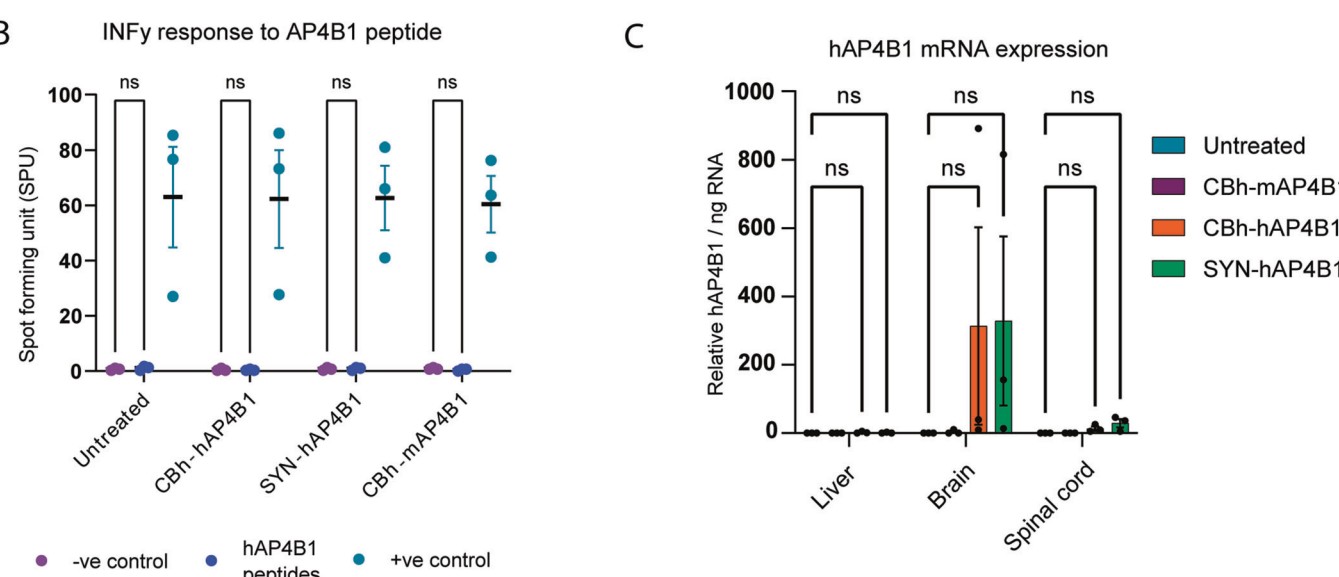

**Figure EV5.  In vivo safety study in WT mice through ELISpot assay, AAV9/hAP4B1 treatment generated no B cell response to the hAP4B1 peptides.**

Vectors CBh-hAP4B1 and SYN-hAP4B1 were tested, alongside the vector containing the mouse *Ap4b1* gene (mAP4B1). Splenocytes that were prepared from WT mice following were assessed through an ELISpot assay for IFN-γ responses to AP4B1 peptides. (**A**) Representative images of the spot forming detection revealed following the exposure to the negative control (DMSO), the hAP4B1 peptides and the positive control (Concanavalin A). (**B**) Treated mice did not show any inflammatory response to the peptides. (**C**) RT-qPCR of total RNA extracted demonstrated elevated hAP4B1 mRNA expression in the brain, liver and spinal cord of treated WT mice with lower expression within the liver. Data are presented as mean ± SEM, $n = 3$ per group. Data are analysed by a two-way ANOVA with Tukey's post hoc multiple comparison test. ns = not significant.

