## [Peer Review File · EMBO Molecular Medicine]

Pre-clinical Development Of AP4B1 Gene Replacement Therapy For Hereditary Spastic Paraplegia Type 47

Jessica Wiseman, Joseph Scarrott, Joao Alves-Cruzeiro, Afshin Saffari, Cedric Boger, Evangelia Karyka, Emily Dawes, Alexandra Davies, Paolo Marchi, Emily Graves, Fiona Fernandes, Zih-Liang Yang, Ian Coldicott, Jennifer Hirst, Christopher Webster, J Robin Highley, Neil Hackett, Adrienn Angyal, Thushan Silva, Adrian Higginbottom, Pamela Shaw, Laura Ferraiuolo, Darius Ebrahimi-Fakhari, and Mimoun Azzouz

Corresponding author: Mimoun Azzouz (m.azzouz@sheffield.ac.uk)

Review Timeline:

Submission Date:	15th Mar 24
Editorial Decision:	10th Apr 24
Revision Received:	9th Jul 24
Editorial Decision:	16th Aug 24
Revision Received:	17th Sep 24
Accepted:	19th Sep 24

Editor: Zeljko Durdevic

Transaction Report:

10th Apr 2024

Dear Prof. Azzouz,

Thank you for the submission of your manuscript to EMBO Molecular Medicine. We have now received feedback from the two reviewers who agreed to evaluate your manuscript. As you will see from the reports pasted below, both referees recognize potential interest of the study but also raise important concerns that should be addressed in a major revision. If you would like to discuss further the points raised by the referees, I am available to do so via email or video. Let me know if you are interested in this option.

We would welcome the submission of a revised version within three months for further consideration. Please let us know if you require longer to complete the revision.

I look forward to seeing a revised form of your manuscript as soon as possible.

I look forward to receiving your revised manuscript.

Yours sincerely,

Zeljko Durdevic

We require:

- 1) A .docx formatted version of the manuscript text (including legends for main figures, EV figures and tables). Please make sure that the changes are highlighted to be clearly visible.
- 2) Individual production quality figure files as .eps, .tif, .jpg (one file per figure). For guidance, download the 'Figure Guide PDF': (<https://www.embopress.org/page/journal/17574684/authorguide#figureformat>).
- 3) A .docx formatted letter INCLUDING the reviewers' reports and your detailed point-by-point responses to their comments. As part of the EMBO Press transparent editorial process, the point-by-point response is part of the Review Process File (RPF), which will be published alongside your paper.
- 4) A complete author checklist, which you can download from our author guidelines (<https://www.embopress.org/page/journal/17574684/authorguide#submissionofrevisions>). Please insert information in the checklist that is also reflected in the manuscript. The completed author checklist will also be part of the RPF.
- 5) Please note that all corresponding authors are required to supply an ORCID ID for their name upon submission of a revised manuscript.
- 6) It is mandatory to include a 'Data Availability' section after the Materials and Methods. Before submitting your revision, primary

datasets produced in this study need to be deposited in an appropriate public database, and the accession numbers and database listed under 'Data Availability'. Please remember to provide a reviewer password if the datasets are not yet public (see <https://www.embopress.org/page/journal/17574684/authorguide#dataavailability>).

13) Author contributions: You will be asked to provide CRediT (Contributor Role Taxonomy) terms in the submission system. These replace a narrative author contribution section in the manuscript.

14) A Conflict of Interest statement should be provided in the main text.

Please also suggest a striking image or visual abstract to illustrate your article as a PNG file 550 px wide x 300-800 px high.

**** Reviewer's comments ****

Referee #1 (Comments on Novelty/Model System for Author):

The authors have completed an extensive set of experiments in vitro using AP4B1 knockout (KO) i) cell lines and ii) patient-derived cells.

They have completed various in vivo investigations using a mouse model of KO AP4B1. They performed experiments in neonatal and adult mice and included both male and female genders in their experiments. Biodistribution of the AP4B1 expression following vector administration was first investigated before initiation of therapy experiments.

The study also reports evaluation of the gene therapy efficacy using various tests including cellular, anatomical and behavioural].

Finally, the team has completed toxicity studies both in mice and non-human primates (NHP).

They also carried out a thorough statistical analysis of the data.

Referee #1 (Remarks for Author):

The study by Wiseman et al. reports an investigation of gene therapy for the restoration of expression of the wild type AP4B1 gene in spastic paraplegia 47. Mutation of both alleles of the AP4B1 gene has been associated with the disease. They have carried out a great deal of experiments in vitro using AP4B1 knockout (KO) cell lines and patient-derived primary cells. Then they completed extensive in vivo investigation using a mouse model of KO AP4B1 in neonatal and adult mice both in male and female genders, performing biodistribution of the AP4B1 expression following vector administration via the intracisterna magna (ICM) route. This was followed by assessment of the therapeutic efficacy of this gene therapy approach using various cellular anatomical and behavioural tests. Moreover, they have completed toxicity studies in mice then in large animals using non-human primates (NHP).

This is a comprehensive and systematic study, clearly written with strong rationale. The outcome has definitely strengthened the position of the team for a clinical trial in patients in the near future, and they already have entered into discussion with the regulator MHRA. This manuscript reports a large set of data, includes many figures, and provides extra findings on toxicity both in mice and NHP, paving the way toward clinical testing. However, before publication of the study the manuscript should be improved. I have the following points:

1- What is the rationale of using the chicken beta-actin hybrid promoter? There also are new versions of AAV9 for CNS targeting with improved BBB-crossing following IV delivery, can the authors provide justification for the choice of the conventional AAV vector in this study.

2- As mentioned in the manuscript, cell toxicity occurs as the lentiviral vector MOI increases. How can this affect the accumulation of the ATG9A in the Golgi? Did the authors normalise the data to protein level or cell count?

3- In biodistribution studies the liver and the heart were chosen as the vital organs to check, I wonder why not other organs were included such as the kidneys, spleen and lungs.

4- Figure 3 shows expression induced by the ICM delivery route in the cerebellum, cerebrum and spinal cord, but the IV route had produced a low expression of APA4B1. Can the authors provide an explanations since AAV9 can cross the BBB and target brain tissues, in particular via the facial vein injection. Figures 3G and 3I show WB of AP4E1 levels in the cerebrum and spinal cord but the cerebellum is not shown.

5- In figure 3A, I would have expected the KO V5-empty IV results in similar vector genome copies in the liver compared to the KO Cbh-hAP4B1 IV, the levels of vector copies are similar in the heart between the two vectors. Can the authors provide an explanation? I would have expected the same for all other tissues since the capsid allowing viral vector entry is the same for the two vectors.]

6- In figure 4, the mice were injected with a higher dose than the one investigated in the biodistribution study, can the authors provide some rationale for this? Why the rotarod testing data are shown at 6 months and didn't include 3 and 9 months? The SYN promoter vector was used in the behavioural testing in figure, but was this included in other experiments in vitro and in vivo side-by-side with the hybrid promoter?

7- In biodistribution studies using the viral genome copies, it is not clear to me why other vectors don't show similar distribution when given via similar routes in 3 A and 5E, since they all have the same capsid. Is this associated with the size of vectors?

8- What is the rationale for choosing the vector doses in the dose escalation experiment in figure 6?

9- Intravenous injection, IV, of neonatal mice was performed using the facial vein. The tail vein injection is less invasive and why it has not been used.

10- Figure 6C, no expression in the liver after a mild-dose injection while a low dose does, can this be explained?

11- Figure 7A, 68.36% rescue with an MOI 1.6×10^7 . Did you do any quantification?

12- In figure 7B, can the cell viability affect the data in figure 7A?

13- In the safety study 1, I would like to see some rationale for the use of 1×10^{11} vg/mouse?

14- There is no data on repeated dosing in mice and large animals, can the authors discuss this in their discussion section.

15- In figure 1E, I understand that the authors have normalised the results of AP4E1 expression to WT, but did they do this after each WT and conditions have been normalised to their own GAPDH internal control? This should apply to all the other figures where data were normalised to WT.

Referee #2 (Comments on Novelty/Model System for Author):

Technical quality is medium. Generally the quality of the figures is good. But statistics are missing in some figures and bar graphs without individual data points are often used.

The novelty is medium. Gene therapy using AAV vectors administered to the brain has been done for a growing number of neurological diseases. NfL is a known biomarker for neurological diseases.

Medical impact is high. There is currently no effective treatment for this condition and there is a high unmet need for a new therapy.

The mouse model being used to test the therapy is far from ideal as it only partially recapitulates the symptoms of HSP47. However, the other models of HSPs are also not great either and this is as good as it is going to get with what is currently available.

Referee #2 (Remarks for Author):

The study evaluates the pre-clinical efficacy of the AAV9-mediated gene therapy for Hereditary Spastic Paraplegia Type 47 (HSP47). This condition has an unmet need for the development of novel therapies. The authors conduct studies using lentiviral and AAV9 vectors in cell culture and a new mouse model of HSP47. They also demonstrate that the NfL biomarker is modulated in response to therapy. After demonstrating therapeutic efficacy, they move along the translational pipeline developing a potency assay and GLP tox studies in NHPs.

Major Points:

The mouse model has been published by the authors and it partially recapitulates the phenotype of HSP47. In terms of neuropathology the authors have looked at some abnormalities reported such as ventricular megaly and thinning of the CC in this study. However, they also reported the presence of calbindin positive spheroids in the DCN. Given the lack of multiple

markers of disease in this model, these should be assessed in the brain section and quantified to provide a further readout that they already know about.

Similarly in the behavioural tests, the authors have published hyperactivity in an open field test in this mouse model. The effect of therapy on this should be reported on to further the battery of tests.

The clinical description of the disease needs improving in the Introduction eg. incidence, what is the average age of onset, survival, most frequent mutations, co-morbidities etc.

The Introduction describes how HSPs are characterised by degeneration of the corticospinal tract. Have the authors looked in the spinal cord for pathology and subsequent therapeutic efficacy? This should be looked into.

The Authors make the interesting observation that later admin of gene therapy to the older mice results in loss of efficacy in treating enlarged ventricles and thinning of the white tract. It also reduces efficacy in behavioural tests. This is to be expected but what is the consequence of this finding in terms of translation? What age would the Authors be looking at intervening in patients and how early are they diagnosed. Is this compatible with a diagnosis that is beyond the neonatal period when efficacy is best? This needs to be addressed in the Discussion.

The Authors conduct a dose finding study. What has this told them about the dose required in patients at the proposed age of intervention? This should be commented on in the Discussion.

Which neural cell types are expressing the transgene under the ChB promoter?

Minor Points

Doses administered in the study are often referred to as VG. Please state dosing as vg/kg which is more informative.

Please check materials and methods for typos.

Typo bottom of page 12 'develop' to 'developed'.

Why is the bio distribution between KO-Cbh-hAP4B1 and KO-Syn-hAP4B1 different in Figure 5E. The promoter should not make a difference? There is no statistical analysis of this data to show if it is significant or not. Please comment?

Figs 3A-F bar graphs should also show individual data points for transparency. Also not statistical analysis. Same for Figure 6C.

We would like to thank the reviewers for their constructive feedback. We have addressed each point and provided clear explanations where needed.

Reviewer 1

1- What is the rationale of using the chicken beta-actin hybrid promoter? There also are new versions of AAV9 for CNS targeting with improved BBB-crossing following IV delivery, can the authors provide justification for the choice of the conventional AAV vector in this study.

The rationale behind utilising the CBh promoter in our study is due to its ability to drive robust and widespread transgene expression in the CNS. CBh has been demonstrated to mediate efficient and sustained transgene expression in various cell types, including neurons and glia (Gray et al. 2011, DOI: 10.1089/hum.2010.245). Another advantage of using CBh promoter is because of its mediation of strong transgene expression lower doses of the AAV9 therapy should be required in comparison to other weaker promoters.

This has been clarified in the results section of the revised manuscript - page 6.

Rationale for AAV9: We agree that emerging new AAV9 capsids led to improved BBB crossing and CNS targeting. However, most of these improvements were limited to rodents and gene transfer is not optimal in large animals in particular non-human primate, reducing our confidence (at least at this stage) in using these new capsids for future human applications. Of course, with further investigations, these novel capsids would be a great added value for future gene therapy applications. Another important point is that there are multiple ongoing trials using AAV9 and AAV9-based products that were approved by regulators including FDA and EMA (e.g. Zolgensma). In addition, AAV9 has well established GMP manufacturing processes which is essential for scale up of clinical grade vectors for human trials and commercialisation. Taken together this evidence gave us stronger confidence in the safety and efficacy of our AAV9-based approach which would accelerate regulatory approval to enter clinical trials in SPG47 patients. All this has also been reflected in our Pre-IND correspondence and the positive response received from FDA.

This comment is also addressed in the introduction on page 5 of the revised manuscript.

2- As mentioned in the manuscript, cell toxicity occurs as the lentiviral vector MOI increases. How can this affect the accumulation of the ATG9A in the Golgi? Did the authors normalise the data to protein level or cell count?

We thank the reviewer for this question. This assay quantifies the ATG9A ratio, which is the amount of ATG9A fluorescence overlapping with a trans-Golgi network (TGN) marker versus the cytoplasm outside the TGN at the level of each individual cell (Saffari et al. Nature Communications 2024 PMID: 38233389; Chen et al. J Clin Invest. PMID: 36951961; Ebrahimi-Fakhari et al. Brain Commun. 2021 PMID: 34729478; Behne et al. Hum Mol Genet. 2020 PMID: 31915823). This is quantified in hundreds of cells per well in 96-well plates and thousands of cells per experiment. The ATG9A ratio is a robust indicator of AP-4 function, offering an excellent quality metric for a high-throughput assay (Saffari et al. Nature Communications 2024 PMID: 38233389). Therefore, the cell count generally has no impact on the ATG9A ratio.

This comment has been clarified in the revised manuscript, page 33.

3- In biodistribution studies the liver and the heart were chosen as the vital organs to check, I wonder why not other organs were included such as the kidneys, spleen and lungs.

We thank the reviewer for this comment. AAV9 mediates high gene transfer efficiency to the heart and liver when administered to mice. These two organs were chosen for our studies in mice because of their relevance in relation to previously reported safety concerns linked to AAV. Nevertheless, in our non-human primate GLP regulatory safety study, we investigated all organs to ensure a comprehensive understanding given that this model would closely mimic human distribution patterns (please see Table 9, Manuscript Appendix (Supplemental Information), page 37).

We have addressed this point in the mouse biodistribution results section, page 8.

4- Figure 3 shows expression induced by the ICM delivery route in the cerebellum, cerebrum and spinal cord, but the IV route had produced a low expression of AP4B1. Can the authors provide an explanation since AAV9 can cross the BBB and target brain tissues, in particular via the facial vein injection.

While the vector can penetrate the BBB low transgene expression is observed with IV administration, because efficient transduction in brain regions and the spinal cord generally require significantly higher AAV doses when using this route of delivery. However, increasing the viral dose for IV delivery raises the risk of high viral load in peripheral organs like the heart and liver, which are more readily transduced when the vector is administered intravenously and could cause adverse effects.

This comment has been addressed in the revised discussion on page 20.

Figures 3G and 3I show WB of AP4E1 levels in the cerebrum and spinal cord but the cerebellum is not shown.

Thank you for highlighting this point. Cerebellum data has been added to Figure 3 (Figure 3K & L) and stats have been changed to significance between ICM and IV.

Cerebellum western blot data is added to Figure 3. Text referring to this data is included in page 9 and legend to Figure 3, page 50.

5- In figure 3A, I would have expected the KO V5-empty IV results in similar vector genome copies in the liver compared to the KO Cbh-hAP4B1 IV, the levels of vector copies are similar in the heart between the two vectors. Can the authors provide an explanation? I would have expected the same for all other tissues since the capsid allowing viral vector entry is the same for the two vectors.]

We thank the referee for highlighting this point. There are two potential explanations for these observations: (i) Firstly, capsid conformation may vary depending on the size of the packaging sequence. The sequence for CBh-hAP4B1 is larger than the V5-empty vector. Therefore, each vector capsid may have differing conformations, and this has the potential to influence the interaction between viral receptors and cell surface receptors, and thus cellular uptake. In the viral distribution data presented in Figure 3A, the CBh-hAP4B1 vector consistently exhibits marginally higher levels than the V5-empty vector. Notably in the liver, the CBh-hAP4B1 vector shows a significantly greater presence of vector compared to the V5-empty vector, suggesting that liver cells may exhibit increased susceptibility to binding and uptake when interacting with the conformation of the larger packaging vector. The reason for observing higher levels of the CBh-hAP4B1 vector in the liver when delivered IV compared to ICM remains unclear. However, when delivered to the CSF, potential routes for the viral vector to reach the liver include both the vascular and lymphatic systems, which could influence vector transduction. (ii) Secondly, there is potential for minor experimental variability, which may arise from factors such as variations in the injection procedure and differences in immune clearance of the vector among mice. (iii) it is known that AAV packaging is reduced when the transgene sequence is short. For this reason, it is essential to add a stuffer sequence to enhance packaging and viral titre.

The above explanations have been discussed in pages 20 of the revised manuscript

6- In figure 4, the mice were injected with a higher dose than the one investigated in the biodistribution study, can the authors provide some rationale for this? Why the rotarod testing data are shown at 6 months and didn't include 3 and 9 months? The SYN promoter vector was used in the behavioural testing in figure, but was this included in other experiments in vitro and in vivo side-by-side with the hybrid promoter?

We thank the reviewer for highlighting this point and apologies for the confusion. This misunderstanding is caused by the fact that the biodistribution study dose is written in the methods as 4×10^{10} vg/gram and neonatal long-term study was 1×10^{11} vg/mouse. P1-3 mice on average are around 2.5 grams so the two refers to the same dose. To avoid this confusion, dosage has been stated as **vg/kg** throughout the text in the revised manuscript.

Why the rotarod testing data are shown at 6 months and didn't include 3 and 9 months?

Rotarod performance was analysed only at 4 and 6 months. As phenotypic differences between WT and KO mice were not observed at 4 months, we did not include this data in our manuscript.

This comment has now been clarified in the revised manuscript page 10

The SYN promoter vector was used in the behavioural testing in figure, but was this included in other experiments in vitro and in vivo side-by-side with the hybrid promoter?

The SYN promoter was not initially included in other experiments, except for the long-term neonatal study and the WT safety/ biodistribution study. It was later added as a potential contingency plan in case the long-term mouse studies showed that the CBh vector resulted in adverse off-target effects. Unlike the CBh promoter, which is strong at transducing peripheral organs, the SYN promoter was considered as a more neuronal specific promoter and said to reduce transduction of peripheral organs. However, the CBh promoter did not show any adverse effects both in mice and NHP, displayed a stronger transduction of the brain regions and showed a slightly improved rescue of the disease phenotypes.

This comment has been addressed in the discussion page 23 of the revised manuscript.

7- In biodistribution studies using the viral genome copies, it is not clear to me why other vectors don't show similar distribution when given via similar routes in 3 A and 5E, since they all have the same capsid. Is this associated with the size of vectors?

That's correct. As stated in our answer to comment #5 above, the size of the vector could be one of the key factors to explain the observed variability in the biodistribution studies. While the vectors may share the same capsid, they carry different packaging plasmid size. For example, the V5 vector is considered empty which led to significant difference in size when compared to CBh-hAP4B1 vector. This could influence the vector packaging, the yield of the produced vector, and how the vectors interact with host cells, their ability to transduce different tissues, and their overall biodistribution patterns. Further explanation is provided under comment #5 above. Please note that this has been discussed in page 20 of the revised manuscript

8- What is the rationale for choosing the vector doses in the dose escalation experiment in figure 6?

The selection of doses used in dose escalation study summarised in Figure 6 was based on: 1) our previous studies using the therapeutic vector and AAV9 serotype in general; 2) Our extensive gene therapy studies using ICM as route of delivery for AAV9; 3) studies from the literature, for example recent gene therapy studies targeting the AP4 complex, e.g. SPG50 study: Chen et al., 2023: PMID: 36951961

CBh is recognised as a more potent promoter compared to the Jet promoter used in the SPG50 study. Therefore, we opted to start our study with their mid (dose 1) and low (dose 2) doses, along with an additional dose lower (dose 3) than the initial ones. However, two weeks after administering dose 1 mice exhibited signs of illness and had to be culled, whereas those in the other two groups did not. Consequently, this dose was discontinued, and another low dose was introduced into the study.

SPG50 dose response:	Respective dose - SPG47 dose response
Chen et al., 2023: PMID: 36951961	
High dose – 2×10^{10} vg/g	-----
Mid dose – 1×10^{10} vg/g	Tox dose (2×10^{11} vg/mouse) - 1×10^{10} vg/g
Low dose – 5×10^9 vg/g	High dose (1×10^{11} vg/mouse) - 5×10^9 vg/g
-----	Mid dose (8×10^{10} vg/mouse) - 4×10^9 vg/g
-----	Low dose (6×10^{10} vg/mouse) - 3×10^9 vg/g

The rationale for dose selection has been added to the revised manuscript, pages 34.

9- Intravenous injection, IV, of neonatal mice was performed using the facial vein. The tail vein injection is less invasive and why it has not been used.

We agree that tail vein is widely used for intravenous delivery in mice. However, this is mainly done in adult mice. There are multiple reasons for the selection of facial vein in neonatal mice: 1) The facial vein in neonatal mice is relatively large and easily accessible through the skin compared to the tail vein. 2) Facial vein in neonatal mice allows for more precise control over the injection volume and accuracy. 3) Tail vein can be prone to injury in neonatal mice due to delicate tails.

This comment has been clarified in the methods section page 35.

10- Figure 6C, no expression in the liver after a mild-dose injection while a low dose does, can this be explained?

Thank you for bringing this to our attention. We observed that out of the nine treated mice liver samples processed for RNA, only two displayed significantly elevated levels of the transgene within the liver. Specifically, these were one low-dose mouse (92 ng/total RNA) and one high-dose mouse (126 ng/total RNA). In contrast, all other mice showed transgene levels of less than 5 ng/total RNA. This observation suggests that in these two animals, the ICM injection may have resulted in considerable leakage through BBB to reach the liver. This could imply one of two possibilities: either the injection was partly administered outside the CSF, or these mice have an inherently leaky BBB, allowing transgene leakage into the bloodstream and subsequently the liver. Due to the high levels in one low-dose and one high-dose mouse this has skewed the graph to imply that low and high dose treatments result in high liver expression. However, there is no statistical significance between these groups due to the high error bars.

11- Figure 7A, 68.36% rescue with an MOI 1.6x10⁷. Did you do any quantification?

The quantification was described in the text according to our established normal assay (reference: <https://academic.oup.com/braincomms/article/3/4/fcab221/6375442>).

68.36% rescue means that the ATG9A distribution of treated AP4B1-KO cells is, on average, 68.36% closer to WT cells than before. If the ATG9A ratio was 2.0 before treatment, is 1.0 in WT cells, and after the treatment 1.4, this equals a 60% rescue. This is calculated based on the delta between the controls. Since the ATG9A ratios in KO and WT controls can vary between wells, repetitions, staining quality, antibody #lot, etc., this normalization using the delta between both KO and WT proved to be a robust and efficient measurement of ATG9A translocation and rescue.

12- In figure 7B, can the cell viability affect the data in figure 7A?

We thank the reviewer for this question. Please also refer to the answer to question 2. This assay quantifies the ATG9A ratio, which is the amount of ATG9A fluorescence overlapping with a trans-Golgi network (TGN) marker versus the cytoplasm outside the TGN at the level of each individual cell (Saffari et al. Nature Communications 2024 PMID: 38233389; Chen et al. J Clin Invest. PMID: 36951961; Ebrahimi-Fakhari et al. Brain Commun. 2021 PMID: 34729478; Behne et al. Hum Mol Genet. 2020 PMID: 31915823). This is quantified in hundreds of cells per well in 96-well plates and thousands of cells per experiment. The ATG9A ratio is a robust indicator of AP-4 function, offering an excellent quality metric for a high-throughput assay (Saffari et al. Nature Communications 2024 PMID: 38233389). Therefore, the cell count generally has no impact on the ATG9A ratio. We choose a conservative cut-off of 3 standard deviations in mean cell counts per well to indicate toxicity.

This comment has been clarified in the revised manuscript, page 33.

13- In the safety study 1, I would like to see some rationale for the use of 1x10¹¹ vg/mouse?

Safety study 1 was carried out on neonatal mouse pups like the initial biodistribution study. There we used the same dose – 1×10^{11} vg/mouse = 4×10^{13} vg/kg. This is reflected by the maximum volume/dose allowed for CSF delivery in neonatal mice according to the Animal (Scientific Procedures) Act 1986.

14- There is no data on repeated dosing in mice and large animals, can the authors discuss this in their discussion section.

AAV9 gene therapy in this case does not require repeated dosing, a single administration is sufficient to achieve long-lasting therapeutic effect. Once delivered to the target cells the AAV9 vector remains episomal and persists in the cells, particularly in neurons. This leads to a long term and stable expression of the therapeutic transgene. We have provided evidence for the expression of the transgene 9 months post administration and the maintenance of the therapeutic effects. Repeated dosing is more applicable when target cells or tissues have a high turnover of cells and undergo continuous regeneration (this is not the case for SPG47). As AAV9 is non-integrational, thus the therapeutic gene will not be passed on to daughter cells, repeated dosing would be necessary to maintain therapeutic effect in tissues with high cellular turnover. Single administration of AAV9 gene replacement therapies have been proven effective for multiple CNS diseases. All this together supports our rationale for not investigating repeated dosing for this genetic disease.

This has been added to discussion on page 21.

15- In figure 1E, I understand that the authors have normalised the results of AP4E1 expression to WT, but did they do this after each WT and conditions have been normalised to their own GAPDH internal control? This should apply to all the other figures where data were normalised to WT.

Quantification of protein signals was carried out through image studio. AP4E1 protein signal was normalised to the respective GAPDH protein signal to account for loading differences, then each sample was normalised to the WT of that biological repeat. This means AP4E1 expression for each biological repeat of the WT samples will equal 1.

This comment has been clarified in methods section page 30.

Reviewer 2

We thank the reviewer for the constructive feedback aimed at improving our manuscript. We've addressed each point and provided clear explanations where needed.

#1- The mouse model has been published by the authors and it partially recapitulates the phenotype of HSP47. In terms of neuropathology the authors have looked at some abnormalities reported such as ventriculomegaly and thinning of the CC in this study. However, they also reported the presence of calbindin positive spheroids in the DCN. Given the lack of multiple markers of disease in this model, these should be assessed in the brain section and quantified to provide a further readout that they already know about.

We thank the reviewer for this comment. We have now assessed calbindin-positive spheroids in brain sections from both neonatal and adult dose response mouse studies, and the results were included in new Figures EV2 and EV4, respectively. We are pleased to report that the treatment in neonatal treated mice led to an almost 100% reduction of the presence of calbindin-positive spheroids in the DCN. Additionally, we observed a dose-dependent decrease in spheroids within the DNC of adult treated mice.

Description of these additional data is included in page 12 (Figure EV2) and page 14 (Figure EV4) of the revised manuscript and newly added Figures EV2 and EV4. Figure legends were added to pages 54-55.

Staining methods and analysis methods have been added to page 39 and page 40 respectively of the revised manuscript

#2- Similarly in the behavioural tests, the authors have published hyperactivity in an open field test in this mouse model. The effect of therapy on this should be reported on to further the battery of tests.

We would like to confirm that an open field test was also performed in the current therapy studies. Please see the data added to the revised manuscript as Appendix Figure S1. This data describes our findings using the automated open field analysis platform. Although the Ap4b1 KO mice exhibit a slight tendency to explore more than their WT counterparts, this difference is not statistically significant.

Although open field was carried out in the mouse model characterisation study we observed contrasting results. The reason for this is most likely due to the difference in test modality between the open field assessments carried out in the two studies. In the characterisation paper the open field was assessed through **manually counting** the number of grid line crossings a mouse crossed within a box over a 10-minute period. Whereas in this current study we used an **automated open field analysis platform** which tracked the movement of the mice over 5 minutes through a camera and recognition software which then gave a value of total distance travelled.

This point has been addressed in results section of the revised manuscript on pages 10.

#3- The clinical description of the disease needs improving in the Introduction e.g., incidence, what is the average age of onset, survival, most frequent mutations, co-morbidities etc.

We thank the reviewer for this suggestion. Unfortunately, reliable data on prevalence of SPG47 are currently not available, and estimates based on allele-frequencies in certain populations are likely not representative. Over the last 4 years, the International Registry and Natural History Study for Early-Onset Hereditary Spastic Paraplegia (NCT04712812) has enrolled just over 250 individuals with AP-4-related HSP, including about 100 with SPG47. The age at onset has been documented through a cross-sectional analysis of the first 156 cases of AP-4-HSP and is in infancy (Ebrahimi-Fakhari et al. Brain 2020). This information has been added to the revised version of the introduction section which also cites appropriate references on the clinical spectrum of SPG47.

This comment has been addressed in the introduction on Page 3.

#4-The Introduction describes how HSPs are characterised by degeneration of the corticospinal tract. Have the authors looked in the spinal cord for pathology and subsequent therapeutic efficacy? This should be looked into

Following the very relevant recommendation from this reviewer and prior assessing therapeutic efficacy, we decided to first examine for potential spinal cord pathology in the Ap4b1 KO mice. We conducted a histopathological analysis of spinal cords from Ap4b1 KO mice using an anti-Tau antibody (AT8) and a neurofilament antibody (SMI31) to detect Tau deposition and axonal swelling, respectively. We would expect to see Tau deposition in either the white or the gray matter whereas axonal swellings would be observed in the white matter. Analysis of the spinal cord sections did not detect any differences in the white matter of DAB/SMI31-stained cervical spinal cords or in the gray and white matter of DAB/AT8-stained cervical spinal cords. Spinal cord pathology seems therefore not to be a good readout for therapy efficacy testing in our mouse model. For this reason, we did not perform experiments to investigate the impact of our gene therapy approach on the spinal cord of this particular mouse model.

Similar to the other AP-4 deficient mouse model characterised by Chen et al. (2023), AP4b1 KO mice have mild phenotype but still display some very important features of the disease: e.g. Lateral ventricle enlargement, corpus callosum thinning, ATG9A mislocalisation, calbindin-positive spheroids, high NFL levels, rotarod and clasping motor phenotypes. Given the mild motor phenotypes in this mouse model, we did not anticipate major spinal pathology. Although hereditary spastic paraplegias in humans are characterized by corticospinal tract degeneration, this pathology does not directly translate to mice and only a few SPG mouse models have shown spinal cord degeneration.

#5-The Authors make the interesting observation that later admin of gene therapy to the older mice results in loss of efficacy in treating enlarged ventricles and thinning of the white tract. It also reduces efficacy in behavioural tests. This is to be expected but what is the consequence of this finding in terms of translation? What age would the Authors be looking at intervening in patients and how early are they diagnosed. Is this compatible with a diagnosis that is beyond the neonatal period when efficacy is best? This needs to be addressed in the Discussion

We agree, this is a very relevant finding for informing the design of planned future clinical trials in SPG47 patients. Our aim is to treat patients as early as possible. Since 2015 patients have been diagnosed at a younger age due to increased availability of exome sequencing (Ebrahimi-Fakhari et al., 2020). Diagnosis of SPG47 is now possible as early as 12 months (Ebrahimi-Fakhari et al., 2018b). We also expect diagnosis to evolve even further because of the importance of catching up patients with rare diseases at younger ages.

We are pleased to report that our pre-clinical package including our proof-of-concept in vivo data were submitted to FDA as Pre-IND and is currently forming a basis for IND submission. The current clinical trial design proposes to treat 1-5 years old patients in first human clinical trials.

This has been discussed in page 27 together with the comment #6 below.

#6- The Authors conduct a dose finding study. What has this told them about the dose required in patients at the proposed age of intervention? This should be commented on in the Discussion.

Following submission of our Pre-IND, FDA guidance to us was to scale up dosing by cerebrospinal fluid (CSF) volume. We now have a draft clinical design including dosing for clinical human trials in SPG47 patients. The clinical design and IND are being finalised for submission to FDA by September 2024. The table below summarises how we calculated the planned human trial dose based on our proof-of-concept mouse studies, long term safety study in mice, GLP regulatory NHP toxicology study and literature.

Organism	Route	Total dose (genome copies)	CSF volume (ml) ¹	Vector genomes (vg)/ml
Neonatal mice ² Study # ZLY-02	ICM	1×10^{11}	0.002	5×10^{13}
Adult mouse dosing study Study # JPW-01	ICM	0.4×10^{11} to 2×10^{11}	0.04	1 to 5×10^{12}
Mouse long term toxicology Study # JAX-131579	ICM	7.5×10^{10}	0.04	1.9×10^{12}
GLP NHP toxicology ³ Study # ERBC - 20220013TCYP	ICM	High dose: 7.2×10^{13}	12	6×10^{12}
		Low dose: 1.4×10^{13}	12	1.2×10^{12}
Planned human trial dose ⁴	ICM	4.0×10^{14}	75	5.3×10^{12}

1. Volumes from Chen et al. 2023. PMID: 36606687
2. Neonatal mice – assume CSF volume scales with body weight. Note injection volume was 5ul to vector may have spread beyond the intended injection site
3. Primates were an average of 4.3kg - smaller than typical so we propose to reduce Chen’s 15ml estimate
4. Assuming 4–5-year-old patient. Please note that inclusion criteria will include 1–5-year-old patients

The high dose treatment that was validated as safe in non-human primate studies (cynomolgus monkeys), normalised to CSF volume, corresponds to a human dose of **4×10^{14} genome copies** in a 4-year-old child. Based on the adult dose response and non-human primate toxicology studies we propose a safe and

efficacious dose range that corresponds to 5×10^{12} to 6×10^{12} vector genome/ml of CSF. While significant phenotypic improvement was observed in mice, it's possible that a higher dosage may be necessary in humans to achieve comparable outcomes due to the mild phenotypes presented in the diseased mice. Therefore, we may utilise a higher dosage (6×10^{12} viral genomes/ml of CSF) that has been validated as safe in non-human primate studies.

As recommended by the referee, the dosage for human trials has been discussed on pages 26

#7- Which neural cell types are expressing the transgene under the ChB promoter?

The CBh promoter is a ubiquitous promoter. This promoter led to widespread gene transfer throughout the brain and spinal cord. Previous report demonstrated that AAV-CBh promoter mediate strong gene transfer in neuronal and glia cell types in various structures of the brain, spinal cord and dorsal root ganglia (DRG) (Gray et al. 2011, DOI: 10.1089/hum.2010.245).

As recommended by the reviewer, this point has been discussed on page 26 in the discussion section of the revised manuscript.

#8- Doses administered in the study are often referred to as VG. Please state dosing as vg/kg which is more informative

Thank you for highlighting this – this has been changed throughout the manuscript and the dose is stated as vg/Kg in the revised manuscript (various pages)

#9- Please check materials and methods for typos.

Typo bottom of page 12 'develop' to 'developed'.

Thank you for highlighting this, amendments have been made. Amendments to the methods section have been addressed, pages 25-41.

#10- Why is the bio distribution between KO-Cbh-hAP4B1 and KO-Syn-hAP4B1 different in Figure 5E. The promoter should not make a difference? There is no statistical analysis of this data to show if it is significant or not. Please comment?

We agree that theoretically the promoter should not make a difference in relation to the biodistribution. However, several factors could explain this finding: (1) Synapsin promoter is smaller than CBh and as mentioned previously vector size could affect conformational biology of the viral vector and viral yield thus influencing viral uptake via cells and efficacy; (2) variations in actual amount of injected AAV may occur due to pipetting errors or loss during ICM injection.

Please also refer to the answer to question 5 (Reviewer 1).

Statistical analysis was performed on Figure 5E between the CBh and SYN groups. Please see updated Figure 5E. The difference in viral genome copies of the CBh vs SYN vector for the cerebrum and brainstem is statistically significant. It is possible that this difference is due to the susceptibility of the cellular composition within each tissue structure to the conformation of each capsid.

This point has been discussed in the revised manuscript page 20

#11- Figs 3A-F bar graphs should also show individual data points for transparency. Also not statistical analysis. Same for Figure 6C

Thank you for highlighting this – individual data and statistical analysis have been added to the revised figures 3A-F and figure 6C.

16th Aug 2024

Dear Prof. Azzouz,

Thank you for the submission of your revised manuscript to EMBO Molecular Medicine and please accept my apologies for the delay in getting back to you due to the holiday season. I am pleased to inform you that we will be able to accept your manuscript pending the following final amendments:

1) Authors: We note name discrepancies in our submission system and in the manuscript. Zih-Liang Yang in the manuscript and Zig-Liang Yang in our system; Adrienne Angyal in the manuscript and Adrienn Angyal in our system. Please correct.

2) Figures:

- During a standard image analysis, we detected potential misplacement of panels in the figure EV2. Higher magnification images of CBH-DCN and SYN-DCN seem to be mixed up and CBH-Hoechst and SYN-Hoechst images seem to be identical. Please clarify and correct. Also, please make sure that all figures are accurate.

- Please rename all EV figures in the figure legends to Figure EV 1 etc.
- Remove "Schematic 1" from the manuscript file, move it to Appendix and rename it to Appendix Figure S3. Please call it out at the appropriate place in the manuscript.

3) In the main manuscript file, please do the following:

- Please address all comments suggested by our data editors listed below:

o Data availability section:

1. Please note that the accession ID for the Bioimage archive database is not provided in the data availability statement.

o Figure legends:

1. Please define the annotated p values ** as well as provide the exact p-values for the same in the legend of figure 8a; as appropriate.

2. Please note that the exact p values are not provided in the legends of figures 1e; 3b-d, h, j, l; 4a-b; 5a-b, d-f; 6b-g; 7a-b; EV 1c; EV 2; EV 4.

3. Please indicate the statistical test used for data analysis in the legends of figures 3b-f; 8a.

4. Please note that in figures 7a-b; EV 1b-c; EV 5b-c; there is a mismatch between the annotated p values in the figure legend and the annotated p values in the figure file that should be corrected.

5. Please note that information related to n is missing in the legend of figure 2b.

6. Although 'n' is provided, please describe the nature of entity for 'n' in the legends of figures 1e; 2c; 9a-c; EV 1b-c.

7. Please note that the error bars are not defined in the legends of figures 3a-f, h, j, l; 9a-c.

- Rename "Materials and Methods" to "Methods".

- Add callouts for Figure 9C. There is a callout for figure 10C that does not exist, please correct.

- In Methods, provide the statement that informed consent was obtained from all human subjects and confirm that the experiments conformed to the principles set out in the WMA Declaration of Helsinki and the Department of Health and Human Services Belmont Report.

- In Methods, add statistical paragraph that should reflect all information that you have filled in the Authors Checklist, especially regarding randomization, blinding, replication.

- Please include structured Methods section that includes a Reagents and Tools Table followed by a Methods and Protocols section. More information on how to adhere to this format as well as downloadable templates (.docx) for the Reagents and Tools Table can be found in our author guidelines: <https://www.embopress.org/page/journal/17574684/authorguide#structuredmethods>
An example of a paper with Structured Methods can be found here:

<https://www.embopress.org/doi/full/10.1038/s44320-024-00037-6#sec-4>

- Indicate in legends number and nature of replicates and exact p= values, not a range, along with the statistical test used. To keep the figures "clear" some authors found providing an Appendix table Sx with all exact p-values preferable. You are welcome to do this if you want to.

- In data availability statement please add the specific URLs for the deposited datasets.

4) Appendix: All tables in the appendix would have to be labeled Appendix Table S1 etc. which would create a confusion. I would like to suggest that appendix reports are presented as continuous Appendix Tables S1 and S2 with the current descriptions as table legends. See attached edited Appendix as an example. If you wish to present Male and female weight gain plotted over 35 weeks as a figure it should be named Appendix Figure S4 and cited in the main text. Please review and amend the Appendix and upload it as a single PDF file. Please also, update the callouts in the main manuscript file.

5) Funding: Please make sure that information about all sources of funding are complete in both our submission system and in the manuscript. Currently, European Research Council grant (ERC Advanced Award no. 294745, MRC DPFS Award (129016), JPND-MRC (MR/V000470/1), Spastic Paraplegia Foundation, the CureAP4 Foundation, the Boston Children's Hospital Translational Research Program, the Boston Children's Hospital Office of Faculty Development, the National Institutes of Health / National Institute of Neurological Disorders and Stroke (K08NS123552-01) are missing in our submission system.

6) The Paper Explained: Rename "Manuscript summary" to "The Paper Explained" and add it to the main manuscript file.

7) Synopsis: Please check your synopsis text and image before submission with your revised manuscript. Please be aware that in the proof stage minor corrections only are allowed (e.g., typos).

8) As part of the EMBO Publications transparent editorial process initiative (see our Editorial at

<http://embomolmed.embopress.org/content/2/9/329>), EMBO Molecular Medicine will publish online a Review Process File (RPF) to accompany accepted manuscripts. This file will be published in conjunction with your paper and will include the anonymous referee reports, your point-by-point response and all pertinent correspondence relating to the manuscript. Let us know whether you agree with the publication of the RPF and as here, if you want to remove or not any figures from it prior to publication. Please note that the Authors checklist will be published at the end of the RPF.

9) Please provide a point-by-point letter INCLUDING my comments as well as the reviewer's reports and your detailed responses (as Word file).

I look forward to reading a new revised version of your manuscript as soon as possible.

Yours sincerely,

Zeljko Durdevic

*** Instructions to submit your revised manuscript ***

1) a .docx formatted version of the manuscript text (including Figure legends and tables)

2) Separate figure files*

3) supplemental information as Expanded View and/or Appendix. Please carefully check the authors guidelines for formatting Expanded view and Appendix figures and tables at <https://www.embopress.org/page/journal/17574684/authorguide#expandedview>

4) a letter INCLUDING the reviewer's reports and your detailed responses to their comments (as Word file).

5) The paper explained: EMBO Molecular Medicine articles are accompanied by a summary of the articles to emphasize the major findings in the paper and their medical implications for the non-specialist reader. Please provide a draft summary of your article highlighting

6) For more information: There is space at the end of each article to list relevant web links for further consultation by our readers. Could you identify some relevant ones and provide such information as well? Some examples are patient associations, relevant databases, OMIM/proteins/genes links, author's websites, etc...

7) Author contributions: the contribution of every author must be detailed in a separate section.

8) EMBO Molecular Medicine now requires a complete author checklist (<https://www.embopress.org/page/journal/17574684/authorguide>) to be submitted with all revised manuscripts. Please use the checklist as guideline for the sort of information we need WITHIN the manuscript. The checklist should only be filled with page numbers where the information can be found. This is particularly important for animal reporting, antibody dilutions (missing) and exact values and n that should be indicated instead of a range.

9) Every published paper now includes a 'Synopsis' to further enhance discoverability. Synopses are displayed on the journal webpage and are freely accessible to all readers. They include a short stand first (maximum of 300 characters, including space) as well as 2-5 one sentence bullet points that summarise the paper. Please write the bullet points to summarise the key NEW findings. They should be designed to be complementary to the abstract - i.e. not repeat the same text. We encourage inclusion of key acronyms and quantitative information (maximum of 30 words / bullet point). Please use the passive voice. Please attach these in a separate file or send them by email, we will incorporate them accordingly.

You are also welcome to suggest a striking image or visual abstract to illustrate your article. If you do please provide a jpeg file 550 px-wide x 300-600px high.

10) A Conflict of Interest statement should be provided in the main text

11) Please note that we now mandate that all corresponding authors list an ORCID digital identifier. This takes <90 seconds to complete. We encourage all authors to supply an ORCID identifier, which will be linked to their name for unambiguous name identification.

Currently, our records indicate that the ORCID for your account is 0000-0001-6564-5967.

Link Not Available

12) Include a Reagents and Tools Table as part of the Methods section, which can be downloaded from our author guidelines (<https://www.embopress.org/page/journal/17574684/authorguide#structuredmethods>)

Photos 400-800 DPI

*Additional important information regarding figures and illustrations can be found at <https://bit.ly/EMBOPressFigurePreparationGuideline>. See also figure legend preparation guidelines: <https://www.embopress.org/page/journal/17574684/authorguide#figureformat>

***** Reviewer's comments *****

Referee #1 (Remarks for Author):

The authors have adequately addressed my comments in this revised version of the manuscript. Therefore, I have no further comments.

Referee #2 (Comments on Novelty/Model System for Author):

This is an interesting gene therapy study and the authors have now also informed that the data presented has also been submitted to regulatory bodies for assessment and movement to clinical trial. So the timing is right for this to be published. The paper is overall well written and clear. I believe that this is now suitable for publication in EMBO Mol Med.

Referee #2 (Remarks for Author):

The Authors have responded to all my questions and suggestions and amended the manuscript accordingly and where appropriate. The manuscript has benefitted through this process.

The authors addressed the minor editorial issues.

19th Sep 2024

Dear Prof. Azzouz,

We are pleased to inform you that your manuscript is accepted for publication and is now being sent to our publisher to be included in the next available issue of EMBO Molecular Medicine.
